# MMLONGBENCH: Benchmarking Long-Context Vision-Language Models Effectively and Thoroughly

**Zhaowei Wang**[1]    **Wenhao Yu**[2]    **Xiyu Ren**[1]    **Jipeng Zhang**[1]    **Yu Zhao**[3]
**Rohit Saxena**[3]    **Liang Cheng**[3]    **Ginny Wong**[5]    **Simon See**[5]
**Pasquale Minervini**[3,4]    **Yangqiu Song**[1]    **Mark Steedman**[3]
[1]CSE Department, HKUST    [2]Tencent AI Seattle Lab    [3]University of Edinburgh    [4]Miniml.AI
[5]NVIDIA AI Technology Center (NVAITC), NVIDIA, Santa Clara, USA
{zwanggy, yqsong}@cse.ust.hk    m.steedman@ed.ac.uk

## Abstract

The rapid extension of context windows in large vision-language models has given rise to *long-context vision-language models* (LCVLMs), which are capable of handling hundreds of images with interleaved text tokens in a single forward pass. In this work, we introduce MMLONGBENCH, the first benchmark covering a diverse set of long-context vision-language tasks, to evaluate LCVLMs effectively and thoroughly. MMLONGBENCH is composed of 13,331 examples spanning five different categories of downstream tasks, such as Visual RAG and Many-Shot ICL. It also provides broad coverage of image types, including various natural and synthetic images. To assess the robustness of the models to different input lengths, all examples are delivered at five standardized input lengths (8K–128K tokens) via a cross-modal tokenization scheme that combines vision patches and text tokens. Through a thorough benchmarking of 46 closed-source and open-source LCVLMs, we provide a comprehensive analysis of the current models' vision-language long-context ability. Our results show that: i) performance on a single task is a weak proxy for overall long-context capability; ii) both closed-source and open-source models face challenges in long-context vision-language tasks, indicating substantial room for future improvement; iii) models with stronger reasoning ability tend to exhibit better long-context performance. By offering wide task coverage, various image types, and rigorous length control, MMLONGBENCH[1] provides the missing foundation for diagnosing and advancing the next generation of LCVLMs.

## 1 Introduction

Recent advances in long-context modeling have unlocked a wide array of new capabilities for both large language models [LLMs; 1, 2] and large vision–language models [LVLMs; 3, 4]. In particular, long-context vision–language models (LCVLMs) represent an important step forward by enabling LVLMs to process hundreds of images and thousands of interleaved text tokens in a single forward pass. This allows applications such as document-level visual question answering [5], multi-hop reasoning across web pages [6], and instruction following grounded in complex visual contexts [7, 8].

To support such capabilities, researchers have proposed various techniques to extend the context windows of LVLMs, as seen in models such as LongVILA [9] and GPT-4o [10]. However, the development of effective evaluation benchmarks is lagging behind. It remains unclear how well current LCVLMs perform in long-context settings, what types of tasks they struggle with, and how robust they are to input length variation. Here, we take a closer look and find that existing benchmarks suffer from the following shortcomings and provide a summary of key feature comparisons in Table 1:

---

[1]The code and data are available at `https://github.com/EdinburghNLP/MMLongBench`.

Table 1: Comparison of benchmarks for LCVLMs: MM-NIAH [18], Visual Haystack [16], MM-Needle [11], MMLongBench-Doc (MMLB-Doc) [5], M-Longdoc [21], LongDocURL [17], and our MMLONGBENCH. "Summ" and "DocVQA" refer to summarization and long-document visual question answering (VQA). "Mixed" indicates both natural and synthetic images. "$L$" denotes the number of input tokens, and "$L$ Control" means counting both text and image tokens together.

| | Type of tasks | | | | | Benchmark features | | |
| --- | --- | --- | --- | --- | --- | --- | --- | --- |
| | VRAG | NIAH | ICL | Summ | DocVQA | Image Type | $L$ Control | Multiple $L$ |
| MM-NIAH [18] | ✗ | ✓ | ✗ | ✗ | ✗ | Mixed | ✓ | ✗ |
| Visual Haystack [16] | ✗ | ✓ | ✗ | ✗ | ✗ | Natural | ✗ | ✓ |
| MMNeedle [11] | ✗ | ✓ | ✗ | ✗ | ✗ | Natural | ✗ | ✓ |
| MMLB-Doc [5] | ✗ | ✗ | ✗ | ✗ | ✓ | Synthetic | ✗ | ✗ |
| M-Longdoc [21] | ✗ | ✗ | ✗ | ✗ | ✓ | Synthetic | ✗ | ✗ |
| LongDocURL [17] | ✗ | ✗ | ✗ | ✗ | ✓ | Synthetic | ✗ | ✗ |
| MMLONGBENCH (Ours) | ✓ | ✓ | ✓ | ✓ | ✓ | Mixed | ✓ | ✓ |

- **Limited coverage of downstream tasks**: Existing benchmarks predominantly focus on a single type of long-context vision-language task, such as needle-in-a-haystack (NIAH) [11] or long-document VQA [5]. However, performance on a single type of task cannot reflect the broader long-context visual reasoning capabilities required for various downstream applications [12]. Other applications, like long-context RAG [13], many-shot in-context learning [14], and long-document summarization [15] are entirely absent from current evaluations.

- **Insufficient coverage of image types**: Most existing benchmarks are restricted to either natural images [11, 16] — such as photographs of everyday scenes, objects, or people — or synthetic images [5, 17], such as scanned documents, web pages, or app screenshots. This limited focus leads to an incomplete understanding of model performance across diverse image types.

- **Lack of context length control**: Existing benchmarks miss a consensus on cross-modality length control, especially image tokens. For example, while MM-NIAH [18] follows InternVL1.5 [19] to count text and image tokens together, other works, such as Visual Haystack [16] and Long-DocURL [17], only report the number of images as the context length. This inconsistency makes it difficult to compare model performance across different benchmarks.

- **Single context length**: Many long-context benchmarks with text-only inputs standardize context lengths to a few values (e.g., 8K, 32K, 128K) and provide each example with multiple contexts at those lengths [12, 20]. Hence, model developers can easily know the performance change with different lengths. However, such practice is not followed in LCVLM evaluations. For example, in MM-NIAH [18], models are evaluated on long essays (such as web pages) with randomly varying lengths, which complicates systematic analysis of context length effects.

To enable comprehensive evaluation, in this paper, we introduce MMLONGBENCH, a benchmark covering a diverse set of long-context vision-language tasks across five different categories. Specifically, in addition to multimodal *needle-in-a-haystack* (NIAH) and *long-document VQA* (DocVQA), we also include *visual retrieval-augmented generation* (VRAG), *many-shot in-context learning* (ICL), and *summarization* in our benchmark. VRAG examples are drawn from knowledge-based VQA datasets [22, 23] using Wikipedia passages to populate long contexts. ICL examples are image classification problems in four domains [24–27], which require models to perform on-the-fly classification based on hundreds of in-context exemplars. In the summarization task, models are required to summarize image-based PDF documents [15, 28]. Overall, our benchmark includes diverse downstream tasks and image types to enable a comprehensive evaluation.

In MMLONGBENCH, we use a unified token-count method which counts image tokens based on the number of patches produced by current vision encoders, followed by a $2 \times 2$ pixel unshuffle. This approach is consistent with practices adopted in most recent models, such as Qwen2.5-VL [29] and InternVL3 [30], making it well-suited for long-context evaluation. For a thorough evaluation of input length, we equip all examples with five standardized lengths of context, ranging from 8K to 128K tokens, enabling a thorough analysis of performance changes as the context length increases. Further, we also ensure all datasets are easily extendable to longer contexts.

Finally, to understand the progress of LCVLMs and how different multimodal long-context capabilities correlate with one another, we evaluated 46 models of various architectures, scales, and

training approaches. Our analysis reveals three key findings: i) performance on a single task poorly reflects overall long-context ability; ii) while closed-source models show higher scores, long-context vision-language tasks present significant challenges for both closed-source and open-source models, highlighting the need for future improvements; and iii) models with stronger reasoning ability tend to exhibit better long-context capabilities, as exemplified by the thinking versions of Gemini models. Furthermore, our error analysis shows that Optical Character Recognition (OCR) and cross-modality retrieval abilities remain bottlenecks in current LCVLMs. Overall, our benchmark underscores the importance of evaluating LCVLMs across comprehensive long-context vision-language tasks. We hope these insights will help guide future model development and evaluation.

## 2 Related Work

**Long-context vision-language models (LCVLMs).** The context window of LLMs has experienced a fast growth from less than 8K [31, 32] to 128K tokens [1, 10] or more [33, 34]. To support this, techniques such as longer pre-training length [1, 35, 36], position extrapolation [37–39], and efficient architectures [40–42] have been developed. With this progress, recent literature also investigated how to extend the context length of LVLMs to build LCVLMs, such as Gemini-2.5 [43], Qwen2.5-VL [29], and others [3, 30, 44, 45]. In addition, several recent works on LVLMs made efforts to compress vision tokens to accommodate longer input sequences [46–51]. Meanwhile, a growing body of works has adopted various techniques from LLMs to extend LVLMs' context length, such as position extrapolation [52] and more efficient model architectures [53]. With extended context lengths, LCVLMs can support various applications, such as multi-hop reasoning across web pages [6] and instruction following grounded in complex visual contexts [7, 54, 55].

**Long-context benchmarks.** Needle-in-a-haystack (NIAH) [56] is one of the first commonly adopted tasks to evaluate the text-pure long context ability of LLMs, as it can be procedurally generated with arbitrarily long lengths and needle position [57]. This task inserts a "needle" at specific depths of a long essay and tests models' ability to recall it. Recent works have also extended the NIAH task to more complex versions [58–60]. However, several benchmarks [12, 20] discover that using a single NIAH task only partially reflects LLMs' overall long-context ability. As a result, numerous benchmarks with broad coverage of diverse downstream applications have been constructed [12, 14, 20, 61–64] to provide a comprehensive evaluation.

In contrast, the evaluation of LVLMs' long-context capability remains limited. Existing benchmarks only involve either NIAH [11, 16, 18, 65] or long-document VQA [5, 17], lacking comprehensive coverage across diverse vision-language applications. As a result, frontier LCVLMs [10, 33] only report long-context performance on other modalities, such as long video [9, 66, 67] or long audio [10, 33], and neglect the prevalent use cases of long-context vision-language inputs. While recent MileBench [68] claims to be a comprehensive long-context benchmark with various text-image tasks, our closer inspection reveals that it actually contains a lot of short-context tasks, and the average length is only about 9K tokens. Datasets like DocVQA [69], WebQA [70], and OCRVQA [71] contain only one image per sample and minimal context, making MileBench unqualified as a true long-context benchmark. In this work, beyond existing long video and audio benchmarks, we introduce the first comprehensive benchmark that evaluates a wide range of vision-language downstream applications.

## 3 Our Benchmark — MMLONGBENCH

In our work, we seek to address the limitations of current benchmarks by meeting the following criteria: i) broad coverage of both diverse vision-language downstream tasks and different image types, ii) a unified token-count method across different modalities and datasets, and iii) multiple standardized context lengths for each example, ranging from 8K to 128K tokens. In this section, we describe the task categories and datasets included in MMLONGBENCH, highlighting how they address the limitations of existing evaluation benchmarks. An overview of MMLONGBENCH is provided in Table 2, and several ***concrete examples*** are given in Section E.

### 3.1 Diverse Long-Context Applications for LCVLMs

**Visual retrieval-augmented generation (VRAG)** evaluates an LCVLM's ability to ground on relevant information retrieved from a large corpus, while filtering out distractors and irrelevant

Table 2: Overview of datasets in MMLONGBENCH. We include datasets covering key long-context capabilities, with 13,331 examples in total. Image types are shown per dataset; "Mixed" indicates both natural and synthetic images. SubEM and Acc indicate substring exact match and accuracy.

| Category | Dataset | Metrics | Image | Size | Description |
|---|---|---|---|---|---|
| **Visual RAG** | InfoSeek | SubEM | Natural | 1,128 | Long-tail entity question answering |
| | ViQuAE | SubEM | Natural | 1,144 | Question answering based on TriviaQA |
| **Needle-in-a-Haystack** | VH-Single | Acc | Natural | 1,000 | Retrieve an image from an album |
| | VH-Multi | Acc | Natural | 1,000 | Retrieve multiple images from an album |
| | MM-NIAH-Ret | SubEM/Acc | Mixed | 1,200 | Retrieve text/image needles in web pages |
| | MM-NIAH-Count | Acc | Mixed | 1,178 | Count text/image needles in web pages |
| | MM-NIAH-Reason | SubEM/Acc | Mixed | 1,158 | Reason about text/image needles in web pages |
| **Many-Shot In-Context Learning** | Stanford Cars | Acc | Natural | 458 | 50-category car classification |
| | Food101 | Acc | Natural | 500 | 50-category food classification |
| | SUN397 | Acc | Natural | 500 | 50-category scene classification |
| | iNat2021 | Acc | Natural | 500 | 50-category species classification |
| **Summarization** | GovReport | Model-based | Synthetic | 241 | Summarizing government reports in PDF |
| | Multi-LexSum | Model-based | Synthetic | 146 | Summarizing multiple legal documents in PDF |
| **Long-Document VQA** | MMLongBench-Doc | SubEM/Acc | Synthetic | 961 | Long PDF document VQA |
| | LongDocURL | SubEM/Acc | Synthetic | 1,153 | Long PDF document VQA |
| | SlideVQA | SubEM/Acc | Synthetic | 1,064 | Slide deck understanding and reasoning |

content. To evaluate this capability, we use factual knowledge-based VQA as a representative task. This task requires answering questions about the named entity identified in an image, such as "Who designed the building in this picture?" We include InfoSeek [22] and ViQuAE [23] in this category.

To build a long context, we insert the gold passage(s) (the passage with the answer) among a large set of distracting passages retrieved from Wikipedia. For ViQuAE, we use gold passages from KILT [72] since it is constructed upon TriviaQA [73]. For InfoSeek, we choose the lead section of the named entity's Wikipedia page as the gold reference and remove all examples for which the answer cannot be found in the lead section. Then, we split Wikipedia pages into 100-word passages and incrementally add retrieved passages that do not contain the answer or the named entity as distractors until we reach the given input length $L$. For retrieval, we use the named entity instead of the image itself as the query, because text-based retrieval achieves higher recall and provides harder distractors. In ViQuAE, each example contains a single gold passage, and we insert it at six evenly distributed positions, while in Infoseek, the lead sections often contain hundreds of words, resulting in multiple passages. We randomly shuffle them into three permutations. We use the substring exact match (SubEM) as the metric, following previous work [74]. See more details in Section A.1.

**Needle-in-a-haystack (NIAH)** measures how well an LCVLM can recall a small but important piece of information embedded within a long sequence of mostly unrelated visual and textual inputs. NIAH tasks have been widely adopted because they are easy to build (can be procedurally generated with a long corpus) and simple to control (can combine any context length and needle position). For this category, we select multiple tasks from Visual Haystack [VH; 16] and MM-NIAH [18]. VH requires models to retrieve images of target objects (the needle) in an image haystack. It is available in two versions: VH-Single and VH-Multi, for finding a single image and multiple images. MM-NIAH contains retrieval (Ret), counting (Count), and reasoning (Reason) tasks in interleaved text and images; each task features both text and image needles.

In VH, we obtain the needle images and the target objects from the original dataset. Then, we accompany these needles with multiple negative distractor images until the image haystack reaches a given input length $L$. We report accuracy as the metric following the original work [16]. In MM-NIAH, the haystacks are composed of web documents [75] with interleaved text and images. We include all three tasks of retrieval, counting, and reasoning in our benchmark. Similar to VRAG, we split text in each web document into 100-word passages and add passages and images so that the context achieves the input length. We use SubEM and accuracy to evaluate retrieval and reasoning tasks following the original paper [18]. In the counting task, we use the accuracy of summed needle counts for better robustness. Refer to Section A.2 for more details.

**Many-shot in-context learning (ICL)** tests the model's capability to adapt to new multimodal tasks on the fly by observing multiple in-context exemplars, without requiring any parameter updates. Following prior work on long-context LLMs [20, 76, 77], we focus on image classification datasets with large label spaces. Here, we collect four datasets with diverse domains: Stanford Cars [24] for cars, Food101 [25] for food, SUN397 [78] for scenes, and iNat2021 [27] for species. We adjust the

number of shots to control the input length $L$, and the number of exemplars in each class is balanced. The 128K context window can accommodate approximately 500 images. To ensure sufficient shots per class, we randomly sample 50 classes from each dataset. We report accuracy on each dataset.

One difference from existing work on many-shot ICL with LCVLMs [79] is that we map the original natural language labels (e.g., food names) to class IDs (e.g., 0, 1, ...), requiring models to learn new tasks rather than relying on pre-training knowledge. Section A.3 covers more details.

**Summarization (Summ)** evaluates an LCVLM's ability to generate concise outputs from long multimodal documents while preserving all salient information. We choose GovReport [15] (government report summarization) and Multi-LexSum [28] (multi-document legal summarization), as their PDF-formatted documents are long and easily accessible. Our evaluation provides models with PDF-formatted documents rather than OCR-extracted text used in previous works [80, 81]. We truncate the document from the end based on the input length $L$. Following previous work [20], we use LLM-based evaluation for both datasets instead of the commonly used ROUGE-L, as it better reflects human judgment. More details, such as the LLM-based metric, are provided in Section A.4.

**Long-document VQA (DocVQA)** assesses the model's aptitude for answering questions that require reasoning over information dispersed across multiple images and text segments within an extended document. We include commonly adopted datasets for evaluating long-document VQA: SlideVQA [82], MMLongBench-Doc [5], and LongDocURL [17]. For documents longer than input length $L$, we truncate the documents evenly from both sides while keeping the answer pages. For shorter documents, we alternately pad the left and right sides with randomly sampled negative documents up to length $L$. However, the padding documents may occasionally contain information related to the question and potentially change the answer. To ensure the validity of questions, we preface each question with the prompt "Based on the Document *<Original Doc ID>*, answer the following question." We follow the metrics used in LongDocURL but remove questions with long answers, thereby avoiding LLM-based answer extraction. We list specific details in Section A.5.

### 3.2 Cross-Modality Token Counting

Various long-context applications of LCVLMs usually involve varying text-to-image ratios. For example, VRAG contains only one image related to a named entity, whereas the context in Long-Document VQA primarily consists of images. When building a comprehensive benchmark for LCVLMs, the initial challenge lies in standardizing the context length of diverse datasets with different text-image combinations. In this work, we count both text tokens and visual tokens together as the total input length of $L$, in contrast to prior works [11, 16, 65] that simply use the image number as context length. We use the Llama2 tokenizer [31] to calculate the number of text tokens following previous practice [20]. To count image tokens, we divide each image into $14 \times 14$ patches and apply a $2 \times 2$ pixel unshuffle to compress the visual token number. Note that this patch size and the pixel unshuffle operation are both commonly adopted in current LVLMs [19, 29, 30, 44, 47, 51, 83, 84]. This method ensures compatibility with modern LVLMs, making it well-suited for evaluation.

### 3.3 Standardized Input Length

The input length $L$ is an important factor to consider when we evaluate models' long-context ability, as longer inputs can provide more information but also challenge models to filter out distracting information. As aforementioned in Section 3.1, we can control the input length $L$ for each dataset either by adjusting the number of passages, images, or exemplars, or by truncating the PDF-formatted documents. This allows us to present each example in our benchmark under multiple standardized input lengths and better understand how performance changes as the context length increases. Specifically, our benchmark provides five input lengths $L$: 8K, 16K, 32K, 64K, and 128K tokens, using binary prefixes $K = 2^{10}$, and the input length can be easily extended beyond 128K if needed.

## 4 Evaluation and Analysis

With broad task coverage, unified token counting, and standardized input length, we are now able to thoroughly examine LCVLMs' long-context ability across multiple dimensions. In total, we evaluate 46 LCVLMs on MMLONGBENCH. To the best of our knowledge, our evaluation provides the most thorough and controlled comparison of the vision-language long-context ability on broad real-world

**VRAG**

| Model | 8k | 16k | 32k | 64k | 128k |
|---|---|---|---|---|---|
| GPT-4o | 80.5 | 74.7 | 71.8 | 74.2 | 67.3 |
| Claude-3.7-Sonnet | 84.9 | 81.8 | 66.7 | 67.6 | 68.8 |
| Gemini-2.0-Flash | 64.9 | 64.2 | 59.5 | 59.0 | 60.3 |
| Gemini-2.0-Flash-T | 67.0 | 68.5 | 66.7 | 67.0 | 64.4 |
| Gemini-2.5-Flash | 69.8 | 69.3 | 65.1 | 68.6 | 70.6 |
| Gemini-2.5-Pro | 79.8 | 80.9 | 79.9 | 80.8 | 82.7 |
| Qwen2-VL-72B | 64.3 | 64.0 | 60.1 | 56.1 | 46.9 |
| Qwen2.5-VL-7B | 50.1 | 48.7 | 43.2 | 36.8 | 31.6 |
| Qwen2.5-VL-32B | 67.8 | 69.1 | 65.5 | 61.9 | 64.6 |
| Qwen2.5-VL-72B | 67.6 | 67.7 | 64.0 | 54.3 | 50.3 |
| InternVL2.5-26B | 56.6 | 53.3 | 48.1 | 50.0 | 47.9 |
| InternVL3-8B | 52.3 | 51.3 | 45.8 | 40.3 | 36.3 |
| InternVL3-14B | 57.5 | 55.3 | 52.3 | 52.8 | 50.0 |
| InternVL3-38B | 65.7 | 60.8 | 52.2 | 50.4 | 40.3 |
| Ovis2-8B | 52.3 | 48.0 | 47.1 | 47.9 | 42.9 |
| Ovis2-16B | 56.2 | 51.2 | 49.7 | 49.2 | 41.3 |
| Ovis2-34B | 63.4 | 61.5 | 55.5 | 57.2 | 45.7 |
| Gemma3-12B | 58.6 | 52.1 | 46.9 | 43.5 | 41.7 |
| Gemma3-27B | 64.8 | 62.1 | 58.8 | 57.5 | 51.5 |
| Idefics3-8B | 33.3 | 31.8 | 30.3 | 35.2 | 33.2 |
| Phi-4-Multimodal | 36.3 | 37.3 | 35.4 | 32.9 | 25.5 |
| NVILA-Lite-8B | 43.2 | 41.6 | 41.8 | 35.8 | 16.3 |
| Pixtral-12B | 53.6 | 51.0 | 47.9 | 45.9 | 43.8 |

**NIAH**

| Model | 8k | 16k | 32k | 64k | 128k |
|---|---|---|---|---|---|
| GPT-4o | 79.6 | 73.8 | 67.5 | 65.4 | 57.1 |
| Claude-3.7-Sonnet | 63.1 | 61.2 | 54.1 | N/A | N/A |
| Gemini-2.0-Flash | 76.8 | 74.1 | 69.7 | 64.6 | 60.9 |
| Gemini-2.0-Flash-T | 80.8 | 79.2 | 76.2 | 68.7 | 64.8 |
| Gemini-2.5-Flash | 84.1 | 81.5 | 79.8 | 76.4 | 72.5 |
| Gemini-2.5-Pro | 84.7 | 82.7 | 79.8 | 76.0 | 73.4 |
| Qwen2-VL-72B | 63.9 | 61.6 | 57.4 | 51.5 | 38.9 |
| Qwen2.5-VL-7B | 57.3 | 53.0 | 47.7 | 39.5 | 33.2 |
| Qwen2.5-VL-32B | 61.9 | 61.1 | 58.5 | 53.7 | 41.6 |
| Qwen2.5-VL-72B | 68.3 | 63.5 | 61.9 | 55.8 | 43.1 |
| InternVL2.5-26B | 67.8 | 63.1 | 55.5 | 52.2 | 43.8 |
| InternVL3-8B | 62.6 | 57.8 | 51.8 | 49.7 | 42.4 |
| InternVL3-14B | 69.5 | 65.1 | 58.2 | 55.8 | 48.3 |
| InternVL3-38B | 70.5 | 66.4 | 62.5 | 57.0 | 52.0 |
| Ovis2-8B | 61.3 | 57.9 | 54.2 | 41.2 | 35.8 |
| Ovis2-16B | 67.3 | 62.7 | 56.5 | 48.7 | 40.7 |
| Ovis2-34B | 65.7 | 60.4 | 57.0 | 52.9 | 40.0 |
| Gemma3-12B | 60.7 | 55.9 | 51.4 | 47.5 | 41.7 |
| Gemma3-27B | 66.3 | 61.2 | 56.2 | 51.9 | 44.6 |
| Idefics3-8B | 49.2 | 45.2 | 43.1 | 39.6 | 37.5 |
| Phi-4-Multimodal | 48.8 | 44.6 | 41.1 | 36.7 | 34.9 |
| NVILA-Lite-8B | 52.7 | 47.8 | 43.6 | 36.8 | 29.0 |
| Pixtral-12B | 56.3 | 54.2 | 50.2 | 45.2 | 40.9 |

**ICL**

| Model | 8k | 16k | 32k | 64k | 128k |
|---|---|---|---|---|---|
| GPT-4o | 99.0 | 98.2 | 96.0 | 92.4 | 88.4 |
| Claude-3.7-Sonnet | 97.0 | 94.2 | N/A | N/A | N/A |
| Gemini-2.0-Flash | 99.0 | 97.8 | 97.5 | 93.8 | 87.5 |
| Gemini-2.0-Flash-T | 99.5 | 97.8 | 96.2 | 92.5 | 88.2 |
| Gemini-2.5-Flash | 98.5 | 98.5 | 96.5 | 94.0 | 88.0 |
| Gemini-2.5-Pro | 99.5 | 98.5 | 97.2 | 95.0 | 94.2 |
| Qwen2-VL-72B | 98.5 | 94.5 | 91.0 | 80.8 | 80.8 |
| Qwen2.5-VL-7B | 95.6 | 91.5 | 78.5 | 57.2 | 46.2 |
| Qwen2.5-VL-32B | 97.5 | 91.7 | 77.0 | 51.2 | 41.2 |
| Qwen2.5-VL-72B | 98.5 | 95.5 | 92.8 | 74.2 | 73.0 |
| InternVL2.5-26B | 98.5 | 89.2 | 85.0 | 72.5 | 54.0 |
| InternVL3-8B | 97.6 | 87.2 | 75.0 | 61.8 | 8.5 |
| InternVL3-14B | 96.5 | 87.7 | 80.0 | 65.8 | 53.0 |
| InternVL3-38B | 99.5 | 95.0 | 88.5 | 77.5 | 65.2 |
| Ovis2-8B | 94.5 | 44.4 | 7.8 | 4.0 | 1.0 |
| Ovis2-16B | 96.6 | 91.2 | 73.2 | 66.0 | 36.5 |
| Ovis2-34B | 98.5 | 89.5 | 79.2 | 71.0 | 65.2 |
| Gemma3-12B | 99.0 | 96.5 | 93.2 | 82.2 | 59.0 |
| Gemma3-27B | 98.0 | 94.8 | 93.5 | 83.8 | 73.8 |
| Idefics3-8B | 25.6 | 12.3 | 4.5 | 0.8 | 2.0 |
| Phi-4-Multimodal | 82.3 | 42.5 | 12.0 | 2.8 | 2.2 |
| NVILA-Lite-8B | 93.1 | 73.6 | 47.0 | 20.5 | 2.8 |
| Pixtral-12B | 95.0 | 90.0 | 86.0 | 53.2 | 49.8 |

**Summ**

| Model | 8k | 16k | 32k | 64k | 128k |
|---|---|---|---|---|---|
| GPT-4o | 25.1 | 31.1 | 34.3 | 41.0 | 42.4 |
| Claude-3.7-Sonnet | 27.6 | 34.6 | 34.9 | 34.5 | 37.5 |
| Gemini-2.0-Flash | 24.4 | 27.1 | 30.1 | 30.6 | 35.9 |
| Gemini-2.0-Flash-T | 27.7 | 37.9 | 44.3 | 53.0 | 61.2 |
| Gemini-2.5-Flash | 29.2 | 39.4 | 45.9 | 55.3 | 62.4 |
| Gemini-2.5-Pro | 32.0 | 42.8 | 48.1 | 58.0 | 65.3 |
| Qwen2-VL-72B | 25.1 | 29.2 | 32.7 | 37.6 | 39.1 |
| Qwen2.5-VL-7B | 23.5 | 29.1 | 30.8 | 32.7 | 39.3 |
| Qwen2.5-VL-32B | 22.8 | 26.3 | 25.8 | 23.0 | 25.2 |
| Qwen2.5-VL-72B | 20.5 | 26.9 | 31.1 | 38.0 | 28.5 |
| InternVL2.5-26B | 19.1 | 23.8 | 26.3 | 27.8 | 29.5 |
| InternVL3-8B | 22.2 | 28.6 | 32.5 | 36.6 | 40.8 |
| InternVL3-14B | 22.3 | 25.6 | 27.2 | 30.3 | 35.8 |
| InternVL3-38B | 20.7 | 24.8 | 33.1 | 38.4 | 43.6 |
| Ovis2-8B | 23.0 | 29.3 | 30.5 | 32.9 | 28.3 |
| Ovis2-16B | 25.3 | 30.0 | 33.5 | 37.0 | 39.3 |
| Ovis2-34B | 23.5 | 29.8 | 35.7 | 39.6 | 41.6 |
| Gemma3-12B | 21.0 | 24.0 | 25.2 | 26.1 | 28.0 |
| Gemma3-27B | 22.9 | 28.5 | 32.0 | 35.5 | 40.7 |
| Idefics3-8B | 15.7 | 20.4 | 19.2 | 21.8 | 17.7 |
| Phi-4-Multimodal | 12.3 | 17.4 | 17.5 | 18.8 | 15.9 |
| NVILA-Lite-8B | 12.8 | 15.3 | 19.3 | 19.9 | 23.3 |
| Pixtral-12B | 22.7 | 29.6 | 33.5 | 36.7 | 38.5 |

**DocVQA**

| Model | 8k | 16k | 32k | 64k | 128k |
|---|---|---|---|---|---|
| GPT-4o | 67.8 | 70.5 | 67.2 | 62.9 | 59.2 |
| Claude-3.7-Sonnet | 56.7 | 52.0 | 43.1 | 48.5 | N/A |
| Gemini-2.0-Flash | 58.7 | 55.4 | 59.4 | 53.8 | 53.6 |
| Gemini-2.0-Flash-T | 68.1 | 68.8 | 69.9 | 64.3 | 63.7 |
| Gemini-2.5-Flash | 67.5 | 66.9 | 68.6 | 62.5 | 59.3 |
| Gemini-2.5-Pro | 71.5 | 70.0 | 70.8 | 69.2 | 70.4 |
| Qwen2-VL-72B | 69.2 | 65.7 | 66.4 | 60.9 | 53.8 |
| Qwen2.5-VL-7B | 60.7 | 57.1 | 57.2 | 50.7 | 40.2 |
| Qwen2.5-VL-32B | 67.8 | 66.0 | 65.8 | 58.4 | 53.6 |
| Qwen2.5-VL-72B | 71.4 | 67.5 | 65.8 | 57.3 | 48.7 |
| InternVL2.5-26B | 53.5 | 47.6 | 51.4 | 44.6 | 32.8 |
| InternVL3-8B | 58.1 | 53.7 | 55.3 | 48.7 | 42.6 |
| InternVL3-14B | 63.3 | 54.1 | 57.5 | 50.0 | 39.4 |
| InternVL3-38B | 66.3 | 63.8 | 62.9 | 52.2 | 47.9 |
| Ovis2-8B | 59.1 | 49.3 | 42.3 | 30.3 | 10.9 |
| Ovis2-16B | 66.5 | 61.2 | 48.5 | 35.4 | 19.3 |
| Ovis2-34B | 59.9 | 55.2 | 45.2 | 33.6 | 23.5 |
| Gemma3-12B | 42.7 | 43.2 | 43.2 | 39.2 | 41.3 |
| Gemma3-27B | 49.7 | 49.7 | 45.5 | 46.2 | 45.6 |
| Idefics3-8B | 46.3 | 37.1 | 42.0 | 26.4 | 17.3 |
| Phi-4-Multimodal | 44.5 | 45.5 | 47.9 | 41.7 | 26.0 |
| NVILA-Lite-8B | 30.8 | 32.4 | 25.8 | 21.6 | 20.6 |
| Pixtral-12B | 55.0 | 48.1 | 44.4 | 38.7 | 32.4 |

**Avg.**

| Model | 8k | 16k | 32k | 64k | 128k |
|---|---|---|---|---|---|
| GPT-4o | 70.4 | 69.7 | 67.4 | 67.2 | 62.9 |
| Claude-3.7-Sonnet | 65.9 | 64.8 | N/A | N/A | N/A |
| Gemini-2.0-Flash | 64.8 | 63.7 | 63.2 | 60.3 | 59.6 |
| Gemini-2.0-Flash-T | 68.6 | 70.4 | 70.6 | 69.1 | 68.5 |
| Gemini-2.5-Flash | 69.8 | 71.1 | 71.2 | 71.4 | 70.5 |
| Gemini-2.5-Pro | 73.5 | 75.0 | 75.2 | 75.8 | 77.2 |
| Qwen2-VL-72B | 64.2 | 63.0 | 61.5 | 57.4 | 51.9 |
| Qwen2.5-VL-7B | 57.4 | 55.9 | 51.5 | 43.4 | 38.1 |
| Qwen2.5-VL-32B | 63.6 | 62.9 | 58.5 | 49.7 | 45.2 |
| Qwen2.5-VL-72B | 65.2 | 64.2 | 63.1 | 55.9 | 48.7 |
| InternVL2.5-26B | 59.1 | 55.4 | 53.3 | 49.4 | 41.6 |
| InternVL3-8B | 58.5 | 55.7 | 52.1 | 47.4 | 34.1 |
| InternVL3-14B | 61.8 | 57.5 | 55.1 | 50.9 | 45.3 |
| InternVL3-38B | 64.5 | 62.1 | 59.9 | 55.1 | 49.8 |
| Ovis2-8B | 58.0 | 45.8 | 36.4 | 31.3 | 23.8 |
| Ovis2-16B | 62.4 | 59.3 | 52.3 | 47.3 | 35.4 |
| Ovis2-34B | 62.2 | 59.3 | 54.5 | 50.9 | 43.2 |
| Gemma3-12B | 56.4 | 54.4 | 52.0 | 47.7 | 42.3 |
| Gemma3-27B | 60.4 | 59.3 | 57.2 | 55.0 | 51.2 |
| Idefics3-8B | 34.0 | 29.4 | 27.8 | 24.7 | 21.5 |
| Phi-4-Multimodal | 44.8 | 37.5 | 30.8 | 26.6 | 20.9 |
| NVILA-Lite-8B | 46.5 | 42.1 | 35.5 | 26.9 | 18.4 |
| Pixtral-12B | 56.5 | 54.6 | 52.4 | 43.9 | 41.1 |

Figure 1: Performance on MMLONGBENCH. We report results for selected frontier models, and the full results of all models are provided in Figure 23. Note that Claude-3.7-Sonnet supports at most 100 images, and we mark the results as N/A for cases with more images (More in Section D.4)

applications. These models include closed-source models GPT-4o [10], Claude-3.7 [34], and Gemini 2 and 2.5 [43, 85], as well as open-source model families, such as Qwen2.5-VL [29], InternVL3 [30], and Gemma3 [3]. We also consider position extrapolation methods, such as YaRN [37] and V2PE [52] (See Section D.6). The full list of evaluated models is provided in Table 10. Following existing works [20], we use greedy decoding for all models for consistency and randomly sample 100 examples from each dataset. More details on the experimental setup are in Section C.

## 4.1 Evaluation on MMLONGBENCH across Tasks and Context Lengths

We present the performance of selected frontier LCVLMs in Figure 1, and the full results of all 46 models are reported in Figure 23. We analyze model performance from multiple perspectives and summarize our main findings as follows:

**All models struggle, but closed-source models perform better.** Here, we consider the performance at the longest input length of 128K tokens. In general, we observe that all models struggle on our vision-language long-context tasks. For example, even GPT-4o only achieves 62.9 on average, while open-source models perform even worse. We find that Gemini-2.5-Pro stands out as the strongest

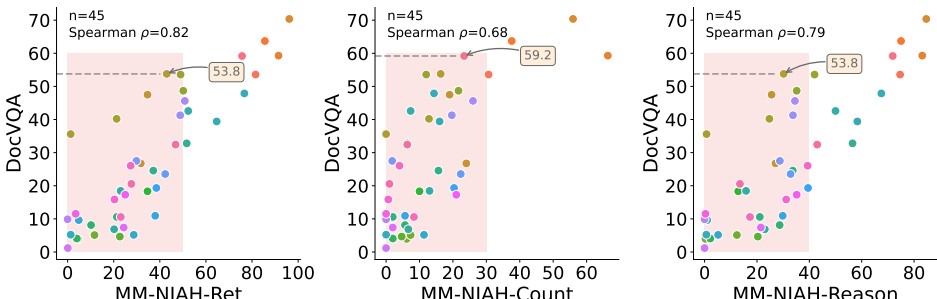

Figure 2: Distribution of long-document VQA (DocVQA) with respect to performance on MM-NIAH variants. We find that the models are concentrated in the coral-shaded areas.

LCVLM. Other than ICL, Gemini-2.5-Pro outperforms open-source models by about 20 absolute points. On ICL, although the gap is relatively smaller, due to the strong performance of Qwen2-VL-72B, there is still a difference of about 14 points. While the other closed-source models continue to surpass open-source models, the margin is often under 10 points. Further, Ovis2-34B achieves a score of 41.6 on summarization, similar to GPT-4o (42.4). Qwen2.5-VL-32B achieves a SubEM score of 64.6 on VRAG, even better than Gemini-2.0-Flash. These findings show that while current closed-source models generally perform better, open-source ones are also competitive.

**Models can generalize to longer context lengths.** Another interesting observation is that some models can generalize to longer context lengths than they are officially designed for. For example, although the context window for Qwen2-VL-72B during training is only 32K tokens, the model can generalize to a 128K input length and still achieve an average score of 51.9. We also observe similar effects on other models, such as Ovis2-34B and InternVL2.5-26B. This phenomenon is likely because the underlying LLMs of those LCVLMs have been trained with longer context windows [29]. We leave further investigation to future work.

**Reasoning can improve multimodal long-context ability.** We include Gemini-2.0-Flash-T in our evaluation, which is the thinking variant of Gemini-2.0-Flash. From the results, we observe that the reasoning ability can consistently improve the Gemini-2.0-Flash on all tasks. While the changes for VRAG, Recall, and ICL are modest, summarization and DocVQA exhibit marked improvements of 25.3% and 10.1%, respectively. Then, Gemini-2.5 models exhibit even stronger performance, which are natively designed as thinking models. See more results with newly added models in Section D.5.

**Different models exhibit different strengths.** Generally, we find that model performance varies considerably across different tasks. For instance, Qwen2.5-VL-32B outperforms InternVL3-38B on VRAG, but underperforms on NIAH. Similarly, Ovis2-34B excels at summarization but struggles on DocVQA. These findings further support the necessity of a comprehensive benchmark covering diverse downstream tasks. In Sections D.6 and D.7, we also provide additional analysis about the performance of position extrapolations and the lost-in-the-middle phenomenon.

## 4.2 Can Needle-in-a-Haystack Tasks Reflect LCVLM's Overall Long-Context Ability?

The needle-in-a-haystack (NIAH) has been primarily used to evaluate LCVLMs' long-context abilities. However, it remains unclear whether strong performance on NIAH reliably reflects overall long-context capability on diverse tasks. In this section, we first analyze the difficulty of existing NIAH benchmarks and find that current NIAH tasks are challenging, resulting in limited differentiation between models. Further, we compute Spearman's rank correlation ($\rho$) between NIAH performance and that on other tasks. Our results show that none of these NIAH tasks consistently correlates with performance across diverse, practical scenarios.

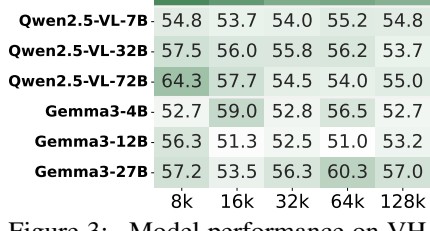

Figure 3: Model performance on VH-Multi dataset. Random guess yields 50% accuracy, highlighting its difficulty.

**Text-image interleaved NIAH tasks are challenging.** In Figure 3, we find that even state-of-the-art models like GPT-4o and Gemini-2.5 struggle to surpass

80% accuracy on VH-Multi when the context length is just 8K tokens (approximately 22 images). Most models are just slightly better than a random guess (50%). This demonstrates that locating objects in a large set of images is still hugely challenging for current LCVLMs. See more discussion in Section D.1. Then, we plot the performance of different models on the retrieval, counting, and reasoning tasks of MM-NIAH against their performance on long-document VQA in Figure 2. We find that most models achieve low performance on the counting and reasoning tasks, with scores below 30 and 40, respectively. The difficulty of the tasks and low performance result in poor separability between models. While the retrieval is an easier task, it still does not align well with DocVQA tasks. In short, we find that *both VH and MM-NIAH present significant challenges to current LCVLMs*, thus showing limited differentiation between models and weak alignment with other tasks.

**NIAH tasks fail to reflect overall long-context abilities.**
As shown in Figure 4, none of the NIAH tasks exhibit strong correlation with the broader set of long-context tasks. This suggests that performance on NIAH tasks may not be a reliable indicator of general long-context capabilities. In particular, Visual Haystack (VH) tasks show especially low correlations due to their high difficulty, as discussed above, which results in limited ability to distinguish between models. In MM-NIAH, counting and reasoning tasks show weak correlations with several downstream tasks, with coefficients below 0.8. The retrieval task also shows weak alignment with ICL performance. Interestingly, simpler tasks — like retrieval with a single needle in unrelated essays — tend to correlate better with diverse task categories, which is consistent with our findings in Figure 2. We further examine the differences between text-based and image-based needles in Section D.2.

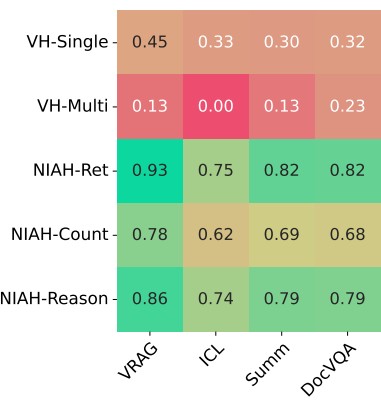

Figure 4: Spearman's $\rho$ across all 46 models at 128K tokens.

### 4.3 Weak Correlation Across Categories Calls for Diverse Evaluation

We perform a cross-category correlation analysis of model performance. We find that different categories do not consistently show strong correlation ($< 0.85$) with each other, as shown in Figure 5. Specifically, VRAG and NIAH closely correlate because retrieval is the central capability of both tasks. A further investigation shows that VRAG achieves its highest correlation (of 0.93) with the retrieval task (MM-NIAH-Ret) in Figure 15, reinforcing the shared emphasis on retrieval. Meanwhile, summarization and long-document VQA show a high correlation of 0.88, likely due to their shared input format — image-based PDF documents. This suggests that image types affect category correlations. In contrast, ICL tasks show relatively weak correlations with other categories. The ICL tasks evaluate models' ability to induce new classification rules from numerous exem-

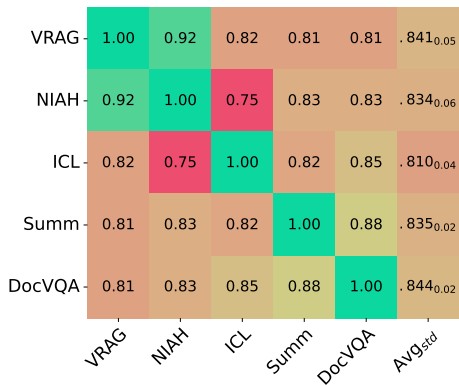

Figure 5: Spearman's $\rho$ between all categories with $L = 128$K. For each category, the *Avg* excludes the correlation with itself.

plars, a skill orthogonal to recalling facts in long contexts. This further demonstrates that model developers should consider various long-context skills to draw a more holistic picture of LCVLMs. See Section D.3 for detailed dataset-level correlations and additional category-wise insights.

**Long-document VQA is a reliable proxy for long-context capabilities.** As shown in Figure 5, long-document VQA achieves the highest average correlation with other categories, indicating that it is more aligned with the broader range of long-context tasks. For example, questions from LongDocURL [17] cover not only simple retrieval but also complex understanding and reasoning. Meanwhile, long-document VQA exhibits the smallest standard deviation, showing that it is also stable and balanced. Taken together, these findings suggest long-document VQA is a more representative and reliable proxy than the commonly adopted NIAH for reflecting overall system performance, allowing model developers to iterate more rapidly without the overhead of full-scale evaluation.

| | MMLB-Doc (All) | | | | | Text-Pure Cases | | | | | Vision-Needed Cases | | | | |
|---|---|---|---|---|---|---|---|---|---|---|---|---|---|---|---|
| | 8k | 16k | 32k | 64k | 128k | 8k | 16k | 32k | 64k | 128k | 8k | 16k | 32k | 64k | 128k |
| Qwen2.5-VL-7B | 52.7 | 50.0 | 42.8 | 35.8 | 17.1 | 78.9 | 75.2 | 67.0 | 50.2 | 11.3 | 36.6 | 36.4 | 29.2 | 28.7 | 19.5 |
| ◇ w/ OCR | 49.2 | 36.9 | 34.8 | 25.3 | 21.1 | 76.1 | 69.0 | 61.4 | 49.5 | 47.6 | 32.7 | 19.6 | 19.9 | 13.3 | 10.3 |
| ◇ w/ LLM | 45.4 | 46.5 | 36.8 | 24.6 | 26.9 | 65.6 | 80.8 | 56.8 | 54.5 | 56.9 | 33.1 | 27.9 | 25.5 | 9.9 | 14.7 |
| Qwen2.5-VL-32B | 58.0 | 58.2 | 48.5 | 42.1 | 31.9 | 85.4 | 77.6 | 60.6 | 51.7 | 26.2 | 41.2 | 47.8 | 41.6 | 37.3 | 34.3 |
| ◇ w/ OCR | 47.4 | 39.6 | 45.0 | 39.7 | 32.4 | 78.7 | 79.9 | 74.5 | 83.8 | 64.3 | 28.2 | 17.9 | 28.3 | 18.0 | 19.4 |
| ◇ w/ LLM | 48.2 | 40.6 | 44.0 | 36.8 | 33.7 | 84.0 | 78.0 | 78.0 | 84.8 | 78.9 | 26.3 | 20.6 | 24.9 | 13.2 | 15.2 |
| Gemma3-27B | 41.4 | 34.1 | 31.4 | 32.3 | 30.0 | 59.5 | 51.8 | 46.0 | 49.1 | 45.7 | 30.3 | 24.6 | 23.1 | 24.1 | 23.6 |
| ◇ w/ OCR | 48.3 | 37.9 | 41.8 | 29.1 | 28.6 | 77.5 | 63.7 | 61.5 | 65.6 | 57.8 | 30.4 | 24.1 | 30.7 | 11.1 | 16.7 |

Figure 6: Error Analysis on MMLongBench-Doc. Instead of PDF-formatted documents, we feed OCR-extracted plain text to LCVLMs (◇ w/ OCR) and also test corresponding LLMs: Qwen2.5-7B and Qwen2.5-32B (◇ w/ LLM). We also show scores on examples with different answer sources.

## 4.4 Error Analysis

Our evaluation shows that current LCVLMs have significant room for improvement. To better understand their limitations, we analyze model predictions in detail.

In Figure 6, we show the performance of using another pipeline for DocVQA on MMLongBench-Doc. Here, we convert PDF-formatted documents to plain text with OCR (◇ w/ OCR) and feed them to LCVLMs. There is no clear winner between the PDF-formatted and OCR-extracted pipelines across all models. While Qwen2.5-VL models perform better with PDF-formatted documents in most cases, Gemma3-27B prefers plain text for shorter input lengths (≤ 32K). Furthermore, we perform a fine-grained analysis by categorizing examples according to their answer sources into two groups: *text-pure* and *vision-needed*. As expected, using PDF documents leads to higher scores in vision-needed cases, whereas plain text yields better

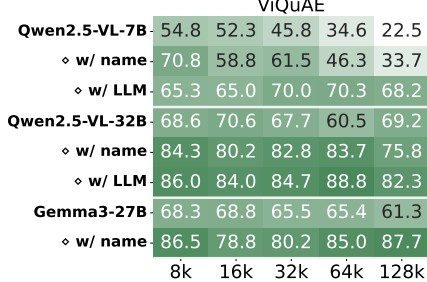

Figure 7: Error analysis on ViQuAE. We replace the image with its original entity name (◇ w/ name) and also test text-only counterparts: Qwen2.5-7B and Qwen2.5-32B (◇ w/ LLM).

performance in text-pure cases, especially with longer inputs (64K and 128K). This suggests that *OCR capability remains a bottleneck for current LCVLMs* when handling long-context inputs. Future work could explore combining both pipelines to further enhance performance. Meanwhile, when using OCR-extracted text, replacing LCVLMs with the corresponding LLMs, Qwen2.5-7B and Qwen2.5-32B (◇ w/ LLM), yields better results in text-pure cases of DocVQA.

We also examine the sources of errors in the VRAG category in Figure 7. Since ViQuAE is built on TriviaQA [73], we replace all images in ViQuAE questions with their corresponding entity names and feed those text-only questions into LCVLMs. All models show varying degrees of improvement, with Gemma3-27B achieving the largest gain of 26.4 points (at 128K), suggesting that *a bottleneck of LCVLMs lies in cross-modality information retrieval.* Besides, providing entity names as input to corresponding LLMs improves model performance. These results illustrate a common trade-off between multimodal and text-only long-context abilities during the training of LCVLMs.

## 5 Conclusion

In this work, we have introduced MMLONGBENCH, the first comprehensive benchmark for evaluating long-context vision-language models (LCVLMs) across a wide spectrum of downstream tasks. By covering five distinct task categories—while unifying cross-modal token counting and standardizing context lengths, MMLONGBENCH provides a rigorous, extensible foundation for diagnosing the strengths and weaknesses of frontier LCVLMs. Our evaluation of 46 models reveals that i) evaluation on a single task does not reliably predict overall long-context capability, ii) even frontier models face significant challenges, particularly in OCR accuracy and cross-modal retrieval, and iii) models endowed with enhanced reasoning mechanisms (e.g., "thinking" variants) consistently outperform their base counterparts in long-context settings. Looking forward, we hope MMLONGBENCH will serve as a standard yardstick for the community to benchmark new LCVLMs and to drive research on more efficient vision-language token encodings, more robust position-extrapolation schemes, and improved multi-modal retrieval and reasoning capabilities.

## Acknowledgments

The authors of this paper were supported by the ITSP Platform Research Project (ITS/189/23FP) from ITC of Hong Kong, SAR, China, and the AoE (AoE/E-601/24-N), the RIF (R6021-20) and the GRF (16205322) from RGC of Hong Kong, SAR, China. We also thank the support from NVIDIA AI Technology Center (NVAITC) and the valuable suggestions provided by Yuxiang Wu, Shizhe Diao, and Hongming Zhang on the design of this benchmark.

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

# A  Dataset Details

In this appendix, we provide more details on how to build long-context examples based on existing datasets.

## A.1  Visual Retrieval-Augmented Generation

**Gold Passage.**  InfoSeek [22] is a large-scale dataset for factual knowledge-based VQA featuring long-tail entities from Wikipedia [86]. For InfoSeek, we use the lead section of the Wikipedia page for the named entity in the question image as gold passages, which is the first section on each page and serves as a summary of the whole page. The lead section may be long, so we chunk it into multiple 100-word passages. We remove all the queries whose corresponding lead section does not contain the correct answer.

ViQuAE [23] replaces the named entities in questions from TriviaQA [73] with corresponding entity images from Wikimedia Commons[2]. We obtain gold passages for each question from the KILT benchmark [72], which provides human annotations of gold passages for queries in TriviaQA.

**Length Control.**  We populate the context with hard negative passages from Wikipedia, and the version we used is the Wikipedia 2019-08-01 dump [72]. We follow the KILT benchmark to preprocess Wikipedia articles into 100-word passages. For retrieval, we adopt a retrieval-and-rerank pipeline, where BM25 is first used for coarse retrieval, followed by reranking with dense embeddings from `Alibaba-NLP/gte-large-en-v1.5` [87]. Here, we replace the image in each question with its original entity name for better retrieval accuracy, because text-based retrieval achieves higher recall and provides harder distractors.

Previous work [20] shows that this pipeline presents a significantly greater challenge than randomly sampled passages. Also, using a real embedding model for retrieval is consistent with various downstream visual retrieval-augmented generation tasks and can better reflect downstream application performance.

## A.2  Needle-in-a-Haystack

**Length Control.**  For the Visual Haystack dataset [16], we directly use the needles and target objects from the original dataset. Then, we change the number of negative distractor images to build long-context examples with a given input length $L$. Note that the original dataset simply reports the image number as context length, ignoring different image sizes. Here, we count image tokens based on patches split by vision encoders as discussed in Section 3.2. There are two tasks in the dataset: VH-Single and VH-Multi, where the target objects are contained in a single needle image or multiple needle images, respectively.

For MM-NIAH [18], the contexts in the original dataset are composed of entire web pages from OBELICS [75]. However, using full web pages makes it difficult to control input length *at a fine granularity* since web pages typically contain tens of thousands of tokens. To solve the issue, we chunk the text content of web pages into 100-word passages, as we did for the Wikipedia corpus in VRAG. Meanwhile, the images in MM-NIAH contain only a few hundred tokens. Thus, we can achieve fine-grained control over the context length and incrementally add text passages and images to reach a given length $L$.

**Metrics.**  We report the accuracy on Visual Haystack, exactly the same as in the original work. The MM-NIAH dataset contains *three* different tasks: needle retrieval, counting, and reasoning. We use MM-NIAH-Ret, MM-NIAH-Count, and MM-NIAH-Reason as their abbreviations, respectively. In each task, there are both text-needle and image-needle examples. In MM-NIAH-Ret and MM-NIAH-Reason, we use substring exact match (SubEM) for text-needle examples and accuracy for image-needle examples, exactly following the original paper [18]. In MM-NIAH-Count, we find that the soft accuracy metric proposed in the original paper [18] can be exploited: simply predicting a list of zeros ([0, 0, ...]) results in a score over 30 on image-needle counting. Thus, we report the accuracy of the total count of the needle in the haystack instead of comparing the list of needle counts, which

---

[2]https://commons.wikimedia.org/

| | GovReport | | | | | | | | | | Multi-LexSum | | | | | | | | | |
| --- | --- | --- | --- | --- | --- | --- | --- | --- | --- | --- | --- | --- | --- | --- | --- | --- | --- | --- | --- | --- |
| | ROUGE-L | | | | | GPT-4o Eval | | | | | ROUGE-L | | | | | GPT-4o Eval | | | | |
| GPT-4o | 32.5 | 32.4 | 33.4 | 37.2 | 38.1 | 14.9 | 19.7 | 24.1 | 36.4 | 37.6 | 22.7 | 23.9 | 24.3 | 24.1 | 25.3 | 35.2 | 42.5 | 44.5 | 45.5 | 47.2 |
| InternVL2.5-4B | 20.1 | 18.3 | 17.0 | 15.5 | 13.9 | 6.3 | 10.5 | 12.3 | 13.2 | 11.2 | 23.9 | 25.3 | 25.1 | 25.2 | 23.7 | 25.8 | 34.7 | 35.4 | 35.0 | 25.4 |
| InternVL2.5-8B | 21.5 | 22.4 | 21.8 | 20.6 | 21.4 | 9.6 | 13.0 | 13.5 | 16.2 | 19.4 | 24.7 | 25.1 | 25.2 | 25.2 | 25.2 | 31.2 | 33.4 | 35.7 | 34.8 | 35.8 |
| InternVL3-2B | 31.6 | 32.9 | 33.8 | 36.3 | 35.5 | 4.5 | 12.4 | 7.7 | 9.6 | 16.3 | 22.5 | 23.1 | 23.6 | 23.2 | 22.9 | 26.6 | 29.1 | 25.6 | 26.6 | 28.5 |
| InternVL3-14B | 32.1 | 33.2 | 34.1 | 36.6 | 37.7 | 10.3 | 11.6 | 14.1 | 20.8 | 31.7 | 25.1 | 26.2 | 26.2 | 24.5 | 23.7 | 34.4 | 39.6 | 40.4 | 39.7 | 39.9 |
| Gemma3-4B | 28.6 | 29.9 | 29.8 | 30.7 | 31.3 | 3.7 | 8.6 | 10.8 | 16.9 | 8.9 | 22.0 | 21.8 | 21.5 | 21.6 | 21.3 | 26.8 | 32.2 | 31.6 | 31.0 | 32.4 |
| Gemma3-12B | 30.6 | 32.2 | 33.3 | 34.3 | 35.5 | 7.8 | 9.3 | 8.3 | 10.1 | 14.5 | 22.2 | 21.8 | 22.6 | 22.7 | 23.2 | 34.2 | 38.7 | 42.0 | 42.1 | 41.5 |
| Gemma3-27B | 30.6 | 31.9 | 33.2 | 34.8 | 36.5 | 10.1 | 16.0 | 21.2 | 28.8 | 36.3 | 22.5 | 22.6 | 23.2 | 22.5 | 23.0 | 35.7 | 40.9 | 42.8 | 42.2 | 45.1 |
| | 8k | 16k | 32k | 64k | 128k | 8k | 16k | 32k | 64k | 128k | 8k | 16k | 32k | 64k | 128k | 8k | 16k | 32k | 64k | 128k |

Figure 8: Comparison between ROUGE-L and the GPT-4o evaluation on summarization datasets. GPT-4o evaluation reflects the performance gain on Gemma3-27B with increased input length, and it also clearly sets apart open-source models with different sizes. In comparison, ROUGE-L remains almost the same for all models and input lengths.

we find is more reliable. Last but not least, we sample text-needle and image-needle examples evenly in all three tasks.

Please refer to the original MM-NIAH paper [18] for comprehensive details of all three tasks and two needle modalities.

## A.3 Many-Shot In-Context Learning

**Class Sampling.** We include Stanford Cars [24], Food101 [25], SUN397 [78], and iNat 2021 [27]. Since the 128K context length can accommodate only about 500 images, 50 different classes are randomly sampled from each dataset. With 50 classes, we can ensure that there are about 10 exemplars from each class, which is sufficient. For iNat 2021, since the dataset contains substantially more classes (over 10,000 species), we randomly sample 50 classes from the "Birds" supercategory and 50 classes from the "Plants" supercategory. For every single example, all the exemplars and the test image are either from the "Birds" classes or the "Plants" classes, ensuring the task remains a 50-way classification problem. Meanwhile, for shorter input lengths, we need to reduce the class number to ensure sufficient shots per class. Specifically, we randomly sample 5, 10, 20, and 40 classes for the input length of 8K, 16K, 32K, and 64K tokens. With those class numbers, we find that the number of exemplars per class is similar to that when there are 128K tokens.

**Label mapping and length control.** We employ a label mapping strategy to ensure that models perform classification based on in-context exemplars instead of relying on their pre-trained knowledge. Each label is randomly mapped to an integer $i \in \{0, 1, \ldots, N - 1\}$, where $N$ is the number of classes, following established practices [20, 88]. Throughout the evaluation, we provide models with images and their corresponding integer labels. Following Li et al. [76], we arrange exemplars into demonstration rounds, each of which includes exactly one exemplar per label in a random order. We concatenate these demonstration rounds, with the last round truncated if needed, to build examples of input length $L$. Thus, the label distribution is balanced in all datasets and input lengths.

## A.4 Summarization

**Preprocessing.** GovReport [15] consists of reports written by the U.S. Government Accountability Office (GAO)[3] and the Congressional Research Service (CRS)[4]. GAO reports constitute the majority of the dataset (more than 12K) and provide enough coverage for evaluation. Since CRS reports have a different format from GAO reports and there are only a few CRS reports available, we only use GAO reports in our benchmark. Summaries of GAO reports are written by experts and are structured into three aspects: "Why GAO did this study," "What GAO found," and "What GAO recommends." Those summaries are written at the beginning pages of the PDF-formatted GAO documents. We use PyMuPDF [5] to detect those answers and remove the corresponding pages to ensure no answer leakage in the inputs.

---

[3] www.gao.gov

[4] crsreports.congress.gov

[5] https://pymupdf.readthedocs.io

Multi-LexSum [28] consists of multi-document summarization problems about civil rights lawsuits, and the summaries are written by domain experts (i.e., lawyers and law students).

Both datasets are initially constructed using the OCR-extracted plain text as the input. In our evaluation, we replace the OCR-extracted plain text with the original PDF-formatted documents. We screenshot each page of all PDF-formatted documents with 144 DPI, following common practices [5]. Different from previous works [5, 17], we *do not concatenate* all screenshot pages into one or a few images to reduce the token numbers but instead directly feed them into LCVLMs since we are stress-testing the model's long-context capability.

**Length Control.** To control the input length $L$, we truncate document pages from the end. When there are multiple documents in Multi-LexSum, we truncate each document evenly from the end. Additionally, we discard examples that exceed the 128K context length by more than 24K tokens, as adding them would require truncating too many pages to fit within the context window of 128K. In this way, we can avoid confounding effects on model performance caused by the loss of key information due to severe page truncation.

**Data Scale.** In long-form generation tasks, each summary typically contains many atomic claims to be verified, in contrast to short outputs of other categories, such as VRAG. There are 15,951 claims in 387 examples in these two datasets, indicating a large scale for evaluation.

**Model-Based Metric.** The N-gram overlap metrics, such as ROUGE-L [89], have long been condemned for their poor correlation with human judgment for long-form generation [90, 91]. To ensure reliable evaluation for summarization, we adopt the reference-based LLM evaluation method proposed in HELMET [20]. Specifically, we first break down the gold reference summary into a set of atomic claims with GPT-4o, following prior work [92–94]. Next, we ask the model to check for three properties of model predictions: precision, recall, and fluency. We utilize GPT-4o to assess if each sentence in the generated summary is supported by the gold reference (precision) and if each atomic claim in the gold reference is present in the generated summary (recall). The F1 score is computed from the recall and precision. We also prompt GPT-4o to assess the fluency of the generated summary. The fluency is assigned a value of 0 if the output is incoherent, incomplete, or repetitive, and a value of 1 if it is fluent and coherent. The final score, Fluency-F1, is the product of fluency and F1 score.

Our empirical study in Figure 8 demonstrates that InternVL3-2B achieves ROUGE-L scores comparable to GPT-4o. Moreover, ROUGE-L exhibits minimal difference across different input lengths from 8K to 128K tokens. These observations reveal that ROUGE-L has low discriminative capacity and often fails to effectively distinguish between the quality of generated texts. In contrast, GPT4-o evaluation shows a significant gap across different input lengths and shows lower scores for models with shorter context windows, such as InternVL2.5.

**Atomic Claims Verification.** We manually checked 100 atomic claims from 25 Multi-LexSum summaries and another 100 atomic claims from 25 GovReport summaries. We found that only one claim was not factually accurate. Then, we checked the coverage of the claims and found no key facts were missing. This manual verification shows that GPT-4o is virtually always reliable for the decomposition task. For Multi-LexSum, we follow HELMET [20] and use the short summary to obtain atomic claims, where the dataset also provides long and tiny summaries for each case.

**GPT-4o Judgment Verification.** We show the detailed prompts for evaluating the fluency, precision, and recall in Tables 4 to 9, following previous works [20, 92, 95, 96]. We further conduct human analysis to verify the evaluation metric.

Quantitatively, we found that GPT-4o can consistently distinguish fluent and non-fluent outputs. The agreement between the model judgements and human judgements is 100% for randomly sampled outputs from GovReport and Multi-LexSum. Then, we sample 10 generated summaries for both GovReport and Multi-LexSum (20 in total) and check 5 atomic claims for each summary. The generated summaries are produced by Gemini-2.5-Pro and Qwen2.5-VL-32B. We follow a similar procedure to the GPT-4o evaluation and manually check the precision and recall of those sampled summaries. For precision, we observed Cohen's $\kappa = 0.90$ for GovReport and $\kappa = 0.89$ for Multi-LexSum, suggesting almost perfect agreement. Meanwhile, for recall, we observed Cohen's $\kappa = 0.90$ for GovReport and $\kappa = 0.93$ for Multi-LexSum, which are also near perfect.

**GovReport** | **Multi-LexSum**

| | ROUGE-L | | | | | Qwen2.5-7B Eval | | | | | ROUGE-L | | | | | Qwen2.5-7B Eval | | | | |
|---|---|---|---|---|---|---|---|---|---|---|---|---|---|---|---|---|---|---|---|---|
| | 8k | 16k | 32k | 64k | 128k | 8k | 16k | 32k | 64k | 128k | 8k | 16k | 32k | 64k | 128k | 8k | 16k | 32k | 64k | 128k |
| GPT-4o | 32.5 | 32.4 | 33.4 | 37.2 | 38.1 | 33.2 | 39.5 | 42.7 | 52.2 | 57.0 | 22.7 | 23.9 | 24.3 | 24.1 | 25.3 | 39.3 | 42.9 | 45.7 | 44.3 | 49.3 |
| InternVL3-2B | 31.6 | 32.9 | 33.8 | 36.3 | 35.5 | 24.9 | 40.1 | 32.9 | 44.1 | 50.2 | 22.5 | 23.1 | 23.6 | 23.2 | 22.9 | 27.6 | 29.4 | 31.0 | 29.7 | 29.6 |
| InternVL3-14B | 32.1 | 33.2 | 34.1 | 36.6 | 37.7 | 34.4 | 35.2 | 42.6 | 50.0 | 57.8 | 25.1 | 26.2 | 26.2 | 24.5 | 23.7 | 37.9 | 38.8 | 40.8 | 37.9 | 42.0 |
| Qwen2.5-VL-7B | 32.1 | 31.5 | 32.8 | 35.9 | 34.9 | 39.2 | 45.6 | 45.0 | 47.6 | 50.8 | 23.9 | 25.4 | 24.6 | 24.3 | 24.7 | 39.5 | 39.1 | 43.4 | 37.9 | 42.2 |
| Qwen2.5-VL-72B | 32.6 | 33.8 | 35.8 | 35.8 | 28.1 | 32.6 | 38.5 | 46.5 | 52.4 | 43.3 | 22.9 | 24.2 | 24.6 | 25.0 | 16.5 | 40.0 | 39.5 | 42.2 | 43.0 | 27.6 |
| Gemma3-4B | 28.6 | 29.9 | 29.8 | 30.7 | 31.3 | 27.8 | 36.4 | 44.5 | 52.9 | 52.5 | 22.0 | 21.8 | 21.5 | 21.6 | 21.3 | 30.8 | 34.4 | 32.1 | 34.6 | 34.2 |
| Gemma3-12B | 30.6 | 32.2 | 33.3 | 34.3 | 35.5 | 31.1 | 37.0 | 44.1 | 47.5 | 53.4 | 22.2 | 21.8 | 22.6 | 22.7 | 23.2 | 40.5 | 40.5 | 44.2 | 41.6 | 40.7 |
| Gemma3-27B | 30.6 | 31.9 | 33.2 | 34.8 | 36.5 | 34.5 | 42.5 | 47.5 | 57.4 | 55.5 | 22.5 | 22.6 | 23.2 | 22.5 | 23.0 | 37.5 | 37.8 | 42.8 | 45.0 | 43.2 |

Figure 9: Comparison between ROUGE-L and the Qwen2.5-7B-Instruct evaluation on summarization datasets.

**GovReport** | **Multi-LexSum**

| | ROUGE-L | | | | | Qwen2.5-72B Eval | | | | | ROUGE-L | | | | | Qwen2.5-72B Eval | | | | |
|---|---|---|---|---|---|---|---|---|---|---|---|---|---|---|---|---|---|---|---|---|
| | 8k | 16k | 32k | 64k | 128k | 8k | 16k | 32k | 64k | 128k | 8k | 16k | 32k | 64k | 128k | 8k | 16k | 32k | 64k | 128k |
| GPT-4o | 32.5 | 32.4 | 33.4 | 37.2 | 38.1 | 29.7 | 38.9 | 45.5 | 54.5 | 59.4 | 22.7 | 23.9 | 24.3 | 24.1 | 25.3 | 58.1 | 64.7 | 69.0 | 69.1 | 73.5 |
| InternVL3-2B | 31.6 | 32.9 | 33.8 | 36.3 | 35.5 | 23.5 | 29.7 | 33.4 | 46.5 | 46.7 | 22.5 | 23.1 | 23.6 | 23.2 | 22.9 | 40.7 | 45.7 | 48.9 | 44.1 | 49.6 |
| InternVL3-14B | 32.1 | 33.2 | 34.1 | 36.6 | 37.7 | 30.3 | 35.2 | 39.7 | 53.0 | 61.1 | 25.1 | 26.2 | 26.2 | 24.5 | 23.7 | 51.5 | 59.4 | 62.5 | 66.8 | 67.4 |
| Qwen2.5-VL-7B | 32.1 | 31.5 | 32.8 | 35.9 | 34.9 | 28.4 | 34.7 | 37.7 | 46.0 | 51.7 | 23.9 | 25.4 | 24.6 | 24.3 | 24.7 | 49.3 | 54.8 | 57.9 | 61.6 | 58.8 |
| Qwen2.5-VL-72B | 32.6 | 33.8 | 35.8 | 35.8 | 28.1 | 27.6 | 32.9 | 38.8 | 50.1 | 45.1 | 22.9 | 24.2 | 24.6 | 25.0 | 16.5 | 53.2 | 60.4 | 61.8 | 62.6 | 39.2 |
| Gemma3-4B | 28.6 | 29.9 | 29.8 | 30.7 | 31.3 | 27.2 | 37.1 | 43.9 | 49.8 | 51.2 | 22.0 | 21.8 | 21.5 | 21.6 | 21.3 | 43.5 | 51.8 | 54.5 | 56.6 | 58.1 |
| Gemma3-12B | 30.6 | 32.2 | 33.3 | 34.3 | 35.5 | 31.6 | 38.5 | 42.3 | 48.7 | 52.0 | 22.2 | 21.8 | 22.6 | 22.7 | 23.2 | 54.5 | 61.0 | 64.4 | 67.4 | 68.8 |
| Gemma3-27B | 30.6 | 31.9 | 33.2 | 34.8 | 36.5 | 32.2 | 40.1 | 46.5 | 59.1 | 63.5 | 22.5 | 22.6 | 23.2 | 22.5 | 23.0 | 54.1 | 62.8 | 64.1 | 68.0 | 70.3 |

Figure 10: Comparison between ROUGE-L and the Qwen2.5-72B-Instruct evaluation on summarization datasets.

Qualitatively, inspecting the disagreements, we find that most disagreements come from partially supported cases. We identified two common underlying reasons for partially supported cases when measuring precision and recall, respectively. First, a sentence in a generated summary may contain two points: "While agencies generally documented their review, inconsistencies and documentation gaps existed." We found the reference summary only supports "inconsistencies and documentation gaps existed," and the "agencies generally documented" part is an entailment inferred by the model. *Such inferred (entailed) information causes a lot of partially supported cases when measuring precision.* Second, the claims in the gold reference may include specific details, such as some geographic locations or organization names. These details may not be explicitly mentioned in the generated summary, *causing the partially supported cases for recall.*

**Open-Source Models as the Judge** We use GPT-4o in the model-based metric. To investigate the impact of using open-source LLMs instead, we introduce two new models as judges for the summarization task: Qwen2.5-7B-Instruct and Qwen2.5-72B-Instruct. We evaluated the summaries from 8 different LVLMs as shown in the Figures 9 and 10. We find that Qwen2.5-7B-Instruct cannot be used as the judge since it cannot distinguish different input lengths and models of different sizes. Better than the 7B model, Qwen2.5-72B-Instruct could be a low-cost replacement for GPT-4o; however, it still makes occasional mistakes. Thus, given the stronger ability of GPT-4o, we choose to use GPT-4o for the evaluation of summarization in our paper.

Here is a detailed discussion of Qwen2.5-7B-Instruct and Qwen2.5-72B-Instruct as the judge:

**(1) Using Qwen2.5-7B-Instruct:** First, we observe that Qwen2.5-7B-Instruct cannot distinguish different input lengths. For example, on Multi-LexSum, Gemma3-12B achieves scores that are consistently around 40, regardless of whether the input length is 8K or 128K tokens. Other models, like Qwen2.5-VL-7B and InternVL3-14B, also demonstrate the same problem on Multi-LexSum.

Second, another issue is that Qwen2.5-7B-Instruct cannot distinguish models of different sizes. For example, on the GovReport, InternVL3-14B obtains scores similar to GPT-4o when the input lengths are 32K, 64K, and 128K. Furthermore, on Multi-LexSum, Qwen2.5-VL-7B achieves scores similar to Qwen2.5-VL-72B when the input lengths are 8K, 16K, and 32K.

**(2) Using Qwen2.5-72B-Instruct:** We find that Qwen2.5-72B-Instruct can distinguish different input lengths and models of different sizes. However, there are still some occasional mistakes. For example, on GovReport, Gemma3-4B and Gemma3-12B obtain quite similar scores when the input length is 128K. Also, InternVL3-14B obtains a score higher than GPT-4o when the input length is 128K on the GovReport.

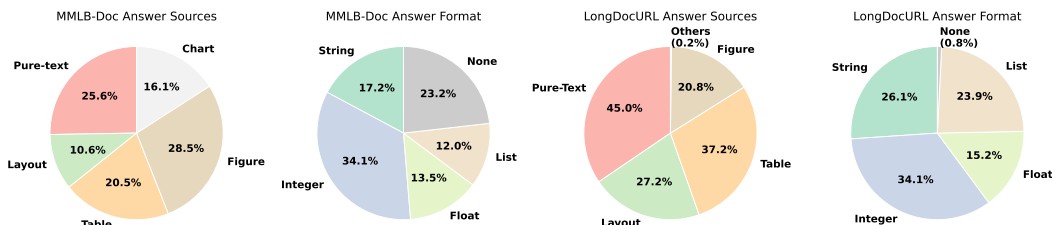

Figure 11: Data distribution of MMLongBench-Doc and LongDocURL after our pre-processing. Both datasets remain well-distributed, and their distributions are similar to the ones in the original paper.

Thus, we conclude that Qwen2.5-72B-Instruct can be a low-cost replacement for GPT-4o; however, it still makes occasional errors. For a more reliable evaluation, we choose to use GPT-4o in our paper.

### A.5    Long-Document VQA

**Preprocessing.**   MMLongBench-Doc [5] and LongDocURL [17] contain questions on various kinds of documents, such as financial reports, guidebooks, and academic papers, and the answer formats include string, integer, float, and list. More importantly, the rule-based evaluation method commonly adopted on those datasets depends on answer formats.

First, we find that there is a proportion of noisy answer format annotations. For example, a list answer like ['Top 10 File Categories Sorted By Disk Space', 'Last 12 Months Modified Disk Space History'] is annotated as being in string format. Conversely, answers in string format are also annotated in list format, such as ['PRIVACY SCREEN OPTIONS']. Therefore, our first step with these datasets is to correct the mislabeled answer formats and discard the instances for which the correct answer format cannot be recovered.

Second, both datasets rely heavily on LLMs, such as GPT-4o, to extract the answer from model predictions. This leads to *high evaluation costs* and poses *challenges for large-scale evaluation*, like 46 models in our work. Then, we take a closer examination of different formats of answers: (1) For integer and float answers, we find that numbers can be extracted with regular expressions; (2) For string answers, if the answer is short (less than 5 words), we find that model predictions are also short. Thus, we can directly use automatic metrics like ROUGE F1 without the need for answer extractions; (3) As a result, only *long-form string answers* require LLM-based extraction; Since long-form string answers ($> 5$ words) constitute only a small proportion of these datasets, we simply discard those instances to enable scalable evaluation without relying on GPT-4o for answer extraction. We find that the retained short string answers are mostly entity names; (4) Note that for list answers, we evaluate each element in the list (i.e., integer, float, or string elements) separately (then take average), and the evaluation rule is determined for each element by its type.

After all the filtering, we find both datasets remain *well-distributed* as shown in Figure 11.

**Evaluation Metrics.**   We follow previous works [5, 17] and employ the same rule-based scoring method that applies different strategies depending on the format of the reference answer: (1) For **String** format answers, we initially use regular expressions to determine whether the answers require exact matching (e.g., telephone numbers, email addresses, website addresses, filenames, times, dates, etc.). If the answer needs an exact match, we perform a substring exact match (SubEM) with a score of 0 or 1. Otherwise, we follow previous works [20, 63, 82, 97] and calculate ROUGE F1 scores; (2) For integer answers, we perform an exact match comparison, and the score is either 0 or 1; (3) For float answers, we treat the model prediction and gold reference as the same if the relative error is less than 1%; (4) For list answers, we evaluate each element separately based on its answer type and take the average. Here, we follow LongDocURL to use the Greedy List Match: for each element in the reference list, we compute its score against all elements in the prediction list and *greedily* select the highest score as its matching score. This metric does not require the predicted list to follow the same element order as the reference list, thereby providing greater tolerance in evaluation.

Different from them, SlideVQA [82] features questions based on 20-page slide decks, which contain rich layout information and less dense text. The answer formats in the dataset are string, integer, and float, and do not cover list answers. We use the same rule-based scoring method as described for MMLongBench-Doc and LongDocURL.

**Length Control.**   The input lengths of DocVQA tasks are also easy to control. If an example exceeds a given length $L$, we truncate the document evenly from both sides while preserving the answer pages. If the document cannot fill the length $L$, we alternately pad the left and right sides with randomly sampled negative documents until the required length is reached. Notably, we may also truncate a few pages of the last padding document as needed to control the length at the granularity of pages, instead of documents.

*The randomly sampled padding documents are not guaranteed to be truly irrelevant or negative.* They may occasionally contain information related to the question, which could potentially change the answer. To ensure models attend to the original document, we preface each question with the prompt "Based on the Document *<Original Doc ID>*, answer the following question."

### A.6   Image Resizing and Statistics

As we discussed in Section 3.3, the number of tokens per image is determined by the image size in our benchmark. In MM-NIAH, we find that many images from the OBELICS dataset are unnecessarily large (up to 8000×6000 pixels) and are not text-rich. Then, we resize those images' longer edge to 1024 pixels while preserving their aspect ratio.

We calculate the average number of images per example in all the datasets and input lengths in Table 3. From the table, we can find that our benchmark covers various text-to-image ratios. For example, VRAG tasks are text-centric and only contain one image per example, while ICL represents image-centric tasks with hundreds of images. MM-NIAH tasks are intermediate and feature both substantial text and multiple images.

### A.7   License

All the data collected are based on previously open-sourced datasets, and all licenses are publicly available.

## B   Full Model List

We list all 46 models [3, 10, 29, 30, 34, 43, 44, 47, 49–51, 83–85, 98, 99] we evaluated in Table 10. All 46 models have a pixel unshuffle operation to reduce the token counts of images. This is consistent with our token counting methods (Section 3.2). The only exception is Pixtral-12B, but we can resize its image (to 0.5× on each side) to reduce the image tokens. Thus, we can fit Pixtral-12B on our GPU server and avoid extremely long input sequences.

Table 3: Average number of images per example in all datasets and input lengths. The values in subscript denote the standard deviations. (T) and (I) represents the text and image needle in each task of MM-NIAH.

| | Data Length | 8K | 16K | 32K | 64K | 128K |
|---|---|---|---|---|---|---|
| VRAG | InfoSeek | $1.0_{0.0}$ | $1.0_{0.0}$ | $1.0_{0.0}$ | $1.0_{0.0}$ | $1.0_{0.0}$ |
| | ViQuAE | $1.0_{0.0}$ | $1.0_{0.0}$ | $1.0_{0.0}$ | $1.0_{0.0}$ | $1.0_{0.0}$ |
| NIAH | VH-Single | $21.7_{0.9}$ | $44.7_{1.3}$ | $90.9_{1.9}$ | $183.4_{2.9}$ | $368.2_{4.3}$ |
| | VH-Multi | $21.7_{0.9}$ | $44.8_{1.3}$ | $91.0_{1.9}$ | $183.4_{2.8}$ | $368.2_{4.3}$ |
| | MM-NIAH-Ret (T) | $3.8_{1.4}$ | $7.6_{2.0}$ | $15.4_{2.7}$ | $30.4_{3.8}$ | $59.3_{5.5}$ |
| | MM-NIAH-Count (T) | $3.7_{1.4}$ | $7.6_{2.0}$ | $15.4_{2.7}$ | $30.3_{3.9}$ | $59.3_{5.7}$ |
| | MM-NIAH-Reason (T) | $3.8_{1.4}$ | $7.6_{2.0}$ | $15.5_{2.7}$ | $30.5_{3.9}$ | $59.3_{5.7}$ |
| | MM-NIAH-Ret (I) | $8.1_{1.0}$ | $11.8_{1.5}$ | $19.3_{2.2}$ | $34.2_{3.4}$ | $63.8_{5.5}$ |
| | MM-NIAH-Count (I) | $5.7_{1.2}$ | $9.3_{1.6}$ | $16.9_{2.2}$ | $31.7_{3.2}$ | $61.5_{5.6}$ |
| | MM-NIAH-Reason (I) | $6.5_{1.0}$ | $10.2_{1.4}$ | $17.6_{2.1}$ | $32.5_{3.4}$ | $62.3_{5.7}$ |
| ICL | Stanford Cars | $36.1_{1.3}$ | $72.6_{2.3}$ | $156.3_{0.9}$ | $324.0_{5.3}$ | $628.8_{9.0}$ |
| | Food101 | $25.0_{0.8}$ | $52.0_{0.8}$ | $106.1_{1.4}$ | $215.5_{2.3}$ | $432.5_{0.9}$ |
| | SUN397 | $37.0_{2.0}$ | $80.1_{3.7}$ | $161.3_{4.2}$ | $326.8_{2.2}$ | $656.6_{8.2}$ |
| | Inat2021 | $31.2_{0.6}$ | $66.1_{0.8}$ | $134.6_{1.4}$ | $271.0_{1.6}$ | $543.8_{1.8}$ |
| Summarization | GovReport | $2.0_{0.1}$ | $6.0_{0.0}$ | $12.0_{0.0}$ | $25.0_{0.0}$ | $50.7_{0.5}$ |
| | Multi-LexSum | $3.0_{0.1}$ | $6.0_{0.2}$ | $12.1_{0.5}$ | $25.2_{0.9}$ | $51.3_{1.8}$ |
| DocVQA | MMLongBench-Doc | $3.3_{1.3}$ | $6.9_{2.4}$ | $13.8_{4.8}$ | $28.0_{8.2}$ | $56.4_{12.1}$ |
| | LongDocURL | $3.6_{2.1}$ | $7.2_{4.5}$ | $14.2_{8.1}$ | $28.6_{14.4}$ | $55.3_{18.3}$ |
| | SlideVQA | $7.5_{0.9}$ | $16.2_{2.1}$ | $33.2_{2.7}$ | $67.2_{3.5}$ | $135.2_{5.2}$ |

Table 4: GovReport Fluency Evaluation Prompt

| Task: GovReport    Metric: Fluency |
| --- |

Please act as an impartial judge and evaluate the fluency of the provided text. The text should be coherent, non-repetitive, fluent, and grammatically correct.

**Below is your grading rubric:**
**- Score 0 (incoherent, repetitive, or incomplete)**: Incoherent sentences, repetitive sentences (even if not by exact words), incomplete answers, or gibberish. Note that even if the answer is coherent, if it is repetitive or incomplete, it should be given a score of 0.
**- Examples:**

- Incomplete: `"Summary:"`

- Incoherent: `"Summary: U.S. agencies engaged export and controls controls controls controls diversion prevent items U.S. activities compliance allies transshipment risk misuse exported misuse misuse illicit illicit against interests or."`

- Repetitive: `"Summary:The audit focused on determining the cost and schedule performance of selected programs. The audit focused on determining the cost and schedule performance of selected programs. The audit focused on determining the cost and schedule performance of selected programs. The audit focused on determining the cost and schedule performance of selected programs."`

**- Score 1 (coherent, non-repetitive answer)**: Coherent, non-repetitive, fluent, grammatically correct answers. If the text is coherent, non-repetitive, and fluent, but the last sentence is truncated, it should still be given a score of 1.
**Examples:**

- `"Why GAO Did This Study: Tobacco use is the leading cause of preventable death and disease in the United States. In 2009, the Family Smoking Prevention and Tobacco Control Act (Tobacco Control Act) granted FDA, an agency within the Department of Health and Human Services (HHS), authority to regulate tobacco products, including marketing and distribution to youth. The act established CTP, which implements the act by educating the public on the dangers of tobacco use; developing the science needed for tobacco regulation; and developing and enforcing regulations on the manufacture, marketing, and distribution of tobacco products. The act authorized FDA to assess and collect user fees from tobacco manufacturers and importers. The Tobacco Control Act mandated that GAO review the authority and resources provided to FDA for regulating the manufacture, marketing, and distribution of tobacco products. This report examines (1) how FDA spent tobacco user fees for key activities using its authorities granted in the act, and (2) any challenges FDA encountered in using its authorities. GAO analyzed data on tobacco user fees collected and spent on key activities by FDA as of March 31, 2014; reviewed documents related to FDA's key activities, as well as relevant laws, regulations, and guidance; and interviewed CTP, public health, and tobacco industry officials... [about 150 more words]"`

Now, read the provided text, and evaluate the fluency using the rubric. Then output your score in the following json format: `{"fluency": 1}`.
**Text**: `"{text}"`

Table 5: Multi-LexSum Fluency Evaluation Prompt

---

**Task: Multi-LexSum    Metric: Fluency**

---

Please act as an impartial judge and evaluate the fluency of the provided text. The text should be coherent, non-repetitive, fluent, and grammatically correct.

**Below is your grading rubric:**
**Score 0 (incoherent, repetitive, or incomplete)**: Incoherent sentences, repetitive sentences (even if not by exact words), incomplete answers, or gibberish. Note that even if the answer is coherent, if it is repetitive or incomplete, it should be given a score of 0.
 **- Examples:**

- Incomplete: `"Summary:"`

- Incoherent:    `"Summary:  The plaintiff the the the the able the the the the the the the the the the able the the the the the Ã?\n"`

- Repetitive:    `"Summary:  The U.S. government brought a criminal case against four defendants. Summary:  The U.S. government brought a criminal case against four defendants.  Summary: The U.S. government brought a criminal case against four defendants.  Summary:  The U.S. government brought a criminal case against four defendants."`

**Score 1 (coherent, non-repetitive answer)**: Coherent, non-repetitive, fluent, grammatically correct answers. If the text is coherent, non-repetitive, and fluent, but the last sentence is truncated, it should still be given a score of 1.
**- Examples:**

- `"This case is about an apprenticeship test that had a disparate impact on Black apprenticeship applicants.  The Equal Employment Opportunity Commission (EEOC) filed this lawsuit on December 27, 2004, in U.S. District Court for the Southern District of Ohio."`

- `"The plaintiffs sought declaratory and injunctive relief, as well as attorneys' fees and costs, under the Americans with Disabilities Act, the Rehabilitation Act of 1973, the Social Security Act, and the Nursing Home Reform Act.  The case was certified as a class action on behalf of all Medicaid-eligible adults with disabilities in Cook County, Illinois, who are being, or may in the future be, unnecessarily confined to nursing facilities and with appropriate supports and services may be able to live in a community setting.  The defendants denied the allegations and argued that the plaintiffs' claims were not typical of the class and that the class definition was too broad.  The case is ongoing, with discovery and expert testimony scheduled for the fall of"`

Now, read the provided text, and evaluate the fluency using the rubric.  Then output your score in the following json format: {"fluency": 1}.

**Text**: "{text}"

---

Table 6: GovReport Precision Evaluation Prompt

| | |
|---|---|
| **Task: GovReport** | **Metric: Precision** |

Please act as an impartial judge and evaluate the quality of the provided summary of a government report from U.S. Government Accountability Office (GAO). The summary should discuss one or more of the following: why GAO did this study, what GAO found, and what GAO recommends.

**Below is your grading rubric:**
**Precision:**

- Evaluate the provided summary by deciding if each sentence in the provided summary is supported by the information provided in the expert summary. A sentence is still supported even if some minor details (e.g., dates, entity names, or locations) are not explicitly mentioned in the expert summary. A sentence is not supported if its major facts are not mentioned, contradicted, or introduce new information not present in the expert summary (e.g., extra analysis or commentary).

- **Score:** the number of sentences in the provided summary that are supported by the expert summary.

- **Examples:** use the following examples to guide your evaluation.

**Example 1:**

**Expert summary:** <start of summary>Why GAO Did This Study: The Congressional Budget Office projects that federal deficits will reach $1 trillion in 2020 and average $1.2 trillion per year through 2029, further adding to the more than $16 trillion in current debt held by the public. As a result, Treasury will need to issue a substantial amount of debt to finance government operations and refinance maturing debt. To support its goal to borrow at the lowest cost over time, Treasury must maintain strong demand from a diverse group of investors for Treasury securities. GAO prepared this report as part of continuing efforts to assist Congress in identifying and addressing debt management challenges. This report (1) identifies factors that affect demand for Treasury securities and (2) examines how Treasury monitors and analyzes information about the Treasury market to inform its debt issuance strategy. GAO analyzed data on investor holdings of Treasury securities; surveyed a non-generalizable sample of 109 large domestic institutional investors across 10 sectors (67 responded); reviewed Treasury analysis and market research; and interviewed market participants across sectors, experts on foreign investors, and Treasury officials... [about 300 more words] <end of summary>

**Provided summary:** <start of summary>The U.S. Government Accountability Office (GAO) conducted a performance audit from June 2018 to December 2019 to assess the management of federal debt by the Department of the Treasury. The audit aimed to evaluate how Treasury manages its debt to finance the federal deficit and refinances maturing debt while minimizing costs. Treasury issues various types of securities, including Treasury bills, notes, bonds, and inflation-protected securities, with maturities ranging from a few weeks to 30 years, to attract a diverse investor base and maintain a healthy secondary market. The audit found that Treasury's regular and predictable framework for issuing securities supports reliable demand, but changes in market conditions and policies pose risks to the liquidity, depth, and safety of Treasury securities. Treasury uses market outreach, auction and market metrics, and analytical models to inform its debt issuance decisions but lacks policies for bilateral market outreach and quality assurance for analytical models. The report recommends Treasury finalize its market outreach policy and establish a quality assurance policy for analytical models to ensure transparency and appropriate documentation. Treasury agreed with the recommendations and plans to implement them.<end of summary>

**Reasoning:** Sentence 1 is not supported (audit dates and "performance audit" not mentioned). Sentence 2 is supported (aligns with Treasury's goal of borrowing at lowest cost). Sentence 3 is not supported (specific security types and maturity ranges not listed). Sentence 4 is supported (risks to liquidity, depth, safety are mentioned). Sentence 5 is supported (mentions the three inputs and missing policies). Sentence 6 is supported (matches the recommendations). Sentence 7 is supported (Treasury agreed). Therefore, the precision score is 5.

**Output:** {"precision": 5, "sentence_count": 7}

**Example 2:** ...

Now, read the provided summary and expert summary, and evaluate the summary using the rubric. First, think step-by-step and provide your reasoning and assessment on the answer. Please keep your response concise and limited to a single paragraph. Then output your score in the following json format: {"precision": 7, "sentence_count": 20}.
Expert summary: <start of summary>{expert_summary}<end of summary>
Provided summary: <start of summary>{summary}<end of summary>

Table 7: Multi-LexSum Precision Evaluation Prompt

---

**Task: Multi-LexSum    Metric: Precision**

---

Please act as an impartial judge and evaluate the quality of the provided summary of a civil lawsuit. The summary is based on a set of legal documents, and it should contain a short description of the background, the parties involved, and the outcomes of the case.

**Below is your grading rubric:**
**Precision:**

- Evaluate the provided summary by deciding if each sentence in the provided summary is supported by the information provided in the expert summary. A sentence is considered supported if its major facts align with the information in the expert summary. A sentence is still considered supported even if some of its minor details, such as dates, entity names, or the names of laws and previous court cases, are not explicitly mentioned in the expert summary. A sentence is not supported if its major facts are not mentioned or contradicted in the expert summary.

- Score: the number of sentences in the provided summary that are supported by the expert summary.

- Examples: use the following examples to guide your evaluation.

**Example 1:**

**Expert summary:** "This lawsuit, brought in the the U.S. District Court for the Central District of California, was filed on June 3, 2020. The plaintiffs were represented by attorneys from the ACLU of Southern California. This lawsuit followed nation-wide protests that occurred in response to the killing of George Floyd by a police officer in Minneapolis. While most protests were peaceful, some ended in violence, property destruction, rioting, and looting. Many cities, including Los Angeles and San Bernardino, issued curfews in an attempt to quell these riots. This action challenged these curfews as violations of free speech and assembly, free movement, due process, and challenged the San Bernardino curfew as a violation of the establishment clause (the San Bernardino curfew included a provision that exempted attendants of religious meetings from the curfew.)... [about 100 more words]."

**Provided summary:** "In June 2020, Black Lives Matter - Los Angeles and several individuals filed a lawsuit in the U.S. District Court for the Central District of California against Los Angeles Mayor Eric Garcetti, other city officials, and the City of San Bernardino, challenging the constitutionality of curfew orders imposed during protests against police violence. The plaintiffs, represented by the ACLU of Southern California, argued that the curfews violated their First Amendment rights to free speech and assembly, as well as their freedom of movement, by suppressing political protests and other activities. The lawsuit also claimed that the curfews were not narrowly tailored to address any emergency and lacked sufficient notice. However, the plaintiffs voluntarily dismissed the case in July 2020 after the defendants lifted the curfew orders and did not reinstate them in the following weeks."

**Reasoning:** The first sentence in the provided summary is well supported by the expert summary even though some entity names are not explicitly mentioned. The second sentence is also well supported by the expert summary, as it mentions the ACLU of Southern California and the First Amendment rights. The third sentence is not supported by the expert summary, as it does not mention the lack of narrow tailoring or sufficient notice. The fourth sentence is well supported by the expert summary, as it mentions the voluntary dismissal of the case in July 2020. Therefore, the precision score is 3.

**Output:** {"precision":  3, "sentence_count":  4}

**Example 2:** ...

Now, read the provided summary and expert summary, and evaluate the summary using the rubric. First, think step-by-step and provide your reasoning and assessment on the answer. Please keep your response concise and limited to a single paragraph. Then output your score in the following json format: {"precision":  2, "sentence_count":  6}.

Expert summary: "{expert_summary}"

Provided summary: "{summary}"

---

Table 8: GovReport Recall Evaluation Prompt

---

**Task: GovReport    Metric: Recall**

Please act as an impartial judge and evaluate the quality of the provided summary of a government report from U.S. Government Accountability Office (GAO). The summary should discuss one or more of the following: why GAO did this study, what GAO found, and what GAO recommends. The text should contain all the major points in the expert-written summary, which are given to you.

**Below is your grading rubric:**
**Recall:**

- Evaluate the provided summary by deciding if each of the key points is present in the provided summary. A key point is considered present if its factual information is mostly-supported by the provided summary. If a key point contains multiple facts, it is considered supported if most of the facts are present.

- Score: the number of key points mostly-supported by the provided summary.

- Examples: Use the following example to guide your evaluation.

**Example 1:**

Key points:
1. The Future Combat System (FCS) program is the centerpiece of the Army's effort to transition to a lighter combat force.
2. The FCS program is the centerpiece of the Army's effort to transition to a more agile combat force.
3. The FCS program is the centerpiece of the Army's effort to transition to a more capable combat force.
4. By law, GAO is to report annually on the FCS program.
5. Law requires the Department of Defense (DOD) to hold a milestone review of the FCS program.
6. This milestone review is now planned for 2009.
7. This report addresses (1) what knowledge will likely be available in key areas for the review.
8. This report addresses (2) the challenges that lie ahead following the review.
9. To meet these objectives, GAO reviewed key documents and performed analysis.
10. GAO attended demonstrations and design reviews to meet these objectives.
11. GAO interviewed DOD officials to meet these objectives.
12. The Army will be challenged to demonstrate the knowledge needed to warrant an unqualified commitment to the FCS program.
13. This challenge will occur at the 2009 milestone review.
14. The Army has made progress.
15. Knowledge deficiencies remain in key areas. [31 more points]

Summary: <start of summary>Why GAO Did This Study: The Future Combat System (FCS) program is the centerpiece of the Army's effort to transition to a lighter combat force. By law, GAO is to report annually on the FCS program. This report addresses (1) what knowledge will likely be available in key areas for the review, and (2) the challenges that lie ahead following the review. To meet these objectives, GAO reviewed key documents and interviewed DOD officials.

What GAO Found: The Army will be challenged to demonstrate the knowledge needed to warrant an unqualified commitment to the FCS program. While the Army has made progress, knowledge deficiencies remain in key areas. Specifically, all critical technologies are not currently at a minimum acceptable level of maturity. Actual demonstrations of FCS hardware and software have been limited. Network performance is also largely unproven. DOD could have at least three programmatic directions to consider for shaping investments in future capabilities. [106 more words]<end of summary>

Reasoning: The summary covers: FCS as Army's transition centerpiece (point 1), GAO's reporting requirement (point 4), report objectives (points 7, 8), GAO's methods (points 9, 11), Army's challenges (point 12), progress and deficiencies (points 14, 15), technology issues (points 16, 19, 21), three programmatic directions (points 27, 29, 31, 33, 34, 36, 38, 41-43). It omits: "more agile/capable" (points 2, 3), 2009 milestone review (points 5, 6, 13), demonstrations attendance (point 10), design requirements issues (points 17, 18), small-scale concepts (point 20), program immaturity explanation (points 22, 23), funding competition (points 24-26), challenges after review (point 28), production before design demonstration (points 30, 32), technology testing issues (point 35), $50 billion funding (point 37), surrogate systems (points 39, 40), and increment justification (points 44-46). The summary supports 22 key points.

Output: {"supported_key_points": [1, 4, 7, 8, 9, 11, 12, 14, 15, 16, 19, 21, 27, 29, 31, 33, 34, 36, 38, 41, 42, 43], "recall": 22}

Now, read the provided summary and key points, and evaluate the summary using the rubric. First, think step-by-step and provide your reasoning and assessment on the answer. Please keep your response concise and limited to a single paragraph. Then output your score in the following json format: {"supported_key_points": [1, 4, 7, 8, 9, 11, 12, 14, 15, 16, 19, 21, 27, 29, 31, 33, 34, 36, 38, 41, 42, 43], "recall": 22}, where "supported_key_points" contains the key points that are present in the summary and "recall" is the total number of key points present in the summary.

Key points:
{keypoints}

Summary: <start of summary>{summary}<end of summary>

---

Table 9: Multi-LexSum Recall Evaluation Prompt

---

**Task: Multi-LexSum    Metric: Recall**

Please act as an impartial judge and evaluate the quality of the provided summary of a civil lawsuit. The summary is based on a set of legal documents, and it should contain a short description of the background, the parties involved, and the outcomes of the case. The text should contain all the major points in the expert-written summary, which are given to you.

**Below is your grading rubric:**
**Recall:**

- Evaluate the provided summary by deciding if each of the key points is present in the provided summary. A key point is considered present if its factual information is well-supported by the provided summary.

- Score: the number of key points present in the provided summary.

- Examples: use the following examples to guide your evaluation.

**Example 1:**

**Key points:**
1. The case challenged curfews in Los Angeles and San Bernardino, California.
2. The curfews were issued in response to the nationwide protests following the police killing of George Floyd in Minneapolis.
3. The complaint argued that the curfews violated free speech, free assembly, free movement, and Due Process.
4. The complaint also argued that the San Bernardino curfew violated the Establishment Clause.
5. The complaint sought injunctive and declaratory relief.
6. The plaintiffs voluntarily dismissed the case on July 7, 2020.
7. The dismissal occurred because the city had rescinded the curfews and not attempted to reinstate them.

**Summary:** In June 2020, Black Lives Matter - Los Angeles and several individuals filed a lawsuit in the U.S. District Court for the Central District of California against Los Angeles Mayor Eric Garcetti, other city officials, and the City of San Bernardino, challenging the constitutionality of curfew orders imposed during protests against police violence. The plaintiffs, represented by the ACLU of Southern California, argued that the curfews violated their First Amendment rights to free speech and assembly, as well as their freedom of movement, by suppressing political protests and other activities. The lawsuit also claimed that the curfews were not narrowly tailored to address any emergency and lacked sufficient notice. However, the plaintiffs voluntarily dismissed the case in July 2020 after the defendants lifted the curfew orders and did not reinstate them in the following weeks.

**Reasoning:** The summary states that the plaintiffs challenged the constitutionality of curfew orders against Los Angeles and San Bernadino, so key point 1 is present. The summary does not mention that the curfew orders were issued in response to the nationwide protest that resulted from the police killing of George Floyd in Minneapolis, so key point 2 is missing. The summary does mention that the complaint argued that the curfews violated the First Amendment rights to free speech and assembly, so key point 3 is present. The summary does not mention that the complaint argued that the San Bernardino curfew violated the Establishment Clause, so key point 4 is missing. The summary does not mention that the complaint sought injunctive and declaratory relief, so key point 5 is missing. The summary mentions that the plaintiffs voluntarily dismissed the case in July 2020 after the defendants lifted the curfew orders and did not reinstate them in the following weeks, so key point 6 and 7 are present. Finally, key points 1, 3, 6, and 7 are present in the summary, so the recall score is 4.

**Output:** {"recall": 4}

**Example 2:** ...

Now, read the provided summary and key points, and evaluate the summary using the rubric. First, think step-by-step and provide your reasoning and assessment on the answer. Please keep your response concise and limited to a single paragraph. Then output your score in the following json format: {"recall": 2}.

Key points: {keypoints}

Summary: "{summary}"

---

Table 10:  Length means the training length (default) or claimed context window (denoted by †). All LCVLMs are instruction-tuned. "Image Porc." stands for Image Processing, which is mainly Dynamic Resolution ViT [47] or Dynamic Tiling [48]. The positional embedding includes RoPE [100], M-RoPE [47], Linear Scaling [39, 101] LongRoPE [38], Dynamic-NTK, NTK-by-parts or YaRN [37].

| Name | Length | Image Proc. | Positional Emb. | # Params |
|---|---|---|---|---|
| *Proprietary* (No model details except the claimed context lengths. | | | | |
| gpt-4o-2024-11-20 | 128,000† | ? | ? | ? |
| claude-3-7-sonnet-20250219 | 200,000† | ? | ? | ? |
| gemini-2.0-flash-001 | 1,048,576† | ? | ? | ? |
| gemini-2.0-flash-thinking-exp-01-21 | 1,048,576† | ? | ? | ? |
| gemini-2.5-flash-preview-04-17 | 1,048,576† | ? | ? | ? |
| gemini-2.5-pro-preview-03-25 | 1,048,576† | ? | ? | ? |
| *Qwen2-VL & Qwen2.5-VL* | | | | |
| Qwen2-VL-2B-Instruct | 32,768 | Dynamic-Resolution ViT | M-RoPE | 2B |
| Qwen2-VL-7B-Instruct | 32,768 | Dynamic-Resolution ViT | M-RoPE | 7B |
| Qwen2-VL-72B-Instruct-AWQ | 32,768 | Dynamic-Resolution ViT | M-RoPE | 72B |
| Qwen2.5-VL-3B-Instruct | 32,768 | Dynamic-Resolution ViT | M-RoPE | 3B |
| Qwen2.5-VL-7B-Instruct | 32,768 | Dynamic-Resolution ViT | M-RoPE | 7B |
| Qwen2.5-VL-32B-Instruct | 32,768 | Dynamic-Resolution ViT | M-RoPE | 32B |
| Qwen2.5-VL-72B-Instruct-AWQ | 32,768 | Dynamic-Resolution ViT | M-RoPE | 72B |
| *InternVL2, InternVL2.5, & InternVL3* | | | | |
| InternVL2-1B | 8,192 | Dynamic Tiling | RoPE | 0.9B |
| InternVL2-2B | 8,192 | Dynamic Tiling | Dynamic-NTK | 2.21B |
| InternVL2-4B | 8,192 | Dynamic Tiling | LongRoPE | 4.15B |
| InternVL2-8B | 8,192 | Dynamic Tiling | Dynamic-NTK | 8.08B |
| InternVL2_5-1B | 16,348 | Dynamic Tiling | RoPE | 0.9B |
| InternVL2_5-2B | 16,348 | Dynamic Tiling | Dynamic-NTK | 2.2B |
| InternVL2_5-4B | 16,348 | Dynamic Tiling | RoPE | 4.2B |
| InternVL2_5-8B | 16,348 | Dynamic Tiling | Dynamic-NTK | 8.1B |
| InternVL2_5-26B | 16,348 | Dynamic Tiling | Dynamic-NTK | 25.5B |
| InternVL3-1B | 32,768 | Dynamic Tiling | Dynamic-NTK | 0.9B |
| InternVL3-2B | 32,768 | Dynamic Tiling | Dynamic-NTK | 1.9B |
| InternVL3-8B | 32,768 | Dynamic Tiling | Dynamic-NTK | 8.1B |
| InternVL3-14B | 32,768 | Dynamic Tiling | Dynamic-NTK | 15.1B |
| InternVL3-38B | 32,768 | Dynamic Tiling | Dynamic-NTK | 38.4B |
| *Ovis2* | | | | |
| Ovis2-1B | 32,768 | Dynamic Tiling | RoPE | 1B |
| Ovis2-2B | 32,768 | Dynamic Tiling | RoPE | 2B |
| Ovis2-4B | 32,768 | Dynamic Tiling | RoPE | 4B |
| Ovis2-8B | 32,768 | Dynamic Tiling | RoPE | 8B |
| Ovis2-16B | 32,768 | Dynamic Tiling | RoPE | 16B |
| Ovis2-34B | 32,768 | Dynamic Tiling | RoPE | 34B |
| *Gemma-3* | | | | |
| gemma-3-4b-it | 131,072† | Dynamic Tiling | Linear Scaling | 4B |
| gemma-3-12b-it | 131,072† | Dynamic Tiling | Linear Scaling | 12B |
| gemma-3-27b-it | 131,072† | Dynamic Tiling | Linear Scaling | 27B |
| *Idefics2* | | | | |
| idefics2-8b | 8,192 | Dynamic-Resolution ViT | RoPE | 8B |
| idefics2-8b-C (chatty) | 8,192 | Dynamic-Resolution ViT | RoPE | 8B |
| Mantis-8B-Idefics2 | 8,192 | Dynamic-Resolution ViT | RoPE | 8B |
| *Idefics3* | | | | |
| Idefics3-8B-Llama3 | 10,240 | Dynamic Tiling | NTK-by-parts | 8B |
| *Phi-based* | | | | |
| Phi-3-vision-128k-instruct | 131,072 | Dynamic Tiling | LongRoPE | 4.2B |
| Phi-3.5-vision-instruct | 131,072 | Dynamic Tiling | LongRoPE | 4.2B |
| Phi-4-multimodal-instruct | 131,072 | Dynamic Tiling | LongRoPE | 5.6B |
| *NVILA* | | | | |
| NVILA-Lite-2B-hf-preview | 32,768 | Dynamic Tiling | RoPE | 2B |
| NVILA-Lite-8B-hf-preview | 32,768 | Dynamic Tiling | RoPE | 8B |
| *Pixtral* | | | | |
| pixtral-12b | 131,072 | Dynamic-Resolution ViT | RoPE | 12B |

Table 11: The number of tokens produced by models *without pixel unshuffle* for the inputs (64K and 128K tokens) of the ICL and Visual Haystack (VH) datasets. The numbers are in thousands (K). These models cannot process long sequences of images efficiently.

| Model | ICL | | VH | |
|---|---|---|---|---|
| | 64K | 128K | 64K | 128K |
| Llama-3.2-11B | 1,821K | 3,626K | 1,174K | 2,358K |
| Llava-onevision-7b | 558K | 1,142K | 496K | 996K |
| mPLUG-Owl3-7B | 1,398K | 2,786K | 930K | 1,870K |

## B.1 LVLMs Beyond our Evaluation

**Token Efficiency.** Beyond the models in our evaluation, there are also a lot of excellent models, such as Llava-onevision [102], Llama3.2 [1], mPLUG-Owl3 [45]. However, we find that those models don't have pixel unshuffle operations. While Pixtral-12B's ViT can take images with dynamic resolution, the ViTs of these three models use dynamic tiling, such as $560 \times 560$ tiles for Llama3.2-11B. Thus, unlike Pixtral-12B, we cannot reduce the number of image tokens by simply resizing the input image, since these models do not accept images smaller than the predefined tile size. As shown in Table 11, these models cannot efficiently process high-resolution images as a whole, generating a number of tokens that is at least ten times the pre-defined input length (64K or 128K). *This causes extremely long input sequences, and we cannot fit them on our GPU server.*

**Challenge for Model Integration.** DeepSeek-VL [103], DeepSeek-VL2 [48], and Long-Llava [53] are also excellent, but they present additional challenges. These models do not provide a standard API in the HuggingFace Transformers framework [104], which we used for inference. Therefore, these models cannot be loaded directly via Transformers, and we need to develop them based on their GitHub repositories. As a result, integrating these models requires a *prohibitive* amount of engineering effort to adapt their codebases, making it impractical within our current scope. We leave their integration for future work.

## C Experimental Setup

As previously described, we evaluate all 46 models across different input lengths $L \in \{8192, 16384, 32768, 65536, 131072\}$. We evaluate the proprietary models using their API. The specific versions we used are as follows:

- GPT-4o: `gpt-4o-2024-11-20`
- Claude-3.7-Sonnet: `claude-3-7-sonnet-20250219`
- Gemini-2.0-Flash: `gemini-2.0-flash-001`
- Gemini-2.0-Flash-T: `gemini-2.0-flash-thinking-exp-01-21`
- Gemini-2.5-Flash: `gemini-2.5-flash-preview-04-17`
- Gemini-2.5-Pro: `gemini-2.5-pro-preview-03-25`

For all open-source models, we evaluate them on an $8 \times$A100 (80GB) GPU server. We use the HuggingFace Transformers framework [104] to deploy models and generate outputs. Since all models are instruction-tuned, we apply the chat templates to all datasets. We load models in BF16 with FlashAttention2 [105] for faster inference. The largest open-source models tested in our work have 72B parameters. Our computational resources are limited to $8 \times$A100 GPUs; thereby, we cannot evaluate models with over 100B parameters, such as Llama4 [4], at 128K tokens.

We sampled 100 examples from each dataset to evaluate models. Note that in MM-NIAH, we sample the text-needle and image-needle examples evenly, with 50 of each type. This amount actually results in 600 examples for single-needle tasks: ViQuAE, VH-Single, and MM-NIAH-Ret, and 300 examples for multi-needle tasks: InfoSeek, VH-Multi, and MM-NIAH-Count. This is because we test 6 different depths (i.e., [0, 0.2, 0.4, 0.6, 0.8, 1.0]) for single-needle examples and 3 different permutations for multi-needle examples to mitigate the positional bias. The MM-NIAH-Reason is

|              | VH-Single |      |      |      |      |   | VH-Multi |      |      |      |      |
|--------------|------|------|------|------|------|---|------|------|------|------|------|
|              | 8k | 16k | 32k | 64k | 128k | | 8k | 16k | 32k | 64k | 128k |
| GPT-4o        | 67.1 | 64.3 | 61.7 | 65.7 | 64.0 | | 70.3 | 65.0 | 63.3 | 61.0 | 50.7 |
| Gemini-2.5-Pro | 85.4 | 81.8 | 78.0 | 64.9 | 62.7 | | 76.5 | 72.9 | 68.9 | 70.1 | 67.7 |
| Qwen2.5-VL-7B  | 60.8 | 57.5 | 52.5 | 52.5 | 52.5 | | 54.8 | 53.7 | 54.0 | 55.2 | 54.8 |
| Qwen2.5-VL-32B | 64.7 | 58.5 | 55.8 | 54.7 | 51.3 | | 57.5 | 56.0 | 55.8 | 56.2 | 53.7 |
| Qwen2.5-VL-72B | 66.0 | 61.0 | 57.8 | 54.3 | 53.5 | | 64.3 | 57.7 | 54.5 | 54.0 | 55.0 |
| Gemma3-4B      | 58.3 | 54.0 | 53.3 | 54.2 | 51.7 | | 52.7 | 59.0 | 52.8 | 56.5 | 52.7 |
| Gemma3-12B     | 59.2 | 55.5 | 51.0 | 51.8 | 53.2 | | 56.3 | 51.3 | 52.5 | 51.0 | 53.2 |
| Gemma3-27B     | 65.3 | 63.8 | 60.0 | 56.5 | 54.7 | | 57.2 | 53.5 | 56.3 | 60.3 | 57.0 |

Figure 12: The performance of selected models on VH-Single and VH-Multi. The accuracy of a random guess is 50%. We find that these two tasks are very challenging.

more complex since image-needle (I) examples have a single needle, while text-needle (T) ones have multiple needles. There are 300 examples for MM-NIAH-Reason (I) (50 × 6 depths) and 150 examples for MM-NIAH-Reason (T) (50 × 3 permutations). Due to this modality imbalance in MM-NIAH-Reason, we compute the average score for the subset of each modality separately and report their mean as the final result. Together, we evaluate each model on 4,050 examples across five different input lengths, resulting in a total of 20,250 examples.

# D    Additional Results

We provide more evaluation results in addition to Section 4.

## D.1    The Difficulty of Visual Haystack

We discussed the difficulty of Visual Haystack in Section 4.2. As shown in Figure 3, current LCVLMs achieve the performance only slightly higher than random guessing on VH-Multi. Here, in Figure 12, we present the performance of the selected models on both VH-Single and VH-Multi, providing a complete view. We find that models also perform poorly on VH-Single.

**Task Correctness.**    We manually checked a number of examples from the Visual Haystack dataset and didn't find any errors in the task labels. As shown in Figure 12, Gemini-2.5-Pro achieves an accuracy of 85.4 on VH-Single, and GPT-4o achieves an accuracy of 70.3 on VH-Multi. These results are higher than a random guess (50%), which demonstrates the correctness of our implementation.

## D.2    Correlation between NIAH and Various Downstream Tasks

We discussed the correlation between NIAH tasks and various downstream applications in Section 4.2. Here, we provide a detailed version of the task correlation in Figure 13. For the three tasks in MM-NIAH, we also report the correlations on the subsets containing only text-needle or only image-needle examples. We can find that subsets of image-needle examples correlate less with various downstream tasks compared to text-needle examples, especially MM-NIAH-Count (I) and MM-NIAH-Reason (I)

We further show the performance of the selected models on the text-needle and image-needle subsets in Figure 14. We observe that models exhibit weak performance on MM-NIAH-Count (I) and MM-NIAH-Reason (I). This challenging nature leads to a low degree of separability between different models.

## D.3    Correlation between Datasets

We plot the correlation between all MMLONGBENCH datasets and category averages in Figure 15. Generally, the datasets in each category strongly correlate with each other. The VH-Single and VH-Multi are exceptions, due to their high difficulty. Also, MM-NIAH-Count exhibits relatively weak correlations with MM-NIAH-Ret and MM-NIAH-Reason, suggesting that counting is a different skill from retrieving needles (key information) and subsequently reasoning over them.

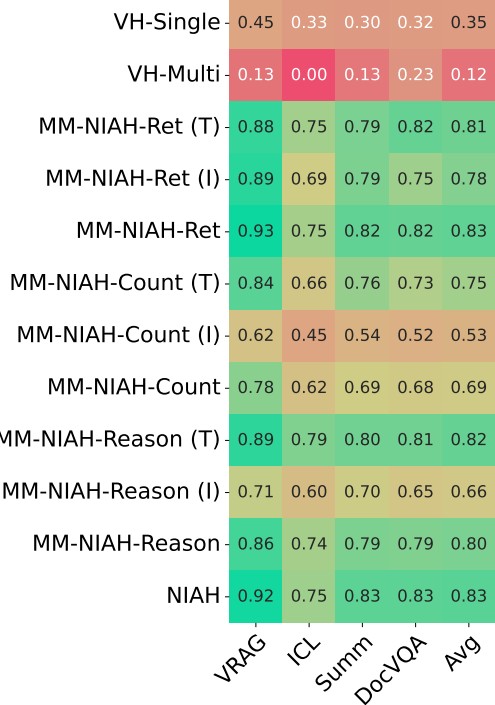

Figure 13: Spearman's correlation at 128K input length, calculated across 46 LCVLMs, between all NIAH and other downstream tasks.

**MM-NIAH-Ret (T)**

| | 8k | 16k | 32k | 64k | 128k |
|---|---|---|---|---|---|
| GPT-4o | 94.3 | 97.0 | 88.3 | 77.6 | 70.3 |
| Gemini-2.5-Pro | 97.3 | 100.0 | 98.3 | 98.7 | 96.7 |
| Qwen2.5-VL-32B | 95.3 | 97.3 | 95.0 | 88.3 | 48.0 |
| Qwen2.5-VL-72B | 94.3 | 96.3 | 95.3 | 81.3 | 48.7 |
| Ovis2-16B | 98.0 | 98.7 | 91.3 | 74.3 | 33.0 |
| Ovis2-34B | 98.3 | 99.0 | 98.3 | 88.3 | 40.7 |
| Gemma3-12B | 97.5 | 93.8 | 89.7 | 79.8 | 60.7 |
| Gemma3-27B | 95.7 | 97.0 | 90.7 | 77.7 | 49.3 |

**MM-NIAH-Ret (I)**

| | 8k | 16k | 32k | 64k | 128k |
|---|---|---|---|---|---|
| GPT-4o | 98.7 | 94.7 | 95.3 | 91.3 | 81.0 |
| Gemini-2.5-Pro | 100.0 | 99.0 | 98.7 | 97.3 | 95.7 |
| Qwen2.5-VL-32B | 79.0 | 78.7 | 69.0 | 63.7 | 50.0 |
| Qwen2.5-VL-72B | 88.3 | 78.7 | 75.7 | 66.3 | 51.7 |
| Ovis2-16B | 91.3 | 84.3 | 79.7 | 62.3 | 44.0 |
| Ovis2-34B | 94.7 | 89.3 | 81.0 | 68.0 | 44.0 |
| Gemma3-12B | 85.3 | 75.0 | 64.3 | 51.3 | 37.0 |
| Gemma3-27B | 81.7 | 74.7 | 62.0 | 60.3 | 52.3 |

**MM-NIAH-Count (T)**

| | 8k | 16k | 32k | 64k | 128k |
|---|---|---|---|---|---|
| GPT-4o | 94.7 | 94.7 | 78.0 | 64.6 | 42.0 |
| Gemini-2.5-Pro | 99.3 | 98.7 | 95.3 | 87.2 | 80.5 |
| Qwen2.5-VL-32B | 51.3 | 53.3 | 43.3 | 35.3 | 9.3 |
| Qwen2.5-VL-72B | 88.0 | 86.0 | 79.3 | 68.0 | 16.0 |
| Ovis2-16B | 42.7 | 46.7 | 27.3 | 22.0 | 9.3 |
| Ovis2-34B | 48.0 | 52.7 | 46.0 | 33.3 | 13.3 |
| Gemma3-12B | 44.7 | 33.0 | 18.0 | 18.3 | 9.3 |
| Gemma3-27B | 64.0 | 57.3 | 43.3 | 28.7 | 12.7 |

**MM-NIAH-Count (I)**

| | 8k | 16k | 32k | 64k | 128k |
|---|---|---|---|---|---|
| GPT-4o | 67.3 | 36.0 | 10.0 | 29.3 | 4.7 |
| Gemini-2.5-Pro | 47.3 | 41.3 | 40.0 | 34.7 | 31.3 |
| Qwen2.5-VL-32B | 11.3 | 23.3 | 33.3 | 20.7 | 14.7 |
| Qwen2.5-VL-72B | 23.3 | 23.3 | 37.3 | 38.0 | 27.3 |
| Ovis2-16B | 52.0 | 53.3 | 40.7 | 39.3 | 31.3 |
| Ovis2-34B | 36.0 | 14.7 | 26.7 | 35.3 | 31.3 |
| Gemma3-12B | 35.0 | 39.7 | 36.7 | 35.7 | 30.0 |
| Gemma3-27B | 30.7 | 30.0 | 24.7 | 34.7 | 39.3 |

**MM-NIAH-Reason (T)**

| | 8k | 16k | 32k | 64k | 128k |
|---|---|---|---|---|---|
| GPT-4o | 81.3 | 81.3 | 80.0 | 66.7 | 58.7 |
| Gemini-2.5-Pro | 96.7 | 95.3 | 92.5 | 93.3 | 90.0 |
| Qwen2.5-VL-32B | 77.3 | 72.7 | 65.3 | 56.7 | 37.3 |
| Qwen2.5-VL-72B | 84.7 | 77.3 | 81.0 | 58.0 | 39.3 |
| Ovis2-16B | 75.3 | 63.3 | 48.0 | 36.0 | 18.7 |
| Ovis2-34B | 78.0 | 75.3 | 68.7 | 46.7 | 26.7 |
| Gemma3-12B | 62.7 | 51.7 | 47.0 | 35.0 | 20.7 |
| Gemma3-27B | 81.3 | 66.7 | 57.3 | 32.7 | 24.0 |

**MM-NIAH-Reason (I)**

| | 8k | 16k | 32k | 64k | 128k |
|---|---|---|---|---|---|
| GPT-4o | 85.3 | 75.7 | 73.7 | 71.0 | 85.0 |
| Gemini-2.5-Pro | 82.7 | 83.0 | 79.7 | 78.3 | 79.3 |
| Qwen2.5-VL-32B | 60.0 | 56.7 | 55.3 | 51.0 | 46.7 |
| Qwen2.5-VL-72B | 43.7 | 36.3 | 35.7 | 30.0 | 31.0 |
| Ovis2-16B | 79.3 | 60.0 | 59.7 | 49.7 | 60.3 |
| Ovis2-34B | 61.3 | 48.7 | 39.3 | 42.7 | 39.0 |
| Gemma3-12B | 51.2 | 52.3 | 51.3 | 49.5 | 46.8 |
| Gemma3-27B | 65.0 | 52.0 | 51.7 | 51.7 | 45.0 |

Figure 14: Results of selected models on the subsets of MM-NIAH containing only text-needle or image-needle examples. (T) and (I) represent text-needle and image-needle examples, respectively.

## D.4 Performance of Claude

At the time of our evaluation, the Claude 3 family of models can take up to 100 images[6] in a single request. However, a few datasets, such as Food101, VH-Single, or SlildeVQA, contain hundreds of images at input lengths of 64K and 128K tokens. As a result, it is impossible to process all images in a single pass through the model. For each input length, if one or more datasets within a category contain samples with more than 100 images, we exclude that category from evaluation at that input length. We provide the statistics about the average image number per example in each dataset at all five input lengths in Table 3 and Section A.6. When testing `Claude-3.7-Sonnet`, we mark those

---

[6] https://docs.anthropic.com/en/docs/build-with-claude/vision#basics-and-limits

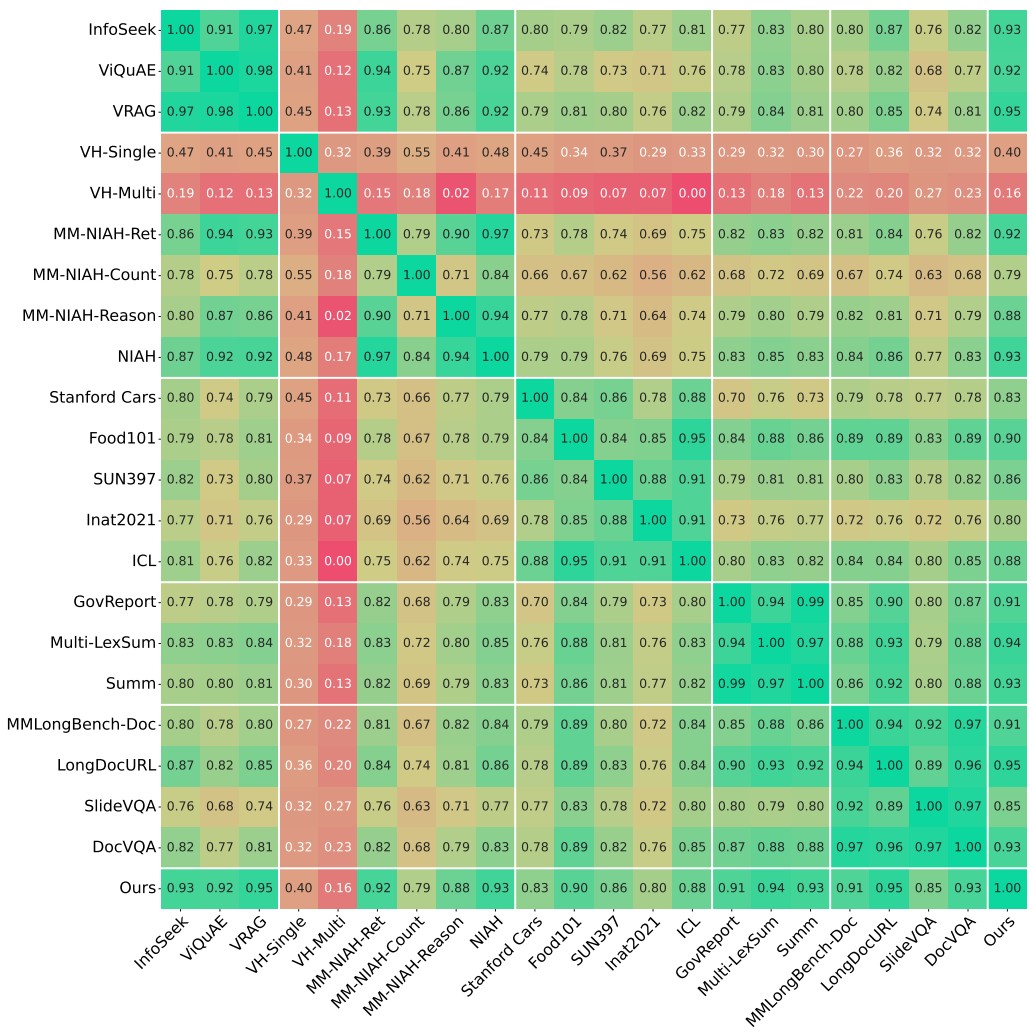

Figure 15: Spearman's correlation at 128K input length, calculated across 46 LCVLMs, between all MMLONGBENCH datasets and category averages.

untestable cases as "N/A" in our results to distinguish them from genuine model failures (which receive a score of 0).

### D.5 More Models for Reasoning Comparison

In Section 4.1, we observe that reasoning ability enhances multimodal long-context performance, as demonstrated by the Gemini model series. Here, we additionally evaluate MiMo-VL-7B-SFT (Reasoning) to further support this finding. As shown in Table 12, we observe that the average performance of MiMo-VL-7B-SFT is much higher than Qwen2.5-VL-7B across all context lengths. At 8K context length, MiMo-VL-7B-SFT achieves 66.0 on average, substantially surpassing Qwen2.5-VL-7B's 57.4. Meanwhile, MiMo-VL-7B-SFT consistently achieves higher performance on each task, except for summarization (this is quite common; see Section 4.1, where different models exhibit varying strengths). This advantage also persists as the context length increases.

### D.6 Positional Embedding Extrapolation Experiments

In this section, we evaluate two positional embedding extrapolation methods, namely YaRN [37] and V2PE [52]. Experimental results indicate that the current positional extrapolation methods still pose significant challenges for effectively extending the context window of LCVLMs.

Table 12: Performance comparison of MiMo-VL-7B-SFT (Reasoning) and Qwen2.5-VL-7B (Non-Reasoning) with different context lengths.

| Context | Model | VRAG | NIAH | ICL | Summ | DocVQA | Avg. |
|---------|-------|------|------|-----|------|--------|------|
| 8K | MiMo-VL-7B-SFT | 67.4 | 75.5 | 98.0 | 20.0 | 69.1 | 66.0 |
|  | Qwen2.5-VL-7B | 50.1 | 57.3 | 95.6 | 23.5 | 60.7 | 57.4 |
| 16K | MiMo-VL-7B-SFT | 63.6 | 71.5 | 93.2 | 21.2 | 68.1 | 63.5 |
|  | Qwen2.5-VL-7B | 48.7 | 53.0 | 91.5 | 29.1 | 57.1 | 55.9 |
| 32K | MiMo-VL-7B-SFT | 63.0 | 65.6 | 90.5 | 21.2 | 67.2 | 61.5 |
|  | Qwen2.5-VL-7B | 43.2 | 47.7 | 78.5 | 30.8 | 57.2 | 51.5 |

Figure 16: Results of applying YaRN [37] to Qwen2.5-VL models. We find that YaRN only improves the performance of Qwen2.5-VL-3B substantially. However, the SoTA performance from larger models (i.e., 32B and 72B) only fluctuates slightly.

**Adding YaRN to Qwen2.5-VL.** According to its technical reports [29], Qwen2.5-VL models are pre-trained with a context length of 32K tokens during the "Long-Context Pre-Training" stage. Meanwhile, its HuggingFace model card shows that we can use YaRN [37], with a scaling factor of 4, to extend its context length to 128K tokens[7]. Note that YaRN is used in a zero-shot manner, as there is no continual training for YaRN on Qwen2.5-VL. In Figure 16, we test the performance of using YaRN. We have two observations: (1) Using YaRN may hurt the performance on shorter input lengths. For example, at 8K tokens, the DocVQA score of Qwen2.5-VL-32B decreases from 67.8% to 63.4%; (2) On average across the entire MMLONGBENCH, YaRN only substantially improves the performance of Qwen2.5-VL-3B (from 21.2 to 30.2 at 128K). However, the SoTA performance from large models (i.e., 32B and 72B) only fluctuates slightly.

To ensure a fair comparison, we do not apply YaRN to Qwen2.5-VL in our main evaluations. Since YaRN is used in a zero-shot way here, applying it would lead to an unfair comparison over other models.

**Adding V2PE to InternVL2.** Ge et al. [52] proposed a positional embedding extrapolation method called V2PE, where it assigns smaller positional increments to visual tokens than textual tokens. They further applied V2PE to InternVL2-2B and trained the model to enhance its performance on MM-NIAH-Ret (I). In this experiment, we evaluate the V2PE-256K checkpoints[8] with the visual increments $\delta \in \{\frac{1}{16}, \frac{1}{64}, \frac{1}{256}\}$. As shown in Figure 17, we find that (1) V2PE is very sensitive to different visual increments. For example, when we use $\frac{1}{16}$ and $\frac{1}{64}$, the performance is even worse than InternVL2-2B, which leads to extra hyperparameter tuning; (2) The V2PE (256) shows extremely high performance on NIAH tasks, which can be attributed to the fact that it was trained on MM-NIAH.

---

[7]https://huggingface.co/Qwen/Qwen2.5-VL-3B-Instruct#processing-long-texts
[8]https://huggingface.co/OpenGVLab/V2PE/tree/main/V2PE-256K

Figure 17: Results of applying V2PE [52] to InternVL2-2B. We find V2PE is very sensitive to the visual increment $\delta$ and overfitted to NIAH tasks. Note that we use ROUGE-L for summarization here, since it is already sufficient to distinguish between models. Thus, there is no need to use the costly GPT-4o evaluation. The numbers in parentheses (i.e., 16, 64, and 256) correspond to the visual increment $\delta \in \left\{\frac{1}{16}, \frac{1}{64}, \frac{1}{256}\right\}$, respectively.

The sharp performance difference on NIAH tasks versus other categories suggests that V2PE is strongly overfitted to the MM-NIAH dataset.

### D.7 Lost in the Middle

Existing works found that text-pure LLMs often struggle to recall needles in the middle of the input sequence, named lost in the middle [57]. On our benchmark, we extend the previous analysis to vision-language tasks with input length up to 128K tokens. We place the needle at six different evenly spaced depths in the context (i.e., $[0, 0.2, 0.4, 0.6, 0.8, 1.0]$) and evaluate the LCVLMs' ability to retrieve it. In our study, the needle may be a gold passage, an image, or a key sentence. We show the results in Figure 18 for ViQuAE, Figure 19 for VH-Single, Figure 20 for MM-NIAH-Ret (T), Figure 21 for MM-NIAH-Ret (I), Figure 22 for MM-NIAH-Reason (I).

We observe a similar lost-in-the-middle phenomenon in many LCVLMs on long-context vision-language tasks. For example, the InternVL3-14B in Figure 20 and Ovis2-34B in Figure 21 both exhibit much better performance when the needle is at depths 0 and 1.0. Furthermore, as we extend the context to longer lengths (e.g., 128K tokens), we observe cases where the model tends to favor either the very beginning or the very end of the context, but not both simultaneously. For example, as shown in Figure 18, InternVL3-8B prefers the very beginning of the context (depth 0) at 128K tokens, whereas Qwen2.5-VL-72B favors the very end (depth 1.0).

### D.8 Error Analysis Details

We conducted two error analyses in Section 4.4. We provide more details of those two analyses in this section. First, for MMLongBench-Doc, we used PyMuPDF [9] to extract the plain text from PDF-formatted documents. For ViQuAE, the entity names are already provided in the dataset, since it is constructed based on TriviaQA. The text-pure LLMs we used, corresponding to Qwen2.5-VL models, are Qwen2.5-7B-Instruct and Qwen2.5-32B-Instruct, which are instruction-tuned versions.

### D.9 Full Model Evaluation Results

In Figure 23, we provide the results of all 46 models. We also plot the performance of all 46 models on each dataset in Figures 24 to 27.

### D.10 Idefics2 Performance

The Idefics2-8B and Idefics-8B-C only have a training context window of 2K tokens [49]. We find this leads to very poor long-context generalization. Also, the LLM used in Idefics2 is Mistral-7B-v0.1 [32], whose training length is only 8K tokens. From Figure 20, we observe that Idefics2 models perform well only when the needle depth is $1.0$ and the context is short (8K or 16K tokens).

---

[9] https://pymupdf.readthedocs.io

Additionally, we conduct a sanity check by removing all negative images and retaining only the needle images in Visual Haystack (i.e., one image for single-needle examples and two or three images for multi-needle examples). As shown in Table 13, we observe that both models achieve performance much higher than a random guess (50%), indicating the correctness of the implementation.

Table 13: Sanity check of Idefics2-8B and Idefics2-8B-C. Here, we use the Visual Haystack dataset. We remove all negative images and only retain needle images (i.e., one image for single-needle examples and two or three images for multi-needle examples).

| Model | VH-Single | VH-Multi |
|---|---|---|
| Idefics2-8B | 79.33 | 67.67 |
| Idefics2-8B-C | 69.00 | 58.67 |

## E   Prompts and Data Examples

We list a few data examples with prompts in Figures 28 to 33. For NIAH tasks, we provide examples of both VH and MM-NIAH, as their input formats are very different.

## F   Limitation

For limitation of evaluated models, while we already provide an extensive coverage of 46 frontier LCVLMs, there are still some models that we cannot cover due to token efficiency or integration challenges of codebases, as we discussed in Section B.1. We leave those works for future study. Meanwhile, the largest open-source models we evaluated are up to 72B in size (Qwen2-VL-72B and Qwen2.5-VL-72B). As we discussed in Section C, our computational resources are limited to 8×A100 (80G) GPUs; thereby it is hard to deploy and evaluate larger models with over 100B parameters at the input length of 128K tokens, such as Llama4 [4].

For evaluating summarization, we use a model-based metric (See Section A.4) that can provide much better alignment with human judgment than N-gram overlap metrics, such as ROUGE-L. However, we find using GPT-4o to provide the evaluation is expensive, which prevents the long-context community from conducting evaluations with hundreds or even thousands of models. Therefore, it is necessary to find an alternative evaluation method with a lower cost.

## G   Broader Impact

The long-context ability of LVLMs has unlocked a large range of applications, including understanding documents with hundreds of pages and reasoning over dozens of web pages automatically. This ability can also help users to summarize a long document or revise a large-scale code repository. Meanwhile, there are a large number of instruction-following scenarios grounded in complex vision-language contexts, such as long-term dialogue with humans or dialogue-based navigation for robots. Looking ahead, our MMLONGBENCH will serve as a standard evaluation for the whole community to benchmark new LCVLMs and to stimulate the development of models with more efficient vision-language token encodings, more robust position-extrapolation schemes, and improved OCR, multi-modal retrieval, and reasoning capabilities.

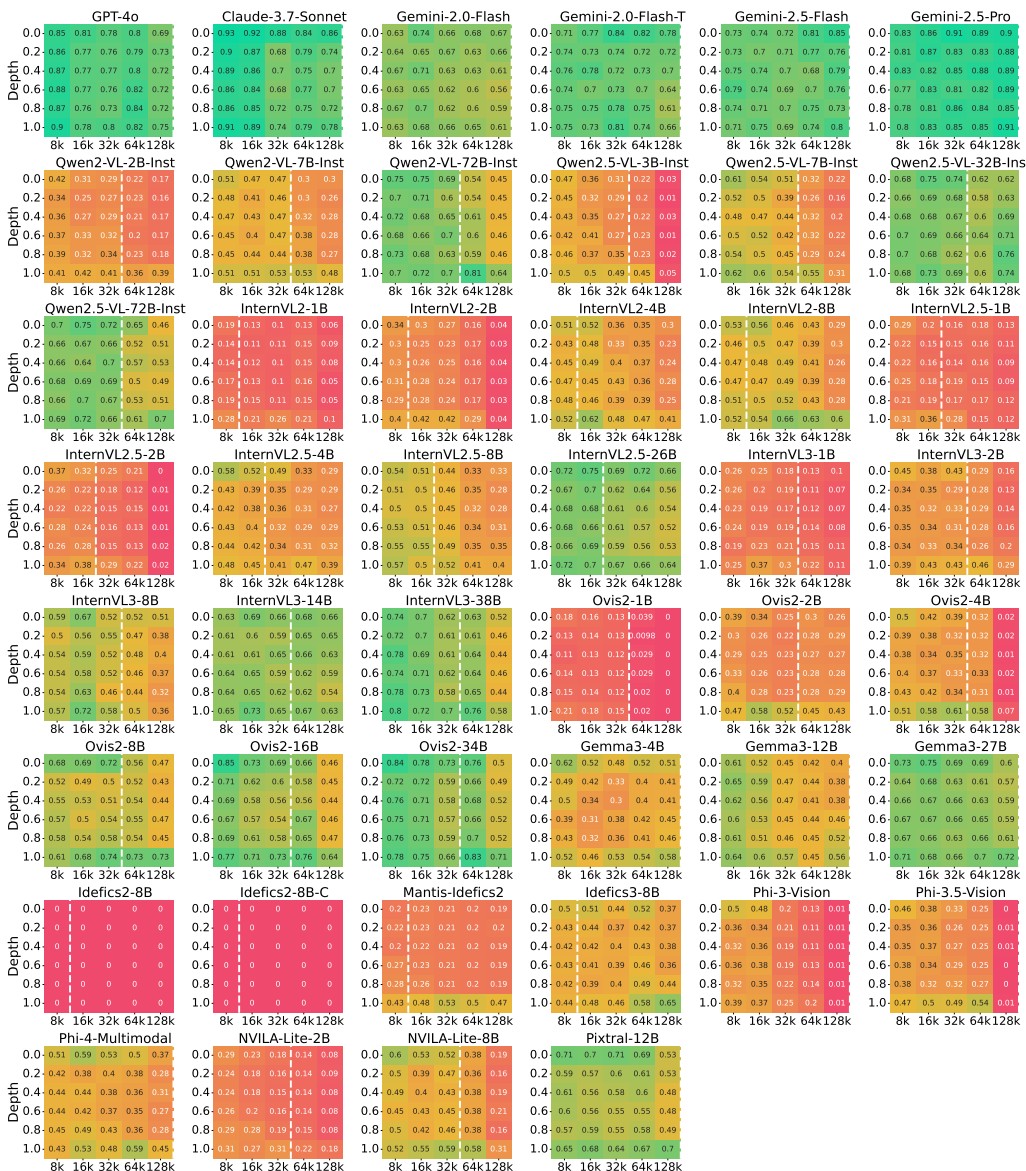

Figure 18: Performance of models on ViQuAE at different depths. Depth is the position of the gold passage, and its values are $[0.0, 0.2, 0.4, 0.6, 0.8, 1.0]$, where $0.0$ is the beginning of the context (the top of each heatmap) and $1.0$ is the end (the bottom of each heatmap).

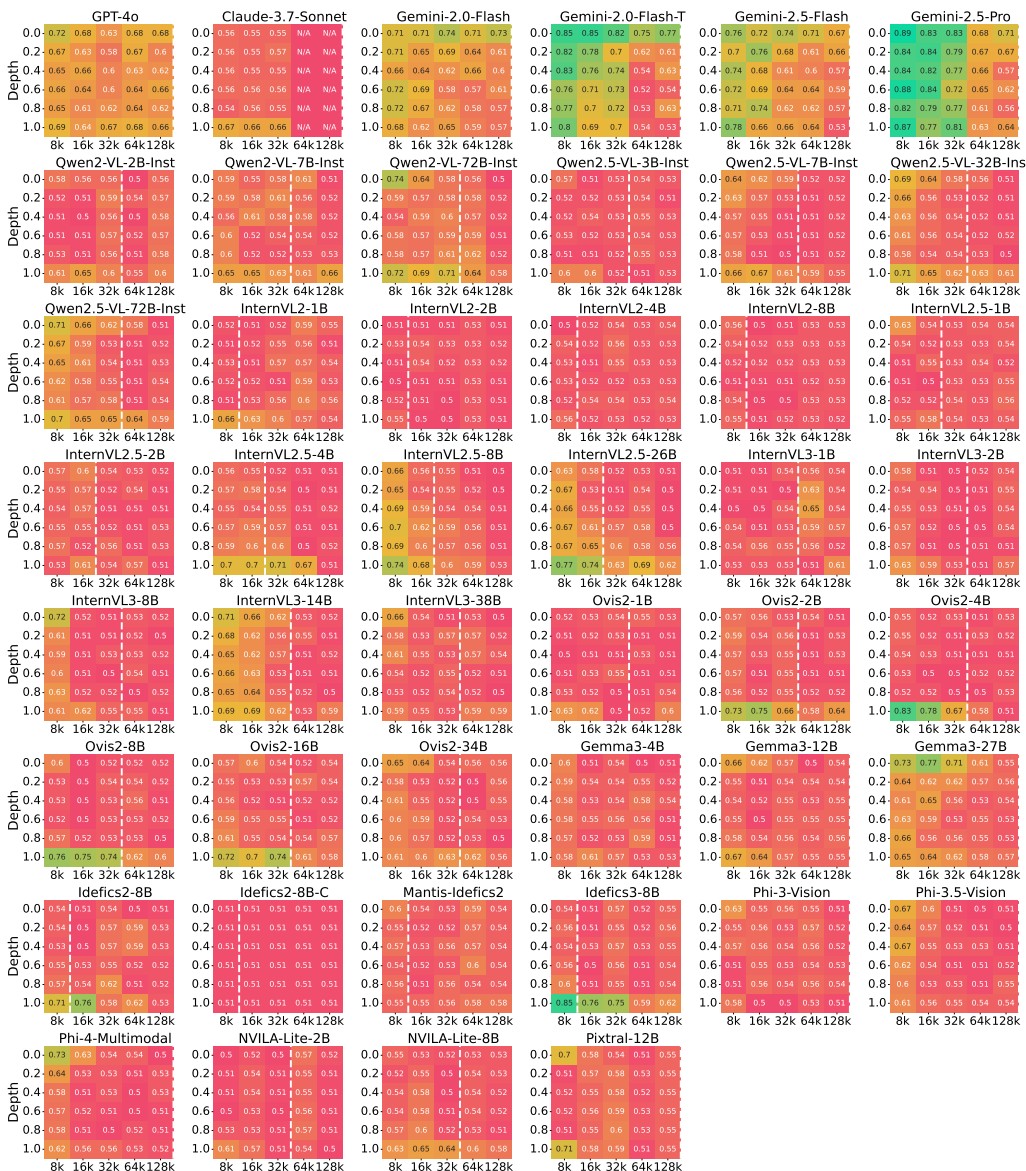

Figure 19: Performance of models on VH-Single at different depths. Depth is the position of the image containing the target object, and its values are $[0.0, 0.2, 0.4, 0.6, 0.8, 1.0]$, where $0.0$ is the beginning of the context (the top of each heatmap) and $1.0$ is the end (the bottom of each heatmap).

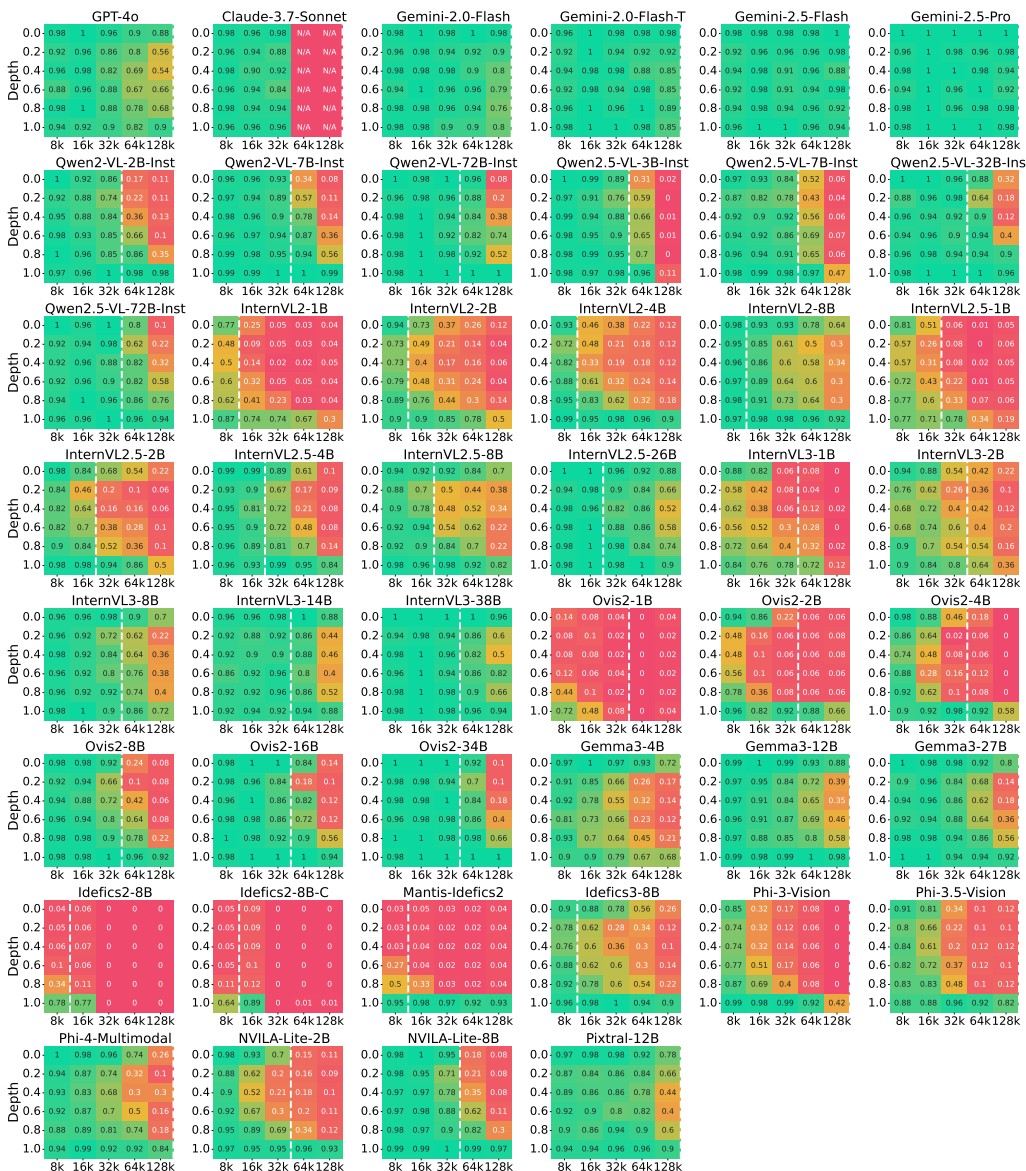

Figure 20: Performance of models on MM-NIAH-Ret (T) at different depths. Depth is the position of the text needle, and its values are $[0.0, 0.2, 0.4, 0.6, 0.8, 1.0]$, where $0.0$ is the beginning of the context (the top of each heatmap) and $1.0$ is the end (the bottom of each heatmap).

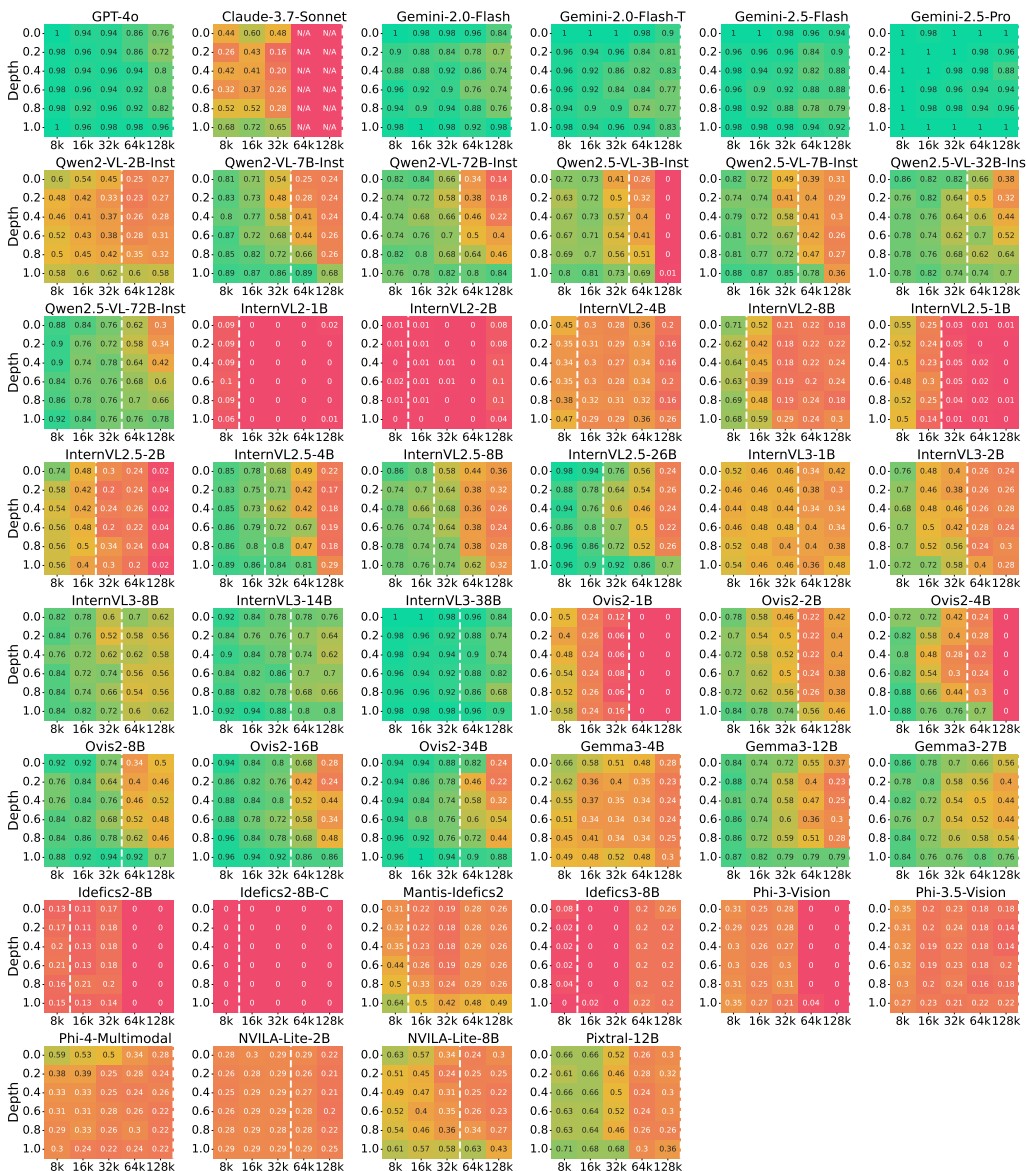

Figure 21: Performance of models on MM-NIAH-Ret (I) at different depths. Depth is the position of the image needle, and its values are $[0.0, 0.2, 0.4, 0.6, 0.8, 1.0]$, where $0.0$ is the beginning of the context (the top of each heatmap) and $1.0$ is the end (the bottom of each heatmap).

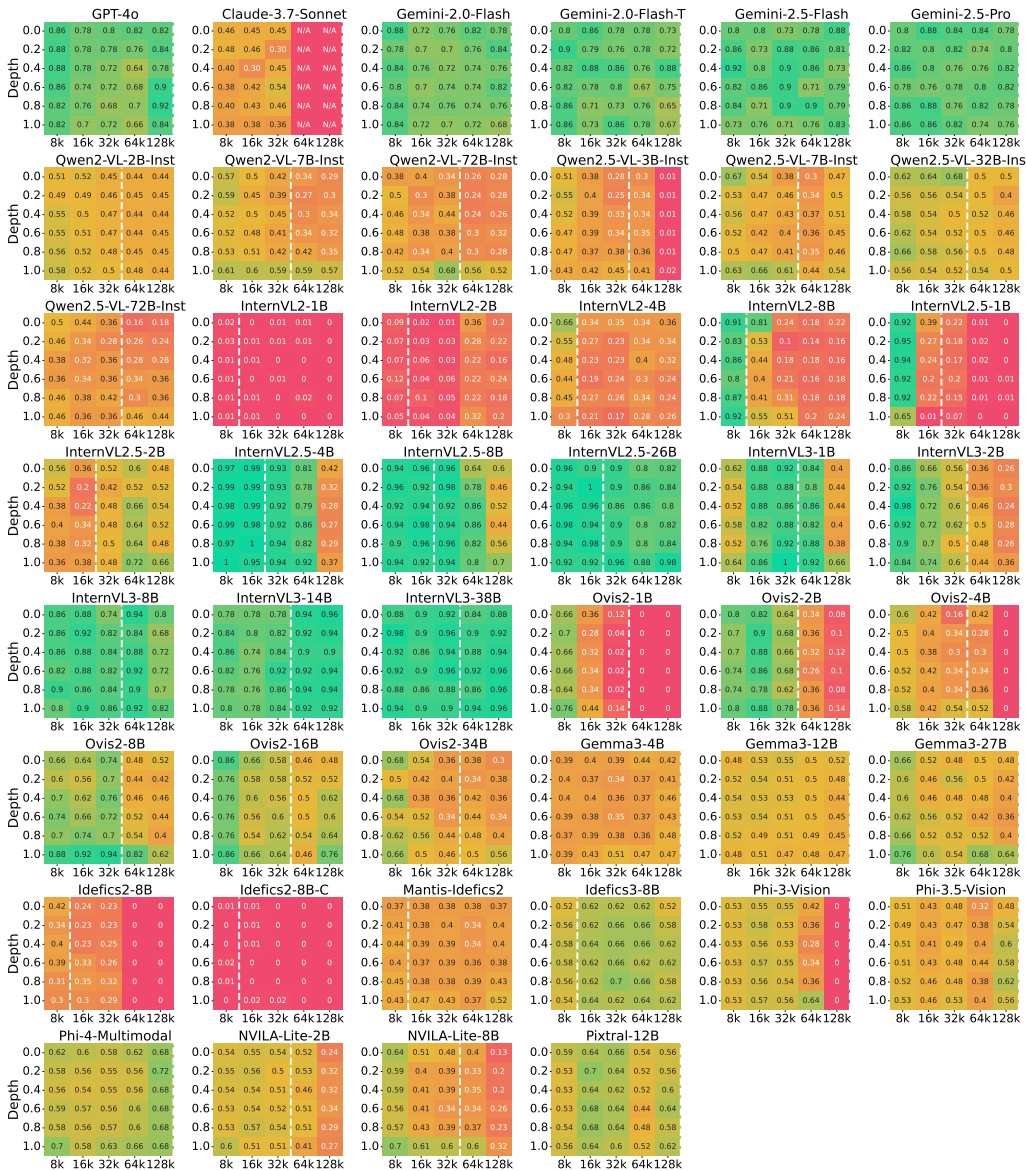

Figure 22: Performance of models on MM-NIAH-Reason (I) at different depths. Depth is the position of the needle image used for reasoning, and its values are $[0.0, 0.2, 0.4, 0.6, 0.8, 1.0]$, where $0.0$ is the beginning of the context (the top of each heatmap) and $1.0$ is the end (the bottom of each heatmap).

**VRAG**

| Model | 8k | 16k | 32k | 64k | 128k |
|---|---|---|---|---|---|
| GPT-4o | 80.5 | 74.7 | 71.8 | 74.2 | 67.3 |
| Claude-3.7-Sonnet | 84.9 | 81.8 | 66.7 | 67.6 | 68.8 |
| Gemini-2.0-Flash | 64.9 | 64.2 | 59.5 | 59.0 | 60.3 |
| Gemini-2.0-Flash-T | 67.0 | 68.5 | 66.7 | 67.0 | 64.4 |
| Gemini-2.5-Flash | 69.8 | 69.3 | 65.1 | 68.6 | 70.6 |
| Gemini-2.5-Pro | 79.8 | 80.9 | 79.9 | 80.8 | 82.7 |
| Qwen2-VL-2B | 36.2 | 31.5 | 32.8 | 31.1 | 27.0 |
| Qwen2-VL-7B | 44.0 | 43.2 | 40.5 | 38.9 | 35.8 |
| Qwen2-VL-72B | 64.3 | 64.0 | 60.1 | 56.1 | 46.9 |
| Qwen2.5-VL-3B | 43.9 | 38.6 | 35.8 | 32.7 | 9.8 |
| Qwen2.5-VL-7B | 50.1 | 48.7 | 43.2 | 36.8 | 31.6 |
| Qwen2.5-VL-32B | 67.8 | 69.1 | 65.5 | 61.9 | 64.6 |
| Qwen2.5-VL-72B | 67.6 | 67.7 | 64.0 | 54.3 | 50.3 |
| InternVL2-1B | 19.5 | 16.8 | 14.0 | 16.1 | 10.2 |
| InternVL2-2B | 27.7 | 25.8 | 22.8 | 18.6 | 4.4 |
| InternVL2-4B | 43.8 | 41.2 | 32.2 | 30.5 | 24.8 |
| InternVL2-8B | 40.5 | 36.5 | 37.6 | 32.6 | 24.4 |
| InternVL2.5-1B | 24.5 | 21.9 | 17.6 | 17.3 | 11.8 |
| InternVL2.5-2B | 26.1 | 23.5 | 16.3 | 14.3 | 2.1 |
| InternVL2.5-4B | 39.0 | 37.5 | 31.4 | 28.7 | 25.8 |
| InternVL2.5-8B | 42.9 | 39.0 | 34.8 | 29.0 | 26.1 |
| InternVL2.5-26B | 56.6 | 53.3 | 48.1 | 50.0 | 47.9 |
| InternVL3-1B | 25.5 | 22.6 | 21.3 | 17.3 | 14.2 |
| InternVL3-2B | 36.5 | 33.9 | 31.4 | 29.7 | 17.5 |
| InternVL3-8B | 52.3 | 51.3 | 45.8 | 40.3 | 36.3 |
| InternVL3-14B | 57.5 | 55.3 | 52.3 | 52.8 | 50.0 |
| InternVL3-38B | 65.7 | 60.8 | 52.2 | 50.4 | 40.3 |
| Ovis2-1B | 17.8 | 17.9 | 13.2 | 1.7 | 0.0 |
| Ovis2-2B | 30.5 | 30.5 | 23.7 | 27.0 | 25.7 |
| Ovis2-4B | 35.9 | 34.1 | 29.5 | 33.6 | 3.8 |
| Ovis2-8B | 52.3 | 48.0 | 47.1 | 47.9 | 42.9 |
| Ovis2-16B | 56.2 | 51.2 | 49.7 | 49.2 | 41.3 |
| Ovis2-34B | 63.4 | 61.5 | 55.5 | 57.2 | 45.7 |
| Gemma3-4B | 48.8 | 41.8 | 40.3 | 44.1 | 42.7 |
| Gemma3-12B | 58.6 | 52.1 | 46.9 | 43.5 | 41.7 |
| Gemma3-27B | 64.8 | 62.1 | 58.8 | 57.5 | 51.5 |
| Idefics2-8B | 0.3 | 0.0 | 0.0 | 0.0 | 0.0 |
| Idefics2-8B-C | 0.0 | 0.0 | 0.0 | 0.0 | 0.0 |
| Mantis-Idefics2 | 27.0 | 23.9 | 27.3 | 27.0 | 27.9 |
| Idefics3-8B | 33.3 | 31.8 | 30.3 | 35.2 | 33.2 |
| Phi-3-Vision | 31.0 | 30.6 | 20.7 | 14.0 | 0.5 |
| Phi-3.5-Vision | 36.9 | 34.7 | 29.8 | 24.0 | 0.6 |
| Phi-4-Multimodal | 36.3 | 37.3 | 35.4 | 32.9 | 25.5 |
| NVILA-Lite-2B | 27.4 | 23.1 | 19.8 | 18.1 | 13.6 |
| NVILA-Lite-8B | 43.2 | 41.6 | 41.8 | 35.8 | 16.3 |
| Pixtral-12B | 53.6 | 51.0 | 47.9 | 45.9 | 43.8 |

**NIAH**

| Model | 8k | 16k | 32k | 64k | 128k |
|---|---|---|---|---|---|
| GPT-4o | 79.6 | 73.8 | 67.5 | 65.4 | 57.1 |
| Claude-3.7-Sonnet | 63.1 | 61.2 | 54.1 | N/A | N/A |
| Gemini-2.0-Flash | 76.8 | 74.1 | 69.7 | 64.6 | 60.9 |
| Gemini-2.0-Flash-T | 80.8 | 79.2 | 76.2 | 68.7 | 64.8 |
| Gemini-2.5-Flash | 84.1 | 81.5 | 79.8 | 76.4 | 72.5 |
| Gemini-2.5-Pro | 84.7 | 82.7 | 79.8 | 76.0 | 73.4 |
| Qwen2-VL-2B | 51.0 | 50.1 | 47.4 | 39.0 | 38.3 |
| Qwen2-VL-7B | 57.9 | 54.3 | 50.8 | 46.5 | 37.3 |
| Qwen2-VL-72B | 63.9 | 61.6 | 57.4 | 51.5 | 38.9 |
| Qwen2.5-VL-3B | 54.1 | 50.8 | 45.0 | 38.7 | 21.6 |
| Qwen2.5-VL-7B | 57.3 | 53.0 | 47.7 | 39.5 | 33.2 |
| Qwen2.5-VL-32B | 61.9 | 61.1 | 58.5 | 53.7 | 41.6 |
| Qwen2.5-VL-72B | 68.3 | 63.5 | 61.9 | 55.8 | 43.1 |
| InternVL2-1B | 31.0 | 27.8 | 27.6 | 26.7 | 23.0 |
| InternVL2-2B | 34.0 | 30.6 | 26.4 | 28.9 | 27.6 |
| InternVL2-4B | 45.2 | 38.4 | 34.3 | 33.7 | 32.0 |
| InternVL2-8B | 54.5 | 49.5 | 40.9 | 37.4 | 33.5 |
| InternVL2.5-1B | 48.6 | 37.2 | 30.7 | 24.6 | 21.9 |
| InternVL2.5-2B | 44.8 | 41.4 | 38.1 | 37.2 | 29.5 |
| InternVL2.5-4B | 59.2 | 55.9 | 52.1 | 44.9 | 29.0 |
| InternVL2.5-8B | 64.1 | 56.6 | 51.8 | 46.8 | 38.4 |
| InternVL2.5-26B | 67.8 | 63.1 | 55.5 | 52.2 | 43.8 |
| InternVL3-1B | 46.1 | 47.2 | 43.8 | 41.3 | 32.1 |
| InternVL3-2B | 55.3 | 48.7 | 41.3 | 36.0 | 31.8 |
| InternVL3-8B | 62.6 | 57.8 | 51.8 | 49.7 | 42.4 |
| InternVL3-14B | 69.5 | 65.1 | 58.2 | 55.8 | 48.3 |
| InternVL3-38B | 70.5 | 66.4 | 62.5 | 57.0 | 52.0 |
| Ovis2-1B | 38.6 | 32.5 | 24.8 | 22.0 | 22.2 |
| Ovis2-2B | 48.3 | 44.1 | 38.6 | 30.5 | 30.9 |
| Ovis2-4B | 51.5 | 43.0 | 35.8 | 32.8 | 22.6 |
| Ovis2-8B | 61.3 | 57.9 | 54.2 | 41.2 | 35.8 |
| Ovis2-16B | 67.3 | 62.7 | 56.5 | 48.7 | 40.7 |
| Ovis2-34B | 65.7 | 60.4 | 57.0 | 52.9 | 40.0 |
| Gemma3-4B | 49.3 | 46.4 | 42.2 | 38.0 | 33.0 |
| Gemma3-12B | 60.7 | 55.9 | 51.4 | 47.5 | 41.7 |
| Gemma3-27B | 66.3 | 61.2 | 56.2 | 51.9 | 44.6 |
| Idefics2-8B | 32.0 | 28.8 | 26.8 | 21.7 | 21.0 |
| Idefics2-8B-C | 23.1 | 24.0 | 20.7 | 20.1 | 20.1 |
| Mantis-Idefics2 | 36.4 | 32.0 | 30.3 | 31.7 | 30.6 |
| Idefics3-8B | 49.2 | 45.2 | 43.1 | 39.6 | 37.5 |
| Phi-3-Vision | 41.1 | 37.7 | 34.6 | 29.0 | 22.7 |
| Phi-3.5-Vision | 45.7 | 39.6 | 34.2 | 29.8 | 30.9 |
| Phi-4-Multimodal | 48.8 | 44.6 | 41.1 | 36.7 | 34.9 |
| NVILA-Lite-2B | 46.0 | 44.0 | 38.7 | 35.6 | 30.0 |
| NVILA-Lite-8B | 52.7 | 47.8 | 43.6 | 36.8 | 29.0 |
| Pixtral-12B | 56.3 | 54.2 | 50.2 | 45.2 | 40.9 |

**ICL**

| Model | 8k | 16k | 32k | 64k | 128k |
|---|---|---|---|---|---|
| GPT-4o | 99.0 | 98.2 | 96.0 | 92.4 | 88.4 |
| Claude-3.7-Sonnet | 97.0 | 94.2 | N/A | N/A | N/A |
| Gemini-2.0-Flash | 99.0 | 97.8 | 97.5 | 93.8 | 87.5 |
| Gemini-2.0-Flash-T | 99.5 | 97.8 | 96.2 | 92.5 | 88.2 |
| Gemini-2.5-Flash | 98.5 | 98.5 | 96.5 | 94.0 | 88.0 |
| Gemini-2.5-Pro | 99.5 | 98.5 | 97.2 | 95.0 | 94.2 |
| Qwen2-VL-2B | 85.8 | 61.5 | 25.2 | 7.8 | 3.0 |
| Qwen2-VL-7B | 90.1 | 56.7 | 28.2 | 9.0 | 7.8 |
| Qwen2-VL-72B | 98.5 | 94.5 | 91.0 | 80.8 | 80.8 |
| Qwen2.5-VL-3B | 95.0 | 69.9 | 19.5 | 7.5 | 9.0 |
| Qwen2.5-VL-7B | 95.6 | 91.5 | 78.5 | 57.2 | 46.2 |
| Qwen2.5-VL-32B | 97.5 | 91.7 | 77.0 | 51.2 | 41.2 |
| Qwen2.5-VL-72B | 98.5 | 95.5 | 92.8 | 74.2 | 73.0 |
| InternVL2-1B | 1.0 | 3.0 | 0.5 | 0.0 | 1.0 |
| InternVL2-2B | 3.5 | 3.0 | 2.2 | 1.0 | 1.2 |
| InternVL2-4B | 68.8 | 27.6 | 1.0 | 0.0 | 0.8 |
| InternVL2-8B | 29.6 | 6.8 | 2.5 | 0.5 | 0.0 |
| InternVL2.5-1B | 5.0 | 0.2 | 0.2 | 0.0 | 2.5 |
| InternVL2.5-2B | 4.4 | 1.5 | 0.0 | 0.2 | 0.2 |
| InternVL2.5-4B | 94.1 | 74.8 | 48.8 | 8.5 | 2.8 |
| InternVL2.5-8B | 96.6 | 58.5 | 45.8 | 13.5 | 1.0 |
| InternVL2.5-26B | 98.5 | 89.2 | 85.0 | 72.5 | 54.0 |
| InternVL3-1B | 23.1 | 8.3 | 3.5 | 1.8 | 0.5 |
| InternVL3-2B | 79.6 | 33.4 | 8.0 | 3.8 | 1.0 |
| InternVL3-8B | 97.6 | 87.2 | 75.0 | 61.8 | 8.5 |
| InternVL3-14B | 96.5 | 87.7 | 80.0 | 65.8 | 53.0 |
| InternVL3-38B | 99.5 | 95.0 | 88.5 | 77.5 | 65.2 |
| Ovis2-1B | 22.1 | 7.3 | 6.0 | 2.0 | 0.8 |
| Ovis2-2B | 3.4 | 0.8 | 0.2 | 0.0 | 0.0 |
| Ovis2-4B | 60.4 | 14.6 | 6.2 | 1.0 | 1.2 |
| Ovis2-8B | 94.5 | 44.4 | 7.8 | 4.0 | 1.0 |
| Ovis2-16B | 96.6 | 91.2 | 73.2 | 66.0 | 36.5 |
| Ovis2-34B | 98.5 | 89.5 | 79.2 | 71.0 | 65.2 |
| Gemma3-4B | 97.6 | 85.5 | 67.5 | 33.0 | 10.8 |
| Gemma3-12B | 99.0 | 96.5 | 93.2 | 82.2 | 59.0 |
| Gemma3-27B | 98.0 | 94.8 | 93.5 | 83.8 | 73.8 |
| Idefics2-8B | 16.2 | 11.3 | 1.5 | 2.5 | 1.5 |
| Idefics2-8B-C | 13.3 | 6.5 | 7.2 | 1.8 | 1.0 |
| Mantis-Idefics2 | 57.6 | 27.3 | 5.2 | 2.5 | 2.2 |
| Idefics3-8B | 25.6 | 12.3 | 4.5 | 0.8 | 2.0 |
| Phi-3-Vision | 54.1 | 21.1 | 5.5 | 2.8 | 0.8 |
| Phi-3.5-Vision | 84.5 | 60.5 | 7.2 | 3.2 | 2.0 |
| Phi-4-Multimodal | 82.3 | 42.5 | 12.0 | 2.8 | 2.2 |
| NVILA-Lite-2B | 47.6 | 20.6 | 7.8 | 0.2 | 0.8 |
| NVILA-Lite-8B | 93.1 | 73.6 | 47.0 | 20.5 | 2.8 |
| Pixtral-12B | 95.0 | 90.0 | 86.0 | 53.2 | 49.8 |

**Summ**

| Model | 8k | 16k | 32k | 64k | 128k |
|---|---|---|---|---|---|
| GPT-4o | 25.1 | 31.1 | 34.3 | 41.0 | 42.4 |
| Claude-3.7-Sonnet | 27.6 | 34.6 | 34.9 | 34.5 | 37.5 |
| Gemini-2.0-Flash | 24.4 | 27.1 | 30.1 | 30.6 | 35.9 |
| Gemini-2.0-Flash-T | 27.7 | 37.9 | 44.3 | 53.0 | 61.2 |
| Gemini-2.5-Flash | 29.2 | 39.4 | 45.9 | 55.3 | 62.4 |
| Gemini-2.5-Pro | 32.0 | 42.8 | 48.1 | 58.0 | 65.3 |
| Qwen2-VL-2B | 13.5 | 17.8 | 13.5 | 16.4 | 16.6 |
| Qwen2-VL-7B | 22.2 | 25.4 | 26.2 | 28.2 | 30.0 |
| Qwen2-VL-72B | 25.1 | 29.2 | 32.7 | 37.6 | 39.1 |
| Qwen2.5-VL-3B | 18.8 | 23.2 | 24.9 | 27.1 | 30.2 |
| Qwen2.5-VL-7B | 23.5 | 29.1 | 30.8 | 32.7 | 39.3 |
| Qwen2.5-VL-32B | 22.8 | 26.3 | 25.8 | 23.0 | 25.2 |
| Qwen2.5-VL-72B | 20.5 | 26.9 | 31.1 | 38.0 | 28.5 |
| InternVL2-1B | 2.6 | 2.9 | 2.6 | 1.9 | 0.0 |
| InternVL2-2B | 7.5 | 9.2 | 8.9 | 4.8 | 2.0 |
| InternVL2-4B | 1.9 | 12.8 | 13.8 | 13.7 | 10.3 |
| InternVL2-8B | 16.4 | 19.8 | 21.9 | 21.9 | 19.0 |
| InternVL2.5-1B | 8.2 | 11.5 | 11.6 | 7.5 | 2.5 |
| InternVL2.5-2B | 10.2 | 12.0 | 12.3 | 11.1 | 5.1 |
| InternVL2.5-4B | 16.0 | 22.6 | 23.9 | 24.1 | 18.3 |
| InternVL2.5-8B | 20.4 | 23.2 | 24.6 | 25.5 | 27.6 |
| InternVL2.5-26B | 19.1 | 23.8 | 26.3 | 27.8 | 29.5 |
| InternVL3-1B | 6.5 | 9.5 | 10.5 | 6.5 | 5.1 |
| InternVL3-2B | 15.6 | 20.7 | 16.6 | 18.1 | 22.4 |
| InternVL3-8B | 22.2 | 28.6 | 32.5 | 36.6 | 40.8 |
| InternVL3-14B | 22.3 | 25.6 | 27.2 | 30.3 | 35.8 |
| InternVL3-38B | 20.7 | 24.8 | 33.1 | 38.4 | 43.6 |
| Ovis2-1B | 8.1 | 8.2 | 3.7 | 1.6 | 0.0 |
| Ovis2-2B | 16.7 | 19.5 | 17.0 | 15.0 | 1.6 |
| Ovis2-4B | 19.6 | 22.5 | 24.6 | 20.7 | 12.5 |
| Ovis2-8B | 23.0 | 29.3 | 30.5 | 32.9 | 28.3 |
| Ovis2-16B | 25.3 | 30.0 | 33.5 | 37.0 | 39.3 |
| Ovis2-34B | 23.5 | 29.8 | 35.7 | 39.6 | 41.6 |
| Gemma3-4B | 15.2 | 20.4 | 21.2 | 24.0 | 20.6 |
| Gemma3-12B | 21.0 | 24.0 | 25.2 | 26.1 | 28.0 |
| Gemma3-27B | 22.9 | 28.5 | 32.0 | 35.5 | 40.7 |
| Idefics2-8B | 2.2 | 1.3 | 0.1 | 0.0 | 0.0 |
| Idefics2-8B-C | 2.8 | 1.6 | 0.0 | 0.0 | 0.0 |
| Mantis-Idefics2 | 1.5 | 1.5 | 0.9 | 0.2 | 0.0 |
| Idefics3-8B | 15.7 | 20.4 | 19.2 | 21.8 | 17.7 |
| Phi-3-Vision | 5.7 | 5.0 | 4.0 | 4.6 | 2.4 |
| Phi-3.5-Vision | 9.2 | 11.4 | 12.5 | 16.8 | 16.1 |
| Phi-4-Multimodal | 12.3 | 17.4 | 17.5 | 18.8 | 15.9 |
| NVILA-Lite-2B | 3.7 | 5.8 | 7.4 | 8.1 | 8.2 |
| NVILA-Lite-8B | 12.8 | 15.3 | 19.3 | 19.9 | 23.3 |
| Pixtral-12B | 22.7 | 29.6 | 33.5 | 36.7 | 38.5 |

**DocVQA**

| Model | 8k | 16k | 32k | 64k | 128k |
|---|---|---|---|---|---|
| GPT-4o | 67.8 | 70.5 | 67.2 | 62.9 | 59.2 |
| Claude-3.7-Sonnet | 56.7 | 52.0 | 43.1 | 48.5 | N/A |
| Gemini-2.0-Flash | 58.7 | 55.4 | 59.4 | 53.8 | 53.6 |
| Gemini-2.0-Flash-T | 68.1 | 68.8 | 69.9 | 64.3 | 63.7 |
| Gemini-2.5-Flash | 67.5 | 66.9 | 68.6 | 62.5 | 59.3 |
| Gemini-2.5-Pro | 71.5 | 70.0 | 70.8 | 69.2 | 70.4 |
| Qwen2-VL-2B | 43.2 | 39.6 | 39.9 | 32.6 | 26.8 |
| Qwen2-VL-7B | 57.5 | 55.4 | 57.2 | 49.7 | 47.5 |
| Qwen2-VL-72B | 69.2 | 65.7 | 66.4 | 60.9 | 53.8 |
| Qwen2.5-VL-3B | 55.5 | 52.0 | 51.7 | 45.0 | 35.6 |
| Qwen2.5-VL-7B | 60.7 | 57.1 | 57.2 | 50.7 | 40.2 |
| Qwen2.5-VL-32B | 67.8 | 66.0 | 65.8 | 58.4 | 53.6 |
| Qwen2.5-VL-72B | 71.4 | 67.5 | 65.8 | 57.3 | 48.7 |
| InternVL2-1B | 15.6 | 10.7 | 9.3 | 4.9 | 4.0 |
| InternVL2-2B | 21.7 | 18.0 | 17.5 | 14.1 | 5.1 |
| InternVL2-4B | 20.6 | 22.8 | 20.9 | 13.3 | 4.6 |
| InternVL2-8B | 42.6 | 34.6 | 30.1 | 21.2 | 18.3 |
| InternVL2.5-1B | 24.2 | 17.9 | 12.5 | 9.2 | 4.1 |
| InternVL2.5-2B | 33.5 | 31.6 | 24.3 | 16.6 | 8.1 |
| InternVL2.5-4B | 51.5 | 44.5 | 41.0 | 33.9 | 10.6 |
| InternVL2.5-8B | 52.7 | 53.2 | 48.1 | 37.4 | 24.6 |
| InternVL2.5-26B | 53.5 | 47.6 | 51.4 | 44.6 | 32.8 |
| InternVL3-1B | 27.9 | 13.5 | 14.3 | 11.9 | 6.9 |
| InternVL3-2B | 42.1 | 38.6 | 35.0 | 22.0 | 18.5 |
| InternVL3-8B | 58.1 | 53.7 | 55.3 | 48.7 | 42.6 |
| InternVL3-14B | 63.3 | 54.1 | 57.5 | 50.0 | 39.4 |
| InternVL3-38B | 66.3 | 63.8 | 62.9 | 52.2 | 47.9 |
| Ovis2-1B | 26.3 | 12.4 | 8.5 | 8.2 | 5.2 |
| Ovis2-2B | 35.5 | 22.1 | 13.3 | 11.9 | 5.2 |
| Ovis2-4B | 50.1 | 39.7 | 29.6 | 18.1 | 9.6 |
| Ovis2-8B | 59.1 | 49.3 | 42.3 | 30.5 | 10.9 |
| Ovis2-16B | 66.5 | 61.2 | 48.5 | 35.4 | 19.3 |
| Ovis2-34B | 59.9 | 55.2 | 45.2 | 33.6 | 23.5 |
| Gemma3-4B | 38.0 | 34.9 | 35.6 | 29.8 | 27.5 |
| Gemma3-12B | 42.7 | 43.2 | 43.2 | 39.2 | 41.3 |
| Gemma3-27B | 49.7 | 49.7 | 45.5 | 46.2 | 45.6 |
| Idefics2-8B | 24.3 | 12.2 | 9.3 | 8.5 | 9.9 |
| Idefics2-8B-C | 12.9 | 6.3 | 0.7 | 0.7 | 1.2 |
| Mantis-Idefics2 | 24.9 | 20.5 | 20.5 | 12.3 | 7.4 |
| Idefics3-8B | 46.3 | 37.1 | 42.0 | 26.4 | 17.3 |
| Phi-3-Vision | 31.4 | 27.9 | 25.5 | 19.2 | 11.5 |
| Phi-3.5-Vision | 40.9 | 33.6 | 33.0 | 25.1 | 15.9 |
| Phi-4-Multimodal | 44.5 | 45.5 | 47.9 | 41.7 | 26.0 |
| NVILA-Lite-2B | 18.7 | 15.6 | 16.6 | 12.3 | 10.6 |
| NVILA-Lite-8B | 30.8 | 32.4 | 25.8 | 21.6 | 20.6 |
| Pixtral-12B | 55.0 | 48.1 | 44.4 | 38.7 | 32.4 |

**Avg.**

| Model | 8k | 16k | 32k | 64k | 128k |
|---|---|---|---|---|---|
| GPT-4o | 70.4 | 69.7 | 67.4 | 67.2 | 62.9 |
| Claude-3.7-Sonnet | 65.9 | 64.8 | N/A | N/A | N/A |
| Gemini-2.0-Flash | 64.8 | 63.7 | 63.2 | 60.3 | 59.6 |
| Gemini-2.0-Flash-T | 68.6 | 70.4 | 70.6 | 69.1 | 68.5 |
| Gemini-2.5-Flash | 69.8 | 71.1 | 71.2 | 71.4 | 70.5 |
| Gemini-2.5-Pro | 73.5 | 75.0 | 75.2 | 75.8 | 77.2 |
| Qwen2-VL-2B | 45.9 | 40.1 | 31.8 | 25.4 | 22.3 |
| Qwen2-VL-7B | 54.3 | 47.0 | 40.6 | 34.5 | 31.6 |
| Qwen2-VL-72B | 64.2 | 63.0 | 61.5 | 57.4 | 51.9 |
| Qwen2.5-VL-3B | 53.5 | 46.9 | 35.4 | 30.2 | 21.2 |
| Qwen2.5-VL-7B | 57.4 | 55.9 | 51.5 | 43.4 | 38.1 |
| Qwen2.5-VL-32B | 63.6 | 62.9 | 58.5 | 49.7 | 45.2 |
| Qwen2.5-VL-72B | 65.2 | 64.2 | 63.1 | 55.9 | 48.7 |
| InternVL2-1B | 13.9 | 12.3 | 10.8 | 9.9 | 7.7 |
| InternVL2-2B | 18.9 | 17.3 | 15.6 | 13.5 | 8.1 |
| InternVL2-4B | 36.0 | 28.6 | 20.5 | 18.3 | 14.5 |
| InternVL2-8B | 36.7 | 29.4 | 26.6 | 22.7 | 19.0 |
| InternVL2.5-1B | 22.1 | 17.7 | 14.5 | 11.7 | 8.6 |
| InternVL2.5-2B | 23.8 | 22.0 | 18.2 | 15.9 | 9.0 |
| InternVL2.5-4B | 52.0 | 47.1 | 39.4 | 28.0 | 17.3 |
| InternVL2.5-8B | 55.3 | 46.1 | 41.0 | 30.4 | 23.5 |
| InternVL2.5-26B | 59.1 | 55.4 | 53.3 | 49.4 | 41.6 |
| InternVL3-1B | 25.8 | 20.2 | 18.7 | 15.8 | 11.7 |
| InternVL3-2B | 45.8 | 35.1 | 26.5 | 21.9 | 18.2 |
| InternVL3-8B | 58.5 | 55.7 | 52.1 | 47.4 | 34.1 |
| InternVL3-14B | 61.8 | 57.5 | 55.1 | 50.9 | 45.3 |
| InternVL3-38B | 64.5 | 62.1 | 59.9 | 55.1 | 49.8 |
| Ovis2-1B | 22.6 | 15.7 | 11.2 | 7.1 | 5.6 |
| Ovis2-2B | 26.9 | 23.4 | 18.6 | 16.9 | 12.7 |
| Ovis2-4B | 43.5 | 30.8 | 25.2 | 21.3 | 9.9 |
| Ovis2-8B | 58.0 | 45.8 | 36.4 | 31.3 | 23.8 |
| Ovis2-16B | 62.4 | 59.3 | 52.3 | 47.3 | 35.4 |
| Ovis2-34B | 62.2 | 59.3 | 54.5 | 50.9 | 43.2 |
| Gemma3-4B | 49.8 | 45.8 | 41.4 | 33.8 | 26.9 |
| Gemma3-12B | 56.4 | 54.4 | 52.0 | 47.7 | 42.3 |
| Gemma3-27B | 60.4 | 59.2 | 57.2 | 55.0 | 51.2 |
| Idefics2-8B | 15.0 | 10.7 | 7.5 | 6.5 | 6.5 |
| Idefics2-8B-C | 10.4 | 7.7 | 5.7 | 4.5 | 4.5 |
| Mantis-Idefics2 | 29.5 | 21.0 | 16.9 | 14.7 | 13.6 |
| Idefics3-8B | 34.0 | 29.4 | 27.8 | 24.7 | 21.5 |
| Phi-3-Vision | 32.6 | 24.5 | 18.1 | 13.9 | 7.6 |
| Phi-3.5-Vision | 43.4 | 36.0 | 23.3 | 19.8 | 13.1 |
| Phi-4-Multimodal | 44.8 | 37.5 | 30.8 | 26.6 | 20.9 |
| NVILA-Lite-2B | 28.7 | 21.8 | 18.0 | 14.9 | 12.6 |
| NVILA-Lite-8B | 46.5 | 42.1 | 35.5 | 26.9 | 18.4 |
| Pixtral-12B | 56.5 | 54.6 | 52.4 | 43.9 | 41.1 |

Figure 23: Results of all 46 models on MMLONGBENCH at various lengths.

| Model | InfoSeek 8k | 16k | 32k | 64k | 128k | ViQuAE 8k | 16k | 32k | 64k | 128k | GovReport 8k | 16k | 32k | 64k | 128k | Multi-LexSum 8k | 16k | 32k | 64k | 128k |
|---|---|---|---|---|---|---|---|---|---|---|---|---|---|---|---|---|---|---|---|---|
| GPT-4o | 74.1 | 71.7 | 67.3 | 67.3 | 62.7 | 86.9 | 77.8 | 76.4 | 81.0 | 72.0 | 14.9 | 19.7 | 24.1 | 36.4 | 37.6 | 35.2 | 42.5 | 44.5 | 45.5 | 47.2 |
| Claude-3.7-Sonnet | 80.7 | 76.3 | 60.0 | 57.0 | 62.7 | 89.2 | 87.2 | 73.3 | 78.2 | 75.0 | 15.1 | 23.4 | 32.3 | 19.4 | 25.5 | 40.2 | 45.9 | 37.6 | 49.7 | 49.6 |
| Gemini-2.0-Flash | 65.0 | 59.3 | 54.7 | 55.0 | 59.0 | 64.9 | 69.1 | 64.3 | 62.9 | 61.7 | 9.2 | 10.1 | 12.9 | 15.2 | 24.0 | 39.6 | 44.1 | 47.4 | 46.1 | 47.8 |
| Gemini-2.0-Flash-T | 59.7 | 62.3 | 56.3 | 59.7 | 60.3 | 74.4 | 74.6 | 77.0 | 74.3 | 68.5 | 17.0 | 28.5 | 40.7 | 57.7 | 70.8 | 38.4 | 47.3 | 47.9 | 48.3 | 51.6 |
| Gemini-2.5-Flash | 65.3 | 65.7 | 60.0 | 63.1 | 63.0 | 74.2 | 73.0 | 70.2 | 74.2 | 78.2 | 18.7 | 31.9 | 41.6 | 59.9 | 72.9 | 39.8 | 47.0 | 50.2 | 50.7 | 51.9 |
| Gemini-2.5-Pro | 79.0 | 78.0 | 74.7 | 76.3 | 76.7 | 80.5 | 83.8 | 85.2 | 85.2 | 88.7 | 20.1 | 34.8 | 44.0 | 63.3 | 76.2 | 43.9 | 50.8 | 52.3 | 52.8 | 54.3 |
| Qwen2-VL-2B | 34.7 | 31.7 | 33.7 | 38.3 | 33.3 | 37.8 | 31.4 | 32.0 | 23.9 | 20.7 | 4.9 | 8.0 | 5.6 | 7.4 | 11.4 | 22.0 | 27.5 | 21.4 | 25.4 | 21.7 |
| Qwen2-VL-7B | 40.3 | 42.3 | 33.7 | 40.7 | 40.3 | 47.7 | 44.1 | 47.3 | 37.1 | 31.2 | 12.8 | 14.6 | 16.8 | 19.6 | 24.0 | 31.6 | 36.2 | 35.6 | 36.8 | 36.0 |
| Qwen2-VL-72B | 57.3 | 58.0 | 54.0 | 50.7 | 45.3 | 71.3 | 70.0 | 66.2 | 61.5 | 48.5 | 13.2 | 14.9 | 18.9 | 29.0 | 33.2 | 37.0 | 43.4 | 46.4 | 46.1 | 45.0 |
| Qwen2.5-VL-3B | 42.7 | 39.0 | 38.7 | 40.0 | 17.0 | 45.0 | 38.1 | 33.0 | 25.5 | 2.5 | 9.6 | 11.2 | 13.3 | 21.6 | 27.3 | 28.0 | 35.2 | 36.5 | 32.7 | 33.1 |
| Qwen2.5-VL-7B | 45.3 | 45.0 | 40.7 | 39.0 | 40.7 | 54.8 | 52.3 | 45.8 | 34.6 | 22.5 | 13.4 | 19.8 | 20.7 | 23.6 | 35.6 | 33.6 | 38.4 | 40.9 | 41.9 | 43.0 |
| Qwen2.5-VL-32B | 67.0 | 67.7 | 63.3 | 63.3 | 60.0 | 68.6 | 70.6 | 67.7 | 60.5 | 69.2 | 9.1 | 11.1 | 11.6 | 14.1 | 21.3 | 36.5 | 41.4 | 40.0 | 31.9 | 29.2 |
| Qwen2.5-VL-72B | 67.3 | 65.7 | 59.7 | 52.3 | 47.3 | 67.8 | 69.8 | 68.3 | 56.2 | 53.3 | 7.8 | 12.9 | 19.8 | 32.0 | 29.2 | 33.2 | 41.0 | 42.5 | 44.1 | 27.9 |
| InternVL2-1B | 20.7 | 19.7 | 15.0 | 16.7 | 13.3 | 18.3 | 14.0 | 13.0 | 15.5 | 7.2 | 1.0 | 1.1 | 1.2 | 1.4 | 0.0 | 4.2 | 4.8 | 3.9 | 2.5 | 0.1 |
| InternVL2-2B | 23.3 | 22.0 | 18.0 | 18.7 | 5.3 | 32.0 | 29.5 | 27.5 | 18.5 | 3.5 | 2.7 | 5.9 | 6.9 | 4.1 | 1.4 | 12.2 | 12.6 | 10.9 | 5.6 | 2.6 |
| InternVL2-4B | 40.0 | 32.3 | 24.7 | 22.7 | 21.0 | 47.5 | 50.2 | 39.8 | 38.4 | 28.5 | 2.8 | 1.6 | 3.4 | 4.6 | 4.0 | 0.9 | 24.1 | 24.1 | 22.8 | 16.5 |
| InternVL2-8B | 31.7 | 22.3 | 23.7 | 20.3 | 15.3 | 49.3 | 50.7 | 51.5 | 44.8 | 33.5 | 8.5 | 10.5 | 10.8 | 16.2 | 13.2 | 24.2 | 29.0 | 32.9 | 27.6 | 24.7 |
| InternVL2.5-1B | 24.3 | 23.3 | 17.0 | 18.7 | 12.7 | 24.8 | 20.5 | 18.2 | 15.8 | 11.0 | 2.6 | 4.4 | 5.6 | 6.4 | 1.7 | 13.8 | 18.5 | 17.5 | 8.6 | 3.3 |
| InternVL2.5-2B | 23.7 | 19.7 | 13.0 | 13.0 | 3.0 | 28.5 | 27.4 | 19.7 | 15.7 | 1.2 | 4.0 | 4.5 | 6.7 | 8.2 | 2.5 | 16.4 | 19.4 | 17.9 | 14.0 | 7.7 |
| InternVL2.5-4B | 32.0 | 32.7 | 25.0 | 23.7 | 20.7 | 46.0 | 42.4 | 37.8 | 33.7 | 30.8 | 6.3 | 10.5 | 12.3 | 13.2 | 11.2 | 25.8 | 34.7 | 35.4 | 35.0 | 25.4 |
| InternVL2.5-8B | 32.0 | 26.7 | 22.7 | 22.7 | 19.7 | 53.8 | 51.3 | 47.0 | 35.3 | 32.5 | 9.6 | 13.0 | 13.5 | 16.2 | 19.4 | 31.2 | 33.4 | 35.7 | 34.8 | 35.8 |
| InternVL2.5-26B | 44.0 | 36.7 | 33.0 | 37.7 | 38.3 | 69.1 | 70.0 | 63.2 | 62.3 | 57.5 | 9.6 | 11.6 | 15.4 | 19.2 | 23.4 | 28.6 | 35.9 | 37.3 | 36.5 | 35.6 |
| InternVL3-1B | 27.3 | 21.7 | 22.0 | 20.3 | 19.3 | 23.6 | 23.6 | 20.7 | 14.2 | 9.0 | 1.7 | 4.1 | 6.4 | 6.1 | 4.4 | 11.4 | 14.9 | 14.6 | 6.9 | 5.9 |
| InternVL3-2B | 36.3 | 32.3 | 27.3 | 28.0 | 17.0 | 36.6 | 35.5 | 35.5 | 31.4 | 18.0 | 4.5 | 12.4 | 7.7 | 9.6 | 16.3 | 26.6 | 29.1 | 25.6 | 26.6 | 28.5 |
| InternVL3-8B | 49.7 | 39.7 | 39.0 | 32.7 | 33.7 | 55.0 | 62.9 | 52.5 | 47.9 | 39.0 | 12.7 | 19.7 | 27.1 | 37.7 | 44.4 | 31.6 | 37.6 | 37.9 | 35.4 | 37.3 |
| InternVL3-14B | 51.7 | 44.7 | 42.0 | 40.7 | 38.3 | 63.4 | 65.8 | 62.7 | 65.0 | 61.7 | 10.3 | 11.6 | 14.1 | 20.8 | 31.7 | 34.4 | 39.6 | 40.4 | 39.7 | 39.9 |
| InternVL3-38B | 55.3 | 50.7 | 42.0 | 35.3 | 32.3 | 76.0 | 70.8 | 62.3 | 65.5 | 48.3 | 7.3 | 9.2 | 21.2 | 33.7 | 43.0 | 34.2 | 40.3 | 45.1 | 43.0 | 44.2 |
| Ovis2-1B | 20.3 | 21.3 | 13.7 | 1.0 | 0.0 | 15.2 | 14.5 | 12.8 | 2.5 | 0.0 | 4.4 | 5.2 | 4.0 | 1.5 | 0.0 | 11.7 | 11.2 | 3.5 | 1.8 | 0.0 |
| Ovis2-2B | 25.0 | 28.3 | 19.3 | 22.7 | 21.0 | 36.0 | 32.7 | 28.0 | 31.4 | 30.3 | 6.0 | 10.4 | 9.9 | 10.6 | 2.2 | 27.3 | 28.6 | 24.0 | 19.4 | 1.1 |
| Ovis2-4B | 28.3 | 26.7 | 20.0 | 30.7 | 5.0 | 43.4 | 41.6 | 39.0 | 36.6 | 2.5 | 9.8 | 10.9 | 14.3 | 12.6 | 6.8 | 29.4 | 34.0 | 35.0 | 28.9 | 18.2 |
| Ovis2-8B | 45.7 | 38.7 | 34.3 | 38.7 | 36.0 | 58.9 | 57.4 | 59.8 | 57.2 | 49.8 | 12.0 | 17.7 | 20.3 | 24.2 | 22.7 | 34.0 | 40.9 | 40.6 | 41.7 | 34.0 |
| Ovis2-16B | 39.3 | 38.7 | 37.7 | 33.7 | 34.0 | 73.0 | 63.7 | 61.7 | 64.7 | 48.7 | 13.7 | 20.2 | 24.5 | 29.7 | 35.6 | 37.0 | 39.9 | 42.6 | 44.2 | 43.0 |
| Ovis2-34B | 50.0 | 49.7 | 49.0 | 43.0 | 37.0 | 76.8 | 73.3 | 62.0 | 71.5 | 54.3 | 11.6 | 17.1 | 27.1 | 35.7 | 39.4 | 35.4 | 42.4 | 44.3 | 43.6 | 43.9 |
| Gemma3-4B | 48.7 | 44.3 | 41.0 | 43.3 | 38.3 | 49.0 | 39.3 | 39.7 | 44.9 | 47.0 | 3.7 | 8.6 | 10.8 | 16.9 | 8.9 | 26.8 | 32.2 | 31.6 | 31.0 | 32.4 |
| Gemma3-12B | 54.7 | 48.7 | 46.0 | 44.3 | 38.3 | 62.5 | 55.6 | 47.8 | 43.6 | 45.0 | 7.8 | 9.3 | 8.3 | 10.1 | 14.5 | 34.2 | 38.7 | 42.0 | 42.1 | 41.5 |
| Gemma3-27B | 61.3 | 55.3 | 52.0 | 49.7 | 41.7 | 68.3 | 68.8 | 65.5 | 65.4 | 61.3 | 10.1 | 16.0 | 21.2 | 28.8 | 36.3 | 35.7 | 40.9 | 42.8 | 42.2 | 45.1 |
| Idefics2-8B | 0.7 | 0.0 | 0.0 | 0.0 | 0.0 | 0.0 | 0.0 | 0.0 | 0.0 | 0.0 | 0.6 | 0.8 | 0.2 | 0.0 | 0.0 | 3.8 | 1.8 | 0.0 | 0.0 | 0.0 |
| Idefics2-8B-C | 0.0 | 0.0 | 0.0 | 0.0 | 0.0 | 0.0 | 0.0 | 0.0 | 0.0 | 0.0 | 1.9 | 1.8 | 0.0 | 0.0 | 0.0 | 3.7 | 1.5 | 0.0 | 0.0 | 0.0 |
| Mantis-Idefics2 | 27.7 | 20.7 | 28.3 | 29.3 | 32.0 | 26.4 | 27.2 | 26.3 | 24.7 | 23.8 | 0.6 | 0.4 | 0.2 | 0.1 | 0.1 | 2.3 | 2.6 | 1.7 | 0.3 | 0.0 |
| Idefics3-8B | 23.0 | 19.7 | 19.7 | 22.0 | 23.7 | 43.6 | 43.9 | 41.0 | 48.4 | 42.8 | 6.6 | 10.1 | 9.1 | 11.0 | 8.8 | 24.7 | 30.8 | 29.3 | 32.7 | 26.6 |
| Phi-3-Vision | 24.7 | 23.7 | 20.3 | 14.7 | 0.0 | 37.3 | 37.6 | 21.0 | 13.4 | 1.0 | 1.4 | 2.5 | 3.0 | 2.6 | 0.5 | 9.9 | 7.5 | 5.1 | 6.7 | 4.3 |
| Phi-3.5-Vision | 34.3 | 32.0 | 27.0 | 18.0 | 0.7 | 39.4 | 37.5 | 32.7 | 30.1 | 0.5 | 2.0 | 1.8 | 4.6 | 10.7 | 11.7 | 16.4 | 20.9 | 20.4 | 22.9 | 20.5 |
| Phi-4-Multimodal | 28.0 | 27.3 | 27.7 | 23.3 | 18.3 | 44.6 | 47.4 | 43.2 | 42.5 | 32.7 | 5.0 | 8.7 | 6.6 | 10.7 | 9.6 | 19.6 | 26.2 | 28.4 | 26.9 | 22.2 |
| NVILA-Lite-2B | 28.0 | 24.0 | 20.3 | 21.0 | 17.3 | 26.9 | 22.1 | 19.2 | 15.2 | 9.8 | 0.4 | 2.0 | 2.5 | 6.2 | 4.8 | 6.9 | 9.5 | 12.3 | 10.0 | 11.5 |
| NVILA-Lite-8B | 35.7 | 38.0 | 35.0 | 30.3 | 12.3 | 50.8 | 45.2 | 48.7 | 41.2 | 20.3 | 4.0 | 7.2 | 11.2 | 12.6 | 17.9 | 21.5 | 23.5 | 27.5 | 27.3 | 28.7 |
| Pixtral-12B | 44.7 | 40.7 | 35.3 | 30.3 | 34.0 | 62.5 | 61.4 | 60.5 | 61.4 | 53.5 | 12.5 | 19.7 | 26.0 | 33.7 | 35.6 | 32.8 | 39.5 | 41.0 | 39.7 | 41.5 |

Figure 24: Results of 46 models on categories VRAG and Summ at various lengths.

**VH-Single**

| Model | 8k | 16k | 32k | 64k | 128k |
|---|---|---|---|---|---|
| GPT-4o | 67.1 | 64.3 | 61.7 | 65.7 | 64.0 |
| Claude-3.7-Sonnet | 50.5 | 51.8 | 52.0 | N/A | N/A |
| Gemini-2.0-Flash | 69.7 | 66.3 | 65.1 | 62.5 | 61.5 |
| Gemini-2.0-Flash-T | 80.1 | 74.8 | 73.4 | 58.2 | 62.0 |
| Gemini-2.5-Flash | 73.6 | 70.9 | 65.7 | 63.8 | 59.9 |
| Gemini-2.5-Pro | 85.4 | 81.8 | 78.0 | 64.9 | 62.7 |
| Qwen2-VL-2B | 53.3 | 54.0 | 57.3 | 50.3 | 57.7 |
| Qwen2-VL-7B | 59.8 | 57.2 | 53.3 | 53.5 | 51.3 |
| Qwen2-VL-72B | 62.5 | 60.5 | 61.2 | 59.3 | 50.8 |
| Qwen2.5-VL-3B | 54.2 | 50.3 | 52.0 | 53.5 | 51.7 |
| Qwen2.5-VL-7B | 60.8 | 57.5 | 52.5 | 52.5 | 52.5 |
| Qwen2.5-VL-32B | 64.7 | 58.5 | 55.8 | 54.7 | 51.3 |
| Qwen2.5-VL-72B | 66.0 | 61.0 | 57.8 | 54.3 | 53.5 |
| InternVL2-1B | 52.5 | 53.7 | 51.8 | 55.7 | 51.2 |
| InternVL2-2B | 50.3 | 50.8 | 51.2 | 53.0 | 51.3 |
| InternVL2-4B | 51.0 | 51.5 | 54.0 | 52.8 | 53.3 |
| InternVL2-8B | 51.3 | 50.3 | 51.2 | 52.8 | 52.8 |
| InternVL2.5-1B | 54.0 | 53.5 | 53.3 | 53.3 | 50.8 |
| InternVL2.5-2B | 55.2 | 57.0 | 53.2 | 52.8 | 51.5 |
| InternVL2.5-4B | 59.0 | 59.3 | 58.2 | 53.3 | 50.3 |
| InternVL2.5-8B | 68.8 | 59.8 | 56.7 | 54.7 | 50.3 |
| InternVL2.5-26B | 67.8 | 61.0 | 55.8 | 57.8 | 52.8 |
| InternVL3-1B | 50.8 | 51.0 | 52.2 | 58.3 | 55.3 |
| InternVL3-2B | 57.3 | 53.3 | 50.3 | 50.5 | 54.2 |
| InternVL3-8B | 62.7 | 52.8 | 50.7 | 51.2 | 51.0 |
| InternVL3-14B | 67.3 | 64.3 | 57.5 | 51.5 | 51.8 |
| InternVL3-38B | 58.5 | 53.5 | 52.5 | 52.0 | 50.7 |
| Ovis2-1B | 53.3 | 53.8 | 52.2 | 52.2 | 53.3 |
| Ovis2-2B | 59.8 | 56.8 | 56.7 | 52.8 | 54.0 |
| Ovis2-4B | 58.7 | 56.2 | 53.7 | 50.8 | 52.0 |
| Ovis2-8B | 58.5 | 54.5 | 56.5 | 51.0 | 50.5 |
| Ovis2-16B | 60.0 | 57.5 | 54.5 | 50.8 | 54.5 |
| Ovis2-34B | 60.8 | 58.0 | 51.5 | 54.2 | 52.3 |
| Gemma3-4B | 58.3 | 54.0 | 53.3 | 54.2 | 51.7 |
| Gemma3-12B | 59.2 | 55.5 | 51.0 | 51.8 | 53.2 |
| Gemma3-27B | 65.3 | 63.8 | 60.0 | 56.5 | 54.7 |
| Idefics2-8B | 57.3 | 53.0 | 57.2 | 54.8 | 50.7 |
| Idefics2-8B-C | 50.0 | 50.0 | 50.0 | 50.0 | 50.0 |
| Mantis-Idefics2 | 56.2 | 52.8 | 54.3 | 57.5 | 54.7 |
| Idefics3-8B | 60.8 | 55.3 | 59.2 | 50.0 | 56.0 |
| Phi-3-Vision | 55.8 | 54.7 | 53.3 | 55.2 | 52.2 |
| Phi-3.5-Vision | 63.5 | 55.8 | 52.5 | 50.0 | 51.5 |
| Phi-4-Multimodal | 62.0 | 53.3 | 52.8 | 51.7 | 51.0 |
| NVILA-Lite-2B | 52.7 | 53.8 | 50.0 | 55.7 | 51.0 |
| NVILA-Lite-8B | 55.8 | 58.2 | 52.5 | 51.3 | 50.5 |
| Pixtral-12B | 56.5 | 51.2 | 55.0 | 51.8 | 55.0 |

**VH-Multi**

| Model | 8k | 16k | 32k | 64k | 128k |
|---|---|---|---|---|---|
| GPT-4o | 70.3 | 65.0 | 63.3 | 61.0 | 50.7 |
| Claude-3.7-Sonnet | 57.3 | 54.7 | 50.3 | N/A | N/A |
| Gemini-2.0-Flash | 62.0 | 61.8 | 58.1 | 54.8 | 56.4 |
| Gemini-2.0-Flash-T | 66.5 | 67.0 | 64.6 | 63.7 | 63.8 |
| Gemini-2.5-Flash | 78.6 | 72.4 | 67.5 | 65.4 | 61.4 |
| Gemini-2.5-Pro | 76.5 | 72.9 | 68.9 | 70.1 | 67.7 |
| Qwen2-VL-2B | 55.5 | 54.5 | 52.2 | 51.8 | 51.0 |
| Qwen2-VL-7B | 54.8 | 53.5 | 51.8 | 51.7 | 57.3 |
| Qwen2-VL-72B | 60.7 | 57.7 | 55.7 | 53.3 | 54.0 |
| Qwen2.5-VL-3B | 51.8 | 52.2 | 54.2 | 51.7 | 54.2 |
| Qwen2.5-VL-7B | 54.8 | 53.7 | 54.0 | 55.2 | 54.8 |
| Qwen2.5-VL-32B | 57.5 | 56.0 | 55.8 | 56.2 | 53.7 |
| Qwen2.5-VL-72B | 64.3 | 57.7 | 54.5 | 54.0 | 55.0 |
| InternVL2-1B | 50.5 | 51.0 | 54.8 | 51.8 | 53.0 |
| InternVL2-2B | 50.8 | 50.5 | 51.7 | 53.5 | 55.2 |
| InternVL2-4B | 55.0 | 57.8 | 55.3 | 56.0 | 58.3 |
| InternVL2-8B | 53.3 | 60.7 | 59.3 | 57.0 | 57.0 |
| InternVL2.5-1B | 56.0 | 56.2 | 55.5 | 56.2 | 50.3 |
| InternVL2.5-2B | 52.5 | 53.7 | 54.7 | 52.7 | 51.3 |
| InternVL2.5-4B | 56.0 | 53.5 | 55.0 | 55.5 | 50.2 |
| InternVL2.5-8B | 57.3 | 53.5 | 50.7 | 52.5 | 53.5 |
| InternVL2.5-26B | 62.8 | 59.0 | 56.0 | 56.7 | 51.8 |
| InternVL3-1B | 52.8 | 54.7 | 55.3 | 53.8 | 55.2 |
| InternVL3-2B | 54.7 | 52.7 | 52.3 | 51.7 | 53.0 |
| InternVL3-8B | 53.5 | 50.2 | 50.0 | 56.0 | 51.5 |
| InternVL3-14B | 57.7 | 57.2 | 54.8 | 51.0 | 50.5 |
| InternVL3-38B | 57.0 | 55.3 | 54.7 | 51.3 | 50.7 |
| Ovis2-1B | 55.2 | 55.0 | 54.8 | 55.3 | 55.5 |
| Ovis2-2B | 55.0 | 55.0 | 54.0 | 53.7 | 54.2 |
| Ovis2-4B | 55.5 | 55.2 | 54.8 | 54.3 | 55.0 |
| Ovis2-8B | 54.7 | 54.5 | 55.0 | 54.5 | 55.0 |
| Ovis2-16B | 57.0 | 52.7 | 54.7 | 51.0 | 50.7 |
| Ovis2-34B | 59.3 | 54.0 | 53.7 | 53.3 | 50.0 |
| Gemma3-4B | 52.7 | 59.0 | 52.8 | 56.5 | 52.7 |
| Gemma3-12B | 56.3 | 51.3 | 52.5 | 51.0 | 53.2 |
| Gemma3-27B | 57.2 | 53.5 | 56.3 | 60.3 | 57.0 |
| Idefics2-8B | 53.5 | 51.2 | 52.0 | 53.5 | 54.2 |
| Idefics2-8B-C | 50.3 | 50.3 | 50.3 | 50.3 | 50.3 |
| Mantis-Idefics2 | 55.2 | 55.0 | 50.2 | 52.7 | 50.3 |
| Idefics3-8B | 54.0 | 54.0 | 52.8 | 50.3 | 50.3 |
| Phi-3-Vision | 50.5 | 54.0 | 56.2 | 55.3 | 57.7 |
| Phi-3.5-Vision | 55.5 | 52.2 | 52.0 | 50.2 | 50.8 |
| Phi-4-Multimodal | 53.3 | 52.5 | 52.2 | 50.8 | 52.7 |
| NVILA-Lite-2B | 50.8 | 54.0 | 50.5 | 50.5 | 50.3 |
| NVILA-Lite-8B | 53.7 | 51.0 | 53.3 | 54.3 | 52.2 |
| Pixtral-12B | 55.3 | 52.8 | 50.2 | 53.3 | 53.2 |

**MM-NIAH-Ret (T)**

| Model | 8k | 16k | 32k | 64k | 128k |
|---|---|---|---|---|---|
| GPT-4o | 94.3 | 97.0 | 88.3 | 77.6 | 70.3 |
| Claude-3.7-Sonnet | 97.0 | 94.0 | 91.9 | N/A | N/A |
| Gemini-2.0-Flash | 97.0 | 98.7 | 95.3 | 93.6 | 83.7 |
| Gemini-2.0-Flash-T | 94.7 | 99.3 | 96.5 | 95.5 | 89.2 |
| Gemini-2.5-Flash | 94.2 | 98.0 | 95.0 | 95.8 | 94.8 |
| Gemini-2.5-Pro | 97.3 | 100.0 | 98.3 | 98.7 | 96.7 |
| Qwen2-VL-2B | 97.0 | 92.2 | 85.7 | 54.2 | 29.7 |
| Qwen2-VL-7B | 97.5 | 96.7 | 93.5 | 75.0 | 37.3 |
| Qwen2-VL-72B | 98.0 | 99.3 | 96.7 | 90.3 | 48.7 |
| Qwen2.5-VL-3B | 98.3 | 95.8 | 89.3 | 64.5 | 2.5 |
| Qwen2.5-VL-7B | 94.0 | 91.7 | 88.2 | 63.7 | 12.7 |
| Qwen2.5-VL-32B | 95.3 | 97.3 | 95.0 | 88.3 | 48.0 |
| Qwen2.5-VL-72B | 94.3 | 96.3 | 95.3 | 81.3 | 48.7 |
| InternVL2-1B | 64.0 | 32.5 | 19.0 | 13.8 | 8.5 |
| InternVL2-2B | 83.0 | 62.7 | 39.2 | 31.3 | 15.0 |
| InternVL2-4B | 88.2 | 61.0 | 45.0 | 35.0 | 26.3 |
| InternVL2-8B | 96.8 | 90.3 | 74.8 | 67.7 | 46.7 |
| InternVL2.5-1B | 70.2 | 47.0 | 25.8 | 7.5 | 7.7 |
| InternVL2.5-2B | 89.0 | 74.3 | 48.0 | 38.3 | 17.3 |
| InternVL2.5-4B | 95.7 | 90.3 | 80.0 | 52.0 | 22.2 |
| InternVL2.5-8B | 92.3 | 86.7 | 71.0 | 67.3 | 44.7 |
| InternVL2.5-26B | 97.7 | 99.0 | 92.0 | 88.7 | 71.3 |
| InternVL3-1B | 70.0 | 59.0 | 28.0 | 26.0 | 2.7 |
| InternVL3-2B | 81.0 | 75.0 | 52.3 | 46.3 | 19.3 |
| InternVL3-8B | 96.7 | 94.0 | 86.0 | 75.3 | 46.3 |
| InternVL3-14B | 91.7 | 92.0 | 94.0 | 89.0 | 59.7 |
| InternVL3-38B | 97.3 | 99.7 | 96.7 | 90.0 | 74.7 |
| Ovis2-1B | 26.3 | 15.0 | 3.7 | 0.0 | 2.7 |
| Ovis2-2B | 70.0 | 40.0 | 23.3 | 19.7 | 16.7 |
| Ovis2-4B | 88.0 | 63.7 | 30.0 | 23.7 | 9.7 |
| Ovis2-8B | 96.0 | 95.0 | 83.3 | 52.3 | 24.0 |
| Ovis2-16B | 98.0 | 98.7 | 91.3 | 74.3 | 33.0 |
| Ovis2-34B | 98.3 | 99.0 | 98.3 | 88.3 | 40.7 |
| Gemma3-4B | 90.7 | 82.7 | 71.2 | 47.7 | 34.0 |
| Gemma3-12B | 97.5 | 93.8 | 89.7 | 79.8 | 60.7 |
| Gemma3-27B | 95.7 | 97.0 | 90.7 | 77.7 | 49.3 |
| Idefics2-8B | 22.8 | 18.8 | 0.0 | 0.0 | 0.0 |
| Idefics2-8B-C | 15.8 | 23.0 | 0.0 | 0.2 | 0.2 |
| Mantis-Idefics2 | 30.2 | 24.7 | 18.2 | 17.0 | 18.8 |
| Idefics3-8B | 86.7 | 74.7 | 60.3 | 49.7 | 29.0 |
| Phi-3-Vision | 82.7 | 52.3 | 33.2 | 21.0 | 7.0 |
| Phi-3.5-Vision | 84.7 | 75.2 | 42.8 | 24.3 | 23.0 |
| Phi-4-Multimodal | 93.5 | 90.5 | 80.2 | 58.7 | 30.7 |
| NVILA-Lite-2B | 93.3 | 76.3 | 50.8 | 33.2 | 24.3 |
| NVILA-Lite-8B | 97.8 | 97.7 | 87.0 | 52.8 | 27.0 |
| Pixtral-12B | 91.5 | 89.7 | 90.0 | 87.0 | 63.0 |

**MM-NIAH-Ret (I)**

| Model | 8k | 16k | 32k | 64k | 128k |
|---|---|---|---|---|---|
| GPT-4o | 98.7 | 94.7 | 95.3 | 91.3 | 81.0 |
| Claude-3.7-Sonnet | 44.0 | 50.8 | 33.7 | N/A | N/A |
| Gemini-2.0-Flash | 94.3 | 92.7 | 92.7 | 86.0 | 79.3 |
| Gemini-2.0-Flash-T | 96.5 | 93.5 | 91.8 | 86.0 | 81.9 |
| Gemini-2.5-Flash | 97.6 | 94.9 | 93.9 | 87.0 | 88.2 |
| Gemini-2.5-Pro | 100.0 | 99.0 | 98.7 | 97.3 | 95.7 |
| Qwen2-VL-2B | 52.3 | 47.5 | 42.8 | 32.8 | 33.8 |
| Qwen2-VL-7B | 84.2 | 77.0 | 64.3 | 48.8 | 32.0 |
| Qwen2-VL-72B | 77.0 | 76.3 | 68.3 | 52.0 | 37.3 |
| Qwen2.5-VL-3B | 69.7 | 73.3 | 55.2 | 43.2 | 0.2 |
| Qwen2.5-VL-7B | 80.8 | 75.7 | 62.0 | 47.8 | 29.8 |
| Qwen2.5-VL-32B | 79.0 | 78.7 | 69.0 | 63.7 | 50.0 |
| Qwen2.5-VL-72B | 88.3 | 78.7 | 75.7 | 66.3 | 51.7 |
| InternVL2-1B | 8.7 | 0.0 | 0.0 | 0.0 | 0.5 |
| InternVL2-2B | 0.8 | 0.8 | 0.3 | 0.0 | 8.3 |
| InternVL2-4B | 39.0 | 30.3 | 28.7 | 34.3 | 19.0 |
| InternVL2-8B | 66.2 | 47.5 | 20.7 | 22.3 | 22.7 |
| InternVL2.5-1B | 51.2 | 23.5 | 3.8 | 1.3 | 0.3 |
| InternVL2.5-2B | 59.0 | 45.0 | 26.3 | 23.3 | 3.0 |
| InternVL2.5-4B | 85.7 | 78.5 | 72.8 | 54.7 | 20.5 |
| InternVL2.5-8B | 78.3 | 73.3 | 67.0 | 42.7 | 29.7 |
| InternVL2.5-26B | 93.0 | 84.0 | 72.3 | 57.3 | 32.0 |
| InternVL3-1B | 49.0 | 47.0 | 45.0 | 37.0 | 37.7 |
| InternVL3-2B | 71.0 | 52.7 | 46.7 | 28.3 | 26.7 |
| InternVL3-8B | 82.3 | 75.7 | 64.3 | 60.0 | 58.3 |
| InternVL3-14B | 88.3 | 83.7 | 80.7 | 73.3 | 69.7 |
| InternVL3-38B | 97.7 | 96.3 | 94.3 | 90.7 | 78.7 |
| Ovis2-1B | 50.3 | 24.7 | 9.0 | 0.0 | 0.0 |
| Ovis2-2B | 74.3 | 62.0 | 54.7 | 28.7 | 40.7 |
| Ovis2-4B | 82.0 | 62.3 | 43.3 | 32.7 | 0.0 |
| Ovis2-8B | 83.3 | 86.7 | 75.7 | 54.3 | 52.0 |
| Ovis2-16B | 91.3 | 84.3 | 79.7 | 62.3 | 44.0 |
| Ovis2-34B | 94.7 | 89.3 | 81.0 | 68.0 | 44.0 |
| Gemma3-4B | 54.7 | 42.3 | 41.0 | 38.8 | 25.7 |
| Gemma3-12B | 85.3 | 75.0 | 64.3 | 51.3 | 37.0 |
| Gemma3-27B | 81.7 | 74.7 | 62.0 | 60.3 | 52.3 |
| Idefics2-8B | 17.0 | 13.7 | 17.5 | 0.0 | 0.0 |
| Idefics2-8B-C | 0.0 | 0.0 | 0.0 | 0.0 | 0.0 |
| Mantis-Idefics2 | 42.7 | 29.3 | 23.3 | 31.8 | 29.8 |
| Idefics3-8B | 3.0 | 0.3 | 0.0 | 20.3 | 21.0 |
| Phi-3-Vision | 31.0 | 25.7 | 27.5 | 0.7 | 0.0 |
| Phi-3.5-Vision | 31.2 | 20.2 | 22.8 | 18.3 | 17.7 |
| Phi-4-Multimodal | 36.7 | 35.5 | 29.3 | 27.7 | 24.0 |
| NVILA-Lite-2B | 27.0 | 29.0 | 28.8 | 27.8 | 21.8 |
| NVILA-Lite-8B | 55.0 | 48.7 | 36.3 | 32.8 | 28.3 |
| Pixtral-12B | 65.0 | 65.7 | 52.3 | 26.3 | 30.7 |

**MM-NIAH-Count (T)**

| Model | 8k | 16k | 32k | 64k | 128k |
|---|---|---|---|---|---|
| GPT-4o | 94.7 | 94.7 | 78.0 | 64.6 | 42.0 |
| Claude-3.7-Sonnet | 96.0 | 90.0 | 72.0 | N/A | N/A |
| Gemini-2.0-Flash | 78.7 | 76.7 | 64.0 | 50.0 | 34.0 |
| Gemini-2.0-Flash-T | 84.7 | 77.8 | 72.8 | 59.7 | 46.1 |
| Gemini-2.5-Flash | 95.3 | 96.5 | 94.6 | 90.3 | 87.7 |
| Gemini-2.5-Pro | 99.3 | 98.7 | 95.3 | 87.2 | 80.5 |
| Qwen2-VL-2B | 17.7 | 14.7 | 16.7 | 11.0 | 13.7 |
| Qwen2-VL-7B | 33.3 | 28.0 | 21.0 | 16.7 | 6.7 |
| Qwen2-VL-72B | 69.3 | 64.0 | 49.3 | 40.0 | 13.3 |
| Qwen2.5-VL-3B | 36.7 | 34.3 | 22.3 | 8.3 | 0.0 |
| Qwen2.5-VL-7B | 33.7 | 26.0 | 19.3 | 5.3 | 9.3 |
| Qwen2.5-VL-32B | 51.3 | 53.3 | 43.3 | 35.3 | 9.3 |
| Qwen2.5-VL-72B | 88.0 | 86.0 | 79.3 | 68.0 | 16.0 |
| InternVL2-1B | 12.0 | 7.7 | 6.0 | 8.3 | 4.0 |
| InternVL2-2B | 14.0 | 10.0 | 2.3 | 2.0 | 2.7 |
| InternVL2-4B | 21.0 | 13.7 | 6.7 | 3.3 | 9.3 |
| InternVL2-8B | 22.0 | 16.7 | 10.0 | 4.0 | 6.7 |
| InternVL2.5-1B | 15.7 | 13.3 | 6.7 | 4.7 | 1.3 |
| InternVL2.5-2B | 5.3 | 3.3 | 3.3 | 9.3 | 4.7 |
| InternVL2.5-4B | 28.7 | 22.3 | 14.7 | 6.7 | 1.7 |
| InternVL2.5-8B | 32.0 | 20.0 | 4.0 | 9.3 | 10.0 |
| InternVL2.5-26B | 54.7 | 46.7 | 22.7 | 16.0 | 10.0 |
| InternVL3-1B | 18.0 | 12.0 | 12.7 | 12.7 | 6.7 |
| InternVL3-2B | 16.7 | 17.3 | 18.7 | 14.7 | 12.0 |
| InternVL3-8B | 42.7 | 49.3 | 34.0 | 21.3 | 8.7 |
| InternVL3-14B | 68.7 | 61.3 | 37.3 | 33.3 | 14.7 |
| InternVL3-38B | 74.0 | 71.3 | 52.7 | 38.7 | 23.3 |
| Ovis2-1B | 3.3 | 5.3 | 7.3 | 0.0 | 0.0 |
| Ovis2-2B | 5.3 | 9.3 | 6.0 | 5.3 | 6.0 |
| Ovis2-4B | 20.7 | 16.7 | 11.3 | 6.0 | 0.0 |
| Ovis2-8B | 36.0 | 35.3 | 19.3 | 7.3 | 6.7 |
| Ovis2-16B | 42.7 | 46.7 | 27.3 | 22.0 | 9.3 |
| Ovis2-34B | 48.0 | 52.7 | 46.0 | 33.3 | 13.3 |
| Gemma3-4B | 21.3 | 15.3 | 7.3 | 3.7 | 3.0 |
| Gemma3-12B | 44.7 | 33.0 | 18.0 | 18.3 | 9.3 |
| Gemma3-27B | 64.0 | 57.3 | 43.3 | 28.7 | 12.7 |
| Idefics2-8B | 5.3 | 2.3 | 0.0 | 0.0 | 0.0 |
| Idefics2-8B-C | 4.3 | 1.3 | 0.0 | 0.0 | 0.0 |
| Mantis-Idefics2 | 3.7 | 2.0 | 1.7 | 4.3 | 3.0 |
| Idefics3-8B | 33.3 | 26.0 | 14.0 | 9.3 | 6.0 |
| Phi-3-Vision | 3.0 | 2.0 | 1.7 | 0.0 | 0.0 |
| Phi-3.5-Vision | 7.0 | 4.0 | 1.3 | 8.0 | 0.7 |
| Phi-4-Multimodal | 18.7 | 15.7 | 7.7 | 1.3 | 4.0 |
| NVILA-Lite-2B | 18.0 | 16.7 | 13.7 | 4.7 | 5.0 |
| NVILA-Lite-8B | 30.7 | 21.7 | 17.3 | 7.7 | 0.0 |
| Pixtral-12B | 42.7 | 37.3 | 26.0 | 20.0 | 6.7 |

**MM-NIAH-Count (I)**

| Model | 8k | 16k | 32k | 64k | 128k |
|---|---|---|---|---|---|
| GPT-4o | 67.3 | 36.0 | 10.0 | 29.3 | 4.7 |
| Claude-3.7-Sonnet | 40.0 | 33.3 | 23.2 | N/A | N/A |
| Gemini-2.0-Flash | 62.7 | 60.7 | 48.7 | 34.0 | 27.3 |
| Gemini-2.0-Flash-T | 63.3 | 66.0 | 58.5 | 46.5 | 29.1 |
| Gemini-2.5-Flash | 71.3 | 71.3 | 75.3 | 63.9 | 45.0 |
| Gemini-2.5-Pro | 47.3 | 41.3 | 40.0 | 34.7 | 31.3 |
| Qwen2-VL-2B | 35.0 | 40.7 | 36.0 | 26.0 | 34.3 |
| Qwen2-VL-7B | 29.0 | 19.0 | 28.3 | 41.3 | 15.0 |
| Qwen2-VL-72B | 22.7 | 20.7 | 13.3 | 20.0 | 19.3 |
| Qwen2.5-VL-3B | 28.0 | 21.0 | 2.3 | 9.3 | 0.0 |
| Qwen2.5-VL-7B | 21.7 | 14.0 | 7.0 | 3.3 | 16.3 |
| Qwen2.5-VL-32B | 11.3 | 23.3 | 33.3 | 20.7 | 14.7 |
| Qwen2.5-VL-72B | 23.3 | 23.3 | 37.3 | 38.0 | 27.3 |
| InternVL2-1B | 6.7 | 18.0 | 30.7 | 26.7 | 8.3 |
| InternVL2-2B | 6.3 | 5.7 | 3.0 | 5.3 | 12.0 |
| InternVL2-4B | 3.3 | 1.3 | 1.0 | 0.0 | 0.0 |
| InternVL2-8B | 15.3 | 24.3 | 25.7 | 28.0 | 13.3 |
| InternVL2.5-1B | 29.0 | 32.7 | 28.3 | 7.7 | 2.7 |
| InternVL2.5-2B | 2.7 | 11.3 | 28.7 | 16.0 | 6.7 |
| InternVL2.5-4B | 4.3 | 4.7 | 5.7 | 19.3 | 2.3 |
| InternVL2.5-8B | 38.7 | 27.3 | 31.3 | 32.0 | 21.3 |
| InternVL2.5-26B | 14.7 | 14.7 | 6.0 | 10.7 | 2.7 |
| InternVL3-1B | 35.3 | 37.3 | 36.0 | 21.3 | 6.7 |
| InternVL3-2B | 29.3 | 28.0 | 13.3 | 16.0 | 14.0 |
| InternVL3-8B | 21.3 | 14.0 | 9.3 | 7.3 | 6.0 |
| InternVL3-14B | 35.3 | 22.0 | 8.7 | 18.0 | 17.3 |
| InternVL3-38B | 29.3 | 17.3 | 8.7 | 8.7 | 5.3 |
| Ovis2-1B | 16.7 | 22.0 | 3.3 | 0.7 | 0.0 |
| Ovis2-2B | 4.0 | 2.7 | 3.3 | 1.3 | 16.7 |
| Ovis2-4B | 0.0 | 0.0 | 0.0 | 2.0 | 0.0 |
| Ovis2-8B | 42.7 | 37.3 | 25.3 | 10.7 | 4.7 |
| Ovis2-16B | 52.0 | 53.3 | 40.7 | 39.3 | 31.3 |
| Ovis2-34B | 36.0 | 14.7 | 26.7 | 35.3 | 31.3 |
| Gemma3-4B | 20.7 | 14.3 | 15.7 | 5.0 | 0.0 |
| Gemma3-12B | 35.0 | 39.7 | 36.7 | 35.7 | 30.0 |
| Gemma3-27B | 30.7 | 30.0 | 24.7 | 34.7 | 39.3 |
| Idefics2-8B | 8.7 | 8.7 | 5.7 | 0.0 | 0.0 |
| Idefics2-8B-C | 4.3 | 7.7 | 6.3 | 0.3 | 0.0 |
| Mantis-Idefics2 | 4.7 | 1.3 | 5.0 | 4.0 | 1.0 |
| Idefics3-8B | 36.7 | 40.7 | 38.7 | 38.0 | 36.0 |
| Phi-3-Vision | 0.0 | 1.7 | 0.3 | 0.0 | 0.0 |
| Phi-3.5-Vision | 0.0 | 1.7 | 1.3 | 0.7 | 0.7 |
| Phi-4-Multimodal | 6.0 | 0.3 | 0.0 | 0.0 | 4.0 |
| NVILA-Lite-2B | 25.0 | 30.3 | 24.7 | 20.7 | 11.7 |
| NVILA-Lite-8B | 8.7 | 2.3 | 2.3 | 1.0 | 2.0 |
| Pixtral-12B | 27.0 | 20.7 | 16.7 | 16.7 | 6.0 |

**MM-NIAH-Reason (T)**

| Model | 8k | 16k | 32k | 64k | 128k |
|---|---|---|---|---|---|
| GPT-4o | 81.3 | 81.3 | 80.0 | 66.7 | 58.7 |
| Claude-3.7-Sonnet | 96.6 | 90.6 | 73.0 | N/A | N/A |
| Gemini-2.0-Flash | 88.7 | 84.0 | 76.9 | 72.3 | 72.0 |
| Gemini-2.0-Flash-T | 91.3 | 92.7 | 86.4 | 80.3 | 76.9 |
| Gemini-2.5-Flash | 95.3 | 90.7 | 89.1 | 87.8 | 85.7 |
| Gemini-2.5-Pro | 96.7 | 95.3 | 92.5 | 93.3 | 90.0 |
| Qwen2-VL-2B | 36.3 | 38.0 | 26.7 | 17.0 | 9.7 |
| Qwen2-VL-7B | 49.7 | 50.7 | 46.3 | 35.0 | 15.0 |
| Qwen2-VL-72B | 79.3 | 80.7 | 69.3 | 56.0 | 28.0 |
| Qwen2.5-VL-3B | 47.7 | 39.3 | 34.7 | 16.3 | 0.3 |
| Qwen2.5-VL-7B | 54.7 | 50.0 | 43.0 | 24.0 | 0.7 |
| Qwen2.5-VL-32B | 77.3 | 72.7 | 65.3 | 56.7 | 37.3 |
| Qwen2.5-VL-72B | 84.7 | 77.3 | 71.3 | 58.0 | 39.3 |
| InternVL2-1B | 11.0 | 9.7 | 6.7 | 2.0 | 0.7 |
| InternVL2-2B | 26.0 | 19.7 | 10.3 | 10.0 | 4.7 |
| InternVL2-4B | 40.3 | 33.3 | 18.7 | 13.3 | 11.3 |
| InternVL2-8B | 48.7 | 41.3 | 31.3 | 14.7 | 6.7 |
| InternVL2.5-1B | 11.7 | 14.3 | 8.3 | 4.3 | 4.0 |
| InternVL2.5-2B | 33.3 | 28.0 | 10.7 | 11.3 | 4.0 |
| InternVL2.5-4B | 49.0 | 38.7 | 28.3 | 16.0 | 9.7 |
| InternVL2.5-8B | 54.0 | 37.3 | 35.3 | 22.7 | 12.7 |
| InternVL2.5-26B | 61.3 | 52.0 | 48.0 | 36.7 | 28.0 |
| InternVL3-1B | 24.7 | 20.7 | 10.7 | 6.7 | 0.7 |
| InternVL3-2B | 40.7 | 30.0 | 20.0 | 7.3 | 3.3 |
| InternVL3-8B | 65.3 | 50.0 | 40.7 | 28.7 | 26.0 |
| InternVL3-14B | 79.3 | 72.0 | 52.7 | 46.7 | 23.3 |
| InternVL3-38B | 84.0 | 72.7 | 66.0 | 46.0 | 41.3 |
| Ovis2-1B | 68.0 | 34.7 | 6.0 | 4.0 | 1.3 |
| Ovis2-2B | 24.7 | 18.0 | 9.3 | 3.3 | 0.0 |
| Ovis2-4B | 42.0 | 24.0 | 22.7 | 16.0 | 2.0 |
| Ovis2-8B | 57.3 | 37.3 | 39.3 | 22.0 | 12.0 |
| Ovis2-16B | 75.3 | 63.3 | 48.0 | 36.0 | 18.7 |
| Ovis2-34B | 78.0 | 75.3 | 68.7 | 46.7 | 26.7 |
| Gemma3-4B | 44.3 | 43.7 | 35.7 | 23.3 | 13.0 |
| Gemma3-12B | 62.7 | 51.7 | 47.0 | 35.0 | 20.7 |
| Gemma3-27B | 81.3 | 66.7 | 57.3 | 32.7 | 24.0 |
| Idefics2-8B | 8.7 | 7.7 | 0.0 | 0.0 | 0.0 |
| Idefics2-8B-C | 5.3 | 6.3 | 0.0 | 0.0 | 0.0 |
| Mantis-Idefics2 | 18.3 | 7.3 | 6.7 | 3.0 | 1.3 |
| Idefics3-8B | 47.3 | 29.3 | 28.7 | 13.3 | 11.3 |
| Phi-3-Vision | 29.0 | 21.7 | 10.3 | 7.3 | 0.7 |
| Phi-3.5-Vision | 45.0 | 35.7 | 15.3 | 7.3 | 6.0 |
| Phi-4-Multimodal | 42.3 | 35.3 | 26.3 | 14.0 | 10.0 |
| NVILA-Lite-2B | 34.3 | 17.0 | 16.3 | 8.7 | 5.0 |
| NVILA-Lite-8B | 55.3 | 43.3 | 38.7 | 22.0 | 4.7 |
| Pixtral-12B | 58.3 | 54.7 | 43.3 | 41.3 | 26.7 |

**MM-NIAH-Reason (I)**

| Model | 8k | 16k | 32k | 64k | 128k |
|---|---|---|---|---|---|
| GPT-4o | 85.3 | 75.7 | 73.7 | 71.0 | 85.0 |
| Claude-3.7-Sonnet | 41.7 | 40.6 | 42.4 | N/A | N/A |
| Gemini-2.0-Flash | 83.0 | 72.3 | 73.0 | 75.3 | 77.3 |
| Gemini-2.0-Flash-T | 84.0 | 79.2 | 79.6 | 75.2 | 73.2 |
| Gemini-2.5-Flash | 82.7 | 77.6 | 83.7 | 81.0 | 80.6 |
| Gemini-2.5-Pro | 82.7 | 83.0 | 79.7 | 78.3 | 79.3 |
| Qwen2-VL-2B | 54.0 | 51.0 | 47.2 | 45.0 | 44.5 |
| Qwen2-VL-7B | 55.7 | 50.7 | 44.7 | 37.7 | 36.2 |
| Qwen2-VL-72B | 46.0 | 38.3 | 43.7 | 31.7 | 32.3 |
| Qwen2.5-VL-3B | 49.2 | 39.2 | 33.8 | 35.0 | 1.2 |
| Qwen2.5-VL-7B | 56.8 | 50.5 | 44.8 | 36.0 | 48.8 |
| Qwen2.5-VL-32B | 60.0 | 56.7 | 55.3 | 51.0 | 46.7 |
| Qwen2.5-VL-72B | 43.7 | 36.3 | 35.7 | 30.0 | 31.0 |
| InternVL2-1B | 1.5 | 7.0 | 0.5 | 0.7 | 0.0 |
| InternVL2-2B | 7.8 | 4.8 | 3.7 | 27.0 | 20.0 |
| InternVL2-4B | 48.0 | 25.2 | 24.7 | 33.3 | 29.3 |
| InternVL2-8B | 86.5 | 52.3 | 25.8 | 17.3 | 19.0 |
| InternVL2.5-1B | 88.3 | 22.2 | 16.5 | 1.2 | 0.3 |
| InternVL2.5-2B | 43.3 | 30.3 | 48.0 | 63.0 | 53.3 |
| InternVL2.5-4B | 98.3 | 98.3 | 93.0 | 83.0 | 32.5 |
| InternVL2.5-8B | 93.3 | 94.7 | 95.0 | 79.3 | 54.7 |
| InternVL2.5-26B | 95.0 | 95.0 | 90.3 | 83.3 | 85.0 |
| InternVL3-1B | 57.0 | 84.7 | 91.0 | 85.3 | 45.3 |
| InternVL3-2B | 90.3 | 71.7 | 57.0 | 43.3 | 28.3 |
| InternVL3-8B | 85.0 | 88.7 | 82.0 | 90.0 | 74.0 |
| InternVL3-14B | 81.3 | 77.0 | 84.3 | 92.3 | 93.3 |
| InternVL3-38B | 92.0 | 88.7 | 92.7 | 89.0 | 93.7 |
| Ovis2-1B | 68.0 | 34.7 | 6.0 | 0.0 | 0.0 |
| Ovis2-2B | 74.7 | 85.3 | 67.7 | 33.3 | 10.3 |
| Ovis2-4B | 53.7 | 40.7 | 33.7 | 37.0 | 0.0 |
| Ovis2-8B | 71.3 | 69.0 | 76.0 | 54.3 | 47.7 |
| Ovis2-16B | 79.3 | 60.0 | 59.7 | 49.7 | 60.3 |
| Ovis2-34B | 61.3 | 48.7 | 39.3 | 42.7 | 39.0 |
| Gemma3-4B | 39.0 | 39.5 | 38.8 | 39.7 | 44.5 |
| Gemma3-12B | 51.2 | 52.3 | 51.3 | 49.5 | 46.8 |
| Gemma3-27B | 65.0 | 52.0 | 51.7 | 51.7 | 45.0 |
| Idefics2-8B | 36.0 | 28.0 | 26.3 | 0.0 | 0.0 |
| Idefics2-8B-C | 0.7 | 0.8 | 0.3 | 0.0 | 0.0 |
| Mantis-Idefics2 | 41.7 | 39.5 | 39.5 | 36.3 | 41.7 |
| Idefics3-8B | 55.7 | 62.7 | 65.3 | 64.3 | 59.0 |
| Phi-3-Vision | 53.0 | 56.5 | 54.3 | 40.0 | 0.0 |
| Phi-3.5-Vision | 51.2 | 43.7 | 48.8 | 38.7 | 56.3 |
| Phi-4-Multimodal | 60.5 | 56.8 | 57.3 | 60.0 | 68.7 |
| NVILA-Lite-2B | 54.8 | 54.5 | 52.0 | 49.0 | 29.7 |
| NVILA-Lite-8B | 60.8 | 46.2 | 43.2 | 39.8 | 22.3 |
| Pixtral-12B | 54.7 | 66.3 | 63.3 | 50.3 | 59.3 |

Figure 25: Results of 46 name on the category NIAH at various lengths.

## Stanford Cars | Food101 | SUN397 | Inat2021

| Model | SC 8k | SC 16k | SC 32k | SC 64k | SC 128k | F 8k | F 16k | F 32k | F 64k | F 128k | S 8k | S 16k | S 32k | S 64k | S 128k | I 8k | I 16k | I 32k | I 64k | I 128k |
|---|---|---|---|---|---|---|---|---|---|---|---|---|---|---|---|---|---|---|---|---|
| GPT-4o | 100.0 | 100.0 | 100.0 | 100.0 | 99.0 | 100.0 | 97.0 | 99.0 | 88.7 | 89.7 | 98.0 | 100.0 | 97.0 | 90.0 | 82.0 | 98.0 | 96.0 | 88.0 | 91.0 | 83.0 |
| Claude-3.7-Sonnet | 100.0 | 100.0 | N/A | N/A | N/A | 100.0 | 91.0 | N/A | N/A | N/A | 96.0 | 98.8 | N/A | N/A | N/A | 92.0 | 87.0 | N/A | N/A | N/A |
| Gemini-2.0-Flash | 100.0 | 100.0 | 100.0 | 100.0 | 96.0 | 98.0 | 97.0 | 99.0 | 93.0 | 87.0 | 98.0 | 99.0 | 99.0 | 88.0 | 85.0 | 100.0 | 95.0 | 92.0 | 94.0 | 82.0 |
| Gemini-2.0-Flash-T | 100.0 | 100.0 | 100.0 | 100.0 | 99.0 | 100.0 | 97.0 | 98.0 | 93.0 | 87.0 | 98.0 | 100.0 | 97.0 | 85.0 | 86.0 | 100.0 | 94.0 | 90.0 | 92.0 | 81.0 |
| Gemini-2.5-Flash | 100.0 | 100.0 | 100.0 | 100.0 | 99.0 | 100.0 | 97.0 | 98.0 | 97.0 | 91.0 | 96.0 | 99.0 | 98.0 | 87.0 | 84.0 | 98.0 | 98.0 | 90.0 | 92.0 | 78.0 |
| Gemini-2.5-Pro | 100.0 | 100.0 | 100.0 | 100.0 | 100.0 | 100.0 | 98.0 | 99.0 | 97.0 | 94.0 | 98.0 | 100.0 | 96.0 | 86.0 | 90.0 | 100.0 | 96.0 | 94.0 | 97.0 | 93.0 |
| Qwen2-VL-2B | 77.4 | 44.9 | 2.0 | 0.0 | 1.0 | 86.0 | 63.0 | 22.0 | 16.0 | 3.0 | 96.0 | 97.0 | 67.0 | 11.0 | 6.0 | 84.0 | 41.0 | 10.0 | 4.0 | 2.0 |
| Qwen2-VL-7B | 90.6 | 36.7 | 9.0 | 3.0 | 5.0 | 98.0 | 77.0 | 36.0 | 13.0 | 7.0 | 90.0 | 62.0 | 43.0 | 12.0 | 10.0 | 82.0 | 51.0 | 25.0 | 8.0 | 9.0 |
| Qwen2-VL-72B | 100.0 | 100.0 | 98.0 | 92.0 | 92.0 | 100.0 | 91.0 | 93.0 | 82.0 | 79.0 | 98.0 | 99.0 | 96.0 | 83.0 | 87.0 | 96.0 | 88.0 | 77.0 | 66.0 | 65.0 |
| Qwen2.5-VL-3B | 98.1 | 74.5 | 22.0 | 5.0 | 1.0 | 98.0 | 59.0 | 11.0 | 6.0 | 10.0 | 96.0 | 74.0 | 23.0 | 5.0 | 14.0 | 88.0 | 72.0 | 22.0 | 14.0 | 11.0 |
| Qwen2.5-VL-7B | 96.2 | 99.0 | 68.0 | 55.0 | 34.0 | 98.0 | 89.0 | 82.0 | 52.0 | 52.0 | 100.0 | 99.0 | 94.0 | 75.0 | 66.0 | 88.0 | 79.0 | 70.0 | 47.0 | 33.0 |
| Qwen2.5-VL-32B | 100.0 | 99.0 | 74.0 | 24.0 | 17.0 | 100.0 | 91.0 | 94.0 | 72.0 | 67.0 | 98.0 | 96.0 | 92.0 | 85.0 | 65.0 | 92.0 | 81.0 | 48.0 | 24.0 | 16.0 |
| Qwen2.5-VL-72B | 100.0 | 100.0 | 98.0 | 62.0 | 61.0 | 100.0 | 95.0 | 94.0 | 80.0 | 72.0 | 98.0 | 98.0 | 95.0 | 78.0 | 79.0 | 96.0 | 89.0 | 84.0 | 77.0 | 80.0 |
| InternVL2-1B | 0.0 | 1.0 | 1.0 | 0.0 | 2.0 | 2.0 | 7.0 | 1.0 | 0.0 | 0.0 | 2.0 | 3.0 | 0.0 | 0.0 | 1.0 | 0.0 | 1.0 | 0.0 | 0.0 | 1.0 |
| InternVL2-2B | 1.9 | 3.1 | 0.0 | 0.0 | 0.0 | 8.0 | 5.0 | 5.0 | 2.0 | 3.0 | 0.0 | 0.0 | 4.0 | 0.0 | 0.0 | 4.0 | 4.0 | 0.0 | 2.0 | 2.0 |
| InternVL2-4B | 49.1 | 11.2 | 1.0 | 0.0 | 1.0 | 86.0 | 26.0 | 2.0 | 0.0 | 0.0 | 94.0 | 49.0 | 0.0 | 0.0 | 0.0 | 46.0 | 24.0 | 1.0 | 0.0 | 2.0 |
| InternVL2-8B | 26.4 | 7.1 | 0.0 | 1.0 | 0.0 | 36.0 | 10.0 | 7.0 | 0.0 | 0.0 | 22.0 | 3.0 | 0.0 | 0.0 | 0.0 | 34.0 | 7.0 | 3.0 | 1.0 | 0.0 |
| InternVL2.5-1B | 0.0 | 0.0 | 0.0 | 0.0 | 1.0 | 18.0 | 0.0 | 0.0 | 0.0 | 0.0 | 0.0 | 0.0 | 0.0 | 0.0 | 5.0 | 2.0 | 1.0 | 1.0 | 0.0 | 4.0 |
| InternVL2.5-2B | 5.7 | 0.0 | 0.0 | 0.0 | 0.0 | 10.0 | 0.0 | 0.0 | 0.0 | 0.0 | 0.0 | 3.0 | 0.0 | 0.0 | 0.0 | 2.0 | 3.0 | 0.0 | 1.0 | 0.0 |
| InternVL2.5-4B | 92.5 | 55.1 | 23.0 | 1.0 | 0.0 | 100.0 | 83.0 | 65.0 | 20.0 | 8.0 | 100.0 | 93.0 | 64.0 | 5.0 | 1.0 | 84.0 | 68.0 | 43.0 | 8.0 | 2.0 |
| InternVL2.5-8B | 94.3 | 41.8 | 21.0 | 20.0 | 2.0 | 100.0 | 69.0 | 57.0 | 25.0 | 1.0 | 100.0 | 61.0 | 58.0 | 1.0 | 1.0 | 92.0 | 62.0 | 47.0 | 8.0 | 0.0 |
| InternVL2.5-26B | 100.0 | 92.9 | 81.0 | 68.0 | 37.0 | 100.0 | 90.0 | 86.0 | 73.0 | 58.0 | 96.0 | 86.0 | 95.0 | 78.0 | 67.0 | 98.0 | 88.0 | 78.0 | 71.0 | 54.0 |
| InternVL3-1B | 28.3 | 10.2 | 5.0 | 1.0 | 1.0 | 22.0 | 9.0 | 5.0 | 2.0 | 0.0 | 20.0 | 3.0 | 2.0 | 2.0 | 1.0 | 22.0 | 11.0 | 2.0 | 2.0 | 0.0 |
| InternVL3-2B | 62.3 | 23.5 | 8.0 | 4.0 | 1.0 | 94.0 | 45.0 | 8.0 | 3.0 | 2.0 | 82.0 | 41.0 | 9.0 | 4.0 | 1.0 | 80.0 | 24.0 | 7.0 | 4.0 | 0.0 |
| InternVL3-8B | 94.3 | 84.7 | 55.0 | 48.0 | 5.0 | 100.0 | 86.0 | 81.0 | 64.0 | 20.0 | 100.0 | 96.0 | 94.0 | 77.0 | 6.0 | 96.0 | 82.0 | 70.0 | 58.0 | 3.0 |
| InternVL3-14B | 98.1 | 84.7 | 66.0 | 49.0 | 25.0 | 100.0 | 88.0 | 86.0 | 68.0 | 63.0 | 96.0 | 98.0 | 95.0 | 86.0 | 77.0 | 92.0 | 80.0 | 73.0 | 60.0 | 47.0 |
| InternVL3-38B | 100.0 | 100.0 | 84.0 | 71.0 | 42.0 | 100.0 | 94.0 | 90.0 | 82.0 | 77.0 | 100.0 | 99.0 | 98.0 | 86.0 | 79.0 | 98.0 | 87.0 | 82.0 | 71.0 | 63.0 |
| Ovis2-1B | 26.4 | 5.1 | 8.0 | 4.0 | 1.0 | 10.0 | 12.0 | 8.0 | 0.0 | 0.0 | 28.0 | 8.0 | 7.0 | 1.0 | 1.0 | 24.0 | 4.0 | 1.0 | 3.0 | 1.0 |
| Ovis2-2B | 3.8 | 2.0 | 1.0 | 0.0 | 0.0 | 2.0 | 1.0 | 0.0 | 0.0 | 0.0 | 0.0 | 0.0 | 0.0 | 0.0 | 0.0 | 8.0 | 0.0 | 0.0 | 0.0 | 0.0 |
| Ovis2-4B | 73.6 | 10.2 | 6.0 | 2.0 | 1.0 | 30.0 | 11.0 | 5.0 | 0.0 | 3.0 | 78.0 | 18.0 | 10.0 | 0.0 | 1.0 | 60.0 | 19.0 | 4.0 | 2.0 | 0.0 |
| Ovis2-8B | 98.1 | 38.8 | 7.0 | 1.0 | 0.0 | 92.0 | 11.0 | 9.0 | 11.0 | 0.0 | 98.0 | 91.0 | 12.0 | 2.0 | 2.0 | 90.0 | 37.0 | 3.0 | 2.0 | 2.0 |
| Ovis2-16B | 94.3 | 100.0 | 72.0 | 56.0 | 18.0 | 100.0 | 92.0 | 78.0 | 80.0 | 59.0 | 100.0 | 99.0 | 89.0 | 76.0 | 46.0 | 92.0 | 74.0 | 54.0 | 52.0 | 23.0 |
| Ovis2-34B | 100.0 | 98.0 | 76.0 | 72.0 | 53.0 | 100.0 | 90.0 | 83.0 | 76.0 | 72.0 | 100.0 | 100.0 | 93.0 | 78.0 | 82.0 | 94.0 | 70.0 | 65.0 | 58.0 | 54.0 |
| Gemma3-4B | 96.2 | 94.9 | 50.0 | 21.0 | 8.0 | 100.0 | 83.0 | 80.0 | 45.0 | 10.0 | 100.0 | 96.0 | 84.0 | 33.0 | 7.0 | 94.0 | 68.0 | 56.0 | 33.0 | 18.0 |
| Gemma3-12B | 100.0 | 100.0 | 99.0 | 90.0 | 63.0 | 100.0 | 96.0 | 95.0 | 87.0 | 74.0 | 96.0 | 99.0 | 95.0 | 85.0 | 71.0 | 100.0 | 91.0 | 84.0 | 67.0 | 28.0 |
| Gemma3-27B | 100.0 | 100.0 | 100.0 | 94.0 | 88.0 | 98.0 | 95.0 | 96.0 | 87.0 | 83.0 | 98.0 | 97.0 | 94.0 | 82.0 | 80.0 | 94.0 | 87.0 | 84.0 | 72.0 | 44.0 |
| Idefics2-8B | 18.9 | 10.2 | 3.0 | 0.0 | 2.0 | 18.0 | 14.0 | 2.0 | 3.0 | 2.0 | 8.0 | 9.0 | 1.0 | 3.0 | 0.0 | 20.0 | 12.0 | 0.0 | 4.0 | 2.0 |
| Idefics2-8B-C | 15.1 | 9.2 | 5.0 | 2.0 | 1.0 | 6.0 | 3.0 | 9.0 | 3.0 | 0.0 | 22.0 | 11.0 | 12.0 | 1.0 | 1.0 | 10.0 | 3.0 | 3.0 | 1.0 | 2.0 |
| Mantis-Idefics2 | 28.3 | 4.1 | 8.0 | 0.0 | 5.0 | 84.0 | 51.0 | 4.0 | 3.0 | 0.0 | 70.0 | 20.0 | 5.0 | 6.0 | 3.0 | 48.0 | 34.0 | 4.0 | 1.0 | 1.0 |
| Idefics3-8B | 26.4 | 5.1 | 3.0 | 2.0 | 0.0 | 30.0 | 15.0 | 2.0 | 1.0 | 5.0 | 20.0 | 18.0 | 4.0 | 0.0 | 0.0 | 26.0 | 11.0 | 9.0 | 0.0 | 1.0 |
| Phi-3-Vision | 58.5 | 18.4 | 8.0 | 1.0 | 0.0 | 60.0 | 20.0 | 9.0 | 2.0 | 0.0 | 60.0 | 34.0 | 3.0 | 0.0 | 1.0 | 38.0 | 12.0 | 2.0 | 8.0 | 2.0 |
| Phi-3.5-Vision | 69.8 | 41.8 | 6.0 | 3.0 | 1.0 | 94.0 | 67.0 | 12.0 | 6.0 | 3.0 | 98.0 | 87.0 | 5.0 | 2.0 | 3.0 | 76.0 | 46.0 | 6.0 | 2.0 | 1.0 |
| Phi-4-Multimodal | 77.4 | 41.8 | 12.0 | 2.0 | 4.0 | 100.0 | 57.0 | 19.0 | 3.0 | 4.0 | 86.0 | 46.0 | 8.0 | 5.0 | 1.0 | 66.0 | 25.0 | 9.0 | 1.0 | 0.0 |
| NVILA-Lite-2B | 60.4 | 29.6 | 5.0 | 0.0 | 0.0 | 68.0 | 17.0 | 11.0 | 0.0 | 0.0 | 48.0 | 24.0 | 11.0 | 0.0 | 1.0 | 42.0 | 12.0 | 4.0 | 1.0 | 2.0 |
| NVILA-Lite-8B | 94.3 | 72.4 | 30.0 | 16.0 | 0.0 | 100.0 | 75.0 | 58.0 | 25.0 | 9.0 | 94.0 | 95.0 | 70.0 | 24.0 | 0.0 | 84.0 | 52.0 | 30.0 | 17.0 | 2.0 |
| Pixtral-12B | 98.1 | 100.0 | 96.0 | 52.0 | 51.0 | 98.0 | 90.0 | 89.0 | 58.0 | 48.0 | 100.0 | 97.0 | 96.0 | 61.0 | 70.0 | 84.0 | 73.0 | 63.0 | 42.0 | 30.0 |

Figure 26: Results of 46 models on the category ICL at various lengths.

Figure 27 data — Results of 46 models on the category DocVQA at various lengths.

| Model | MMLongBench-Doc 8k | 16k | 32k | 64k | 128k | LongDocURL 8k | 16k | 32k | 64k | 128k | SlideVQA 8k | 16k | 32k | 64k | 128k |
|---|---|---|---|---|---|---|---|---|---|---|---|---|---|---|---|
| GPT-4o | 54.1 | 66.1 | 50.8 | 47.1 | 43.0 | 72.3 | 67.9 | 75.9 | 68.2 | 63.2 | 77.0 | 77.5 | 75.0 | 73.5 | 71.4 |
| Claude-3.7-Sonnet | 51.8 | 39.2 | 35.1 | 33.0 | N/A | 61.0 | 56.4 | 63.4 | 55.2 | N/A | 57.4 | 60.5 | 30.7 | 57.4 | N/A |
| Gemini-2.0-Flash | 47.3 | 47.9 | 47.1 | 39.7 | 38.8 | 59.6 | 52.0 | 56.8 | 50.4 | 56.2 | 69.0 | 66.5 | 74.1 | 71.1 | 65.7 |
| Gemini-2.0-Flash-T | 63.5 | 61.1 | 57.0 | 52.1 | 53.2 | 63.3 | 62.0 | 69.4 | 65.1 | 60.5 | 77.5 | 83.1 | 83.2 | 75.5 | 77.5 |
| Gemini-2.5-Flash | 53.8 | 55.6 | 57.4 | 51.3 | 50.9 | 66.2 | 61.2 | 68.5 | 59.6 | 52.0 | 82.6 | 83.9 | 79.8 | 76.7 | 74.9 |
| Gemini-2.5-Pro | 60.2 | 62.2 | 55.9 | 56.0 | 60.0 | 68.9 | 63.5 | 69.2 | 66.2 | 65.4 | 85.3 | 84.3 | 87.3 | 85.3 | 85.6 |
| Qwen2-VL-2B | 28.3 | 21.5 | 21.7 | 21.6 | 13.6 | 52.5 | 38.6 | 49.3 | 31.2 | 25.9 | 48.7 | 58.7 | 48.8 | 45.1 | 40.7 |
| Qwen2-VL-7B | 43.9 | 37.9 | 32.6 | 27.4 | 23.9 | 60.4 | 53.3 | 65.0 | 48.5 | 49.1 | 68.0 | 75.0 | 74.1 | 73.3 | 69.5 |
| Qwen2-VL-72B | 59.8 | 50.8 | 45.8 | 43.7 | 32.1 | 69.5 | 63.7 | 71.2 | 64.4 | 55.6 | 78.2 | 82.7 | 82.1 | 74.7 | 73.7 |
| Qwen2.5-VL-3B | 40.6 | 41.3 | 30.9 | 22.2 | 14.7 | 57.8 | 45.1 | 51.4 | 41.9 | 41.7 | 68.0 | 69.7 | 72.8 | 71.1 | 50.4 |
| Qwen2.5-VL-7B | 52.7 | 50.0 | 42.8 | 35.8 | 17.1 | 61.9 | 52.8 | 62.1 | 48.0 | 46.6 | 67.6 | 68.5 | 66.8 | 68.3 | 56.9 |
| Qwen2.5-VL-32B | 58.0 | 58.2 | 48.5 | 42.1 | 31.9 | 67.5 | 62.2 | 70.7 | 57.0 | 54.5 | 77.8 | 77.7 | 78.2 | 76.1 | 74.3 |
| Qwen2.5-VL-72B | 65.6 | 58.7 | 55.5 | 42.1 | 28.8 | 65.8 | 62.6 | 65.0 | 52.4 | 43.3 | 82.8 | 81.3 | 77.0 | 77.5 | 73.9 |
| InternVL2-1B | 11.3 | 6.9 | 5.3 | 1.8 | 3.4 | 15.6 | 14.5 | 13.5 | 5.0 | 3.5 | 19.9 | 10.7 | 9.0 | 7.9 | 5.0 |
| InternVL2-2B | 15.3 | 12.7 | 9.1 | 7.0 | 2.7 | 24.5 | 19.1 | 19.1 | 12.8 | 4.4 | 25.3 | 22.2 | 24.4 | 22.5 | 8.3 |
| InternVL2-4B | 13.3 | 14.3 | 13.4 | 4.2 | 4.2 | 17.9 | 25.0 | 23.7 | 14.9 | 8.8 | 30.5 | 29.1 | 25.7 | 20.8 | 1.0 |
| InternVL2-8B | 37.1 | 33.7 | 18.7 | 11.3 | 10.5 | 37.1 | 30.2 | 34.6 | 22.2 | 19.9 | 53.5 | 39.9 | 36.8 | 30.0 | 24.6 |
| InternVL2.5-1B | 17.7 | 14.8 | 8.7 | 6.8 | 3.1 | 24.1 | 19.4 | 18.9 | 11.4 | 5.0 | 30.9 | 19.5 | 9.9 | 9.3 | 4.1 |
| InternVL2.5-2B | 23.0 | 23.3 | 12.3 | 13.2 | 3.8 | 34.4 | 32.7 | 27.7 | 16.0 | 8.9 | 43.0 | 38.9 | 32.8 | 20.7 | 11.6 |
| InternVL2.5-4B | 44.2 | 38.0 | 28.8 | 22.9 | 11.3 | 48.1 | 41.4 | 40.3 | 37.3 | 20.1 | 62.1 | 54.2 | 53.9 | 41.6 | 0.3 |
| InternVL2.5-8B | 40.4 | 42.6 | 35.5 | 29.4 | 17.1 | 54.9 | 51.8 | 53.3 | 35.1 | 30.7 | 62.8 | 65.1 | 55.4 | 47.6 | 26.0 |
| InternVL2.5-26B | 38.1 | 36.4 | 32.7 | 30.0 | 16.3 | 54.5 | 44.0 | 54.3 | 43.8 | 34.0 | 67.9 | 62.3 | 67.1 | 60.1 | 48.0 |
| InternVL3-1B | 19.3 | 9.2 | 6.6 | 4.5 | 2.7 | 31.8 | 16.3 | 17.1 | 12.4 | 9.4 | 32.6 | 14.9 | 19.2 | 19.0 | 8.5 |
| InternVL3-2B | 32.6 | 31.3 | 24.3 | 15.6 | 13.2 | 40.9 | 34.1 | 42.2 | 19.7 | 20.1 | 52.7 | 50.5 | 38.6 | 30.7 | 22.0 |
| InternVL3-8B | 48.2 | 39.2 | 33.6 | 28.4 | 25.0 | 57.3 | 52.4 | 61.5 | 48.7 | 43.2 | 68.9 | 69.6 | 70.8 | 69.1 | 59.6 |
| InternVL3-14B | 55.8 | 51.2 | 48.0 | 37.8 | 26.7 | 57.4 | 54.7 | 54.8 | 48.8 | 47.3 | 76.7 | 56.2 | 69.8 | 63.3 | 44.3 |
| InternVL3-38B | 52.1 | 55.2 | 46.5 | 35.2 | 29.4 | 61.5 | 57.6 | 68.0 | 51.5 | 53.6 | 85.2 | 78.6 | 74.2 | 69.8 | 60.7 |
| Ovis2-1B | 19.5 | 13.2 | 6.3 | 5.0 | 2.2 | 39.9 | 13.9 | 7.6 | 9.0 | 6.9 | 19.4 | 10.0 | 11.5 | 10.5 | 6.5 |
| Ovis2-2B | 29.0 | 18.9 | 7.0 | 6.7 | 2.4 | 40.5 | 21.8 | 14.2 | 12.8 | 8.1 | 37.0 | 25.5 | 18.7 | 16.2 | 5.0 |
| Ovis2-4B | 38.4 | 36.7 | 17.3 | 12.9 | 9.0 | 53.2 | 32.7 | 31.9 | 22.4 | 11.3 | 58.7 | 49.9 | 39.6 | 19.1 | 8.5 |
| Ovis2-8B | 44.2 | 37.4 | 24.0 | 16.6 | 8.1 | 55.2 | 51.9 | 51.5 | 36.4 | 16.7 | 77.8 | 58.6 | 51.5 | 37.9 | 8.0 |
| Ovis2-16B | 56.5 | 53.4 | 39.3 | 28.0 | 12.6 | 66.7 | 58.6 | 58.0 | 41.8 | 29.8 | 76.2 | 71.5 | 48.3 | 36.4 | 15.7 |
| Ovis2-34B | 47.3 | 44.5 | 32.0 | 25.3 | 13.1 | 61.4 | 57.3 | 51.0 | 42.6 | 31.5 | 71.2 | 63.8 | 52.5 | 33.0 | 26.0 |
| Gemma3-4B | 35.0 | 32.9 | 26.8 | 19.5 | 17.1 | 38.8 | 32.6 | 35.6 | 29.1 | 25.3 | 40.1 | 39.3 | 44.4 | 40.8 | 40.1 |
| Gemma3-12B | 34.2 | 33.3 | 31.7 | 24.1 | 22.6 | 45.0 | 41.9 | 45.9 | 41.9 | 44.4 | 48.9 | 54.5 | 51.9 | 51.6 | 56.8 |
| Gemma3-27B | 41.4 | 34.1 | 31.4 | 32.3 | 30.0 | 47.2 | 47.3 | 44.9 | 44.4 | 49.9 | 60.6 | 67.9 | 60.3 | 61.8 | 56.9 |
| Idefics2-8B | 16.9 | 8.8 | 4.8 | 4.8 | 7.8 | 22.8 | 15.0 | 10.2 | 6.1 | 4.1 | 33.2 | 12.7 | 12.9 | 14.6 | 17.8 |
| Idefics2-8B-C | 10.4 | 5.3 | 1.0 | 1.0 | 1.1 | 16.5 | 13.0 | 1.0 | 0.8 | 1.5 | 11.8 | 0.7 | 0.0 | 0.2 | 1.0 |
| Mantis-Idefics2 | 15.2 | 14.6 | 12.4 | 9.1 | 2.1 | 23.6 | 13.0 | 19.0 | 7.7 | 4.7 | 36.1 | 33.9 | 30.3 | 20.1 | 15.4 |
| Idefics3-8B | 30.5 | 27.1 | 28.0 | 15.5 | 12.3 | 48.9 | 37.3 | 49.3 | 37.1 | 24.0 | 59.5 | 46.9 | 48.8 | 26.5 | 15.5 |
| Phi-3-Vision | 15.3 | 19.1 | 12.5 | 9.8 | 7.6 | 32.8 | 23.7 | 24.3 | 15.5 | 6.0 | 46.1 | 40.9 | 39.6 | 32.4 | 21.0 |
| Phi-3.5-Vision | 23.4 | 20.6 | 18.9 | 20.9 | 15.2 | 42.2 | 29.8 | 36.2 | 20.3 | 14.4 | 57.0 | 50.4 | 43.8 | 34.0 | 18.1 |
| Phi-4-Multimodal | 23.7 | 25.1 | 26.0 | 23.4 | 20.0 | 44.3 | 43.7 | 54.3 | 39.8 | 14.6 | 65.5 | 67.6 | 63.6 | 61.8 | 43.6 |
| NVILA-Lite-2B | 11.0 | 5.8 | 9.7 | 5.7 | 5.7 | 13.9 | 11.4 | 13.2 | 8.0 | 7.9 | 31.3 | 29.7 | 26.9 | 23.3 | 18.1 |
| NVILA-Lite-8B | 16.9 | 24.5 | 13.4 | 12.0 | 9.9 | 26.2 | 24.9 | 20.8 | 15.6 | 16.2 | 49.5 | 47.7 | 43.1 | 37.2 | 35.6 |
| Pixtral-12B | 34.7 | 34.6 | 24.0 | 26.5 | 15.9 | 52.8 | 41.0 | 49.6 | 36.6 | 36.6 | 77.5 | 68.8 | 59.6 | 52.9 | 44.8 |

Figure 27: Results of 46 models on the category DocVQA at various lengths.

Use the given documents to write a concise and short answer to the question about the entity shown in the image. Write your answer in the following format:
Answer: [answer]

Document (Title: Tropidacris collaris): Tropidacris collaris is a species of grasshopper in the family Romaleidae. A large South American grasshopper, it is also known as the blue-winged grasshopper although they vary greatly in coloration. It is common in both forests and dry areas of South America from Colombia to Argentina. In parts of northern Argentina, they are considered a pest. They are also popular among insect and terrarium enthusiasts.

Document (Title: Anarta myrtilli): [Warren] from Sintra, Portugal, the whole forewing is suffused with blackish, leaving only the white blotch on vein 2 conspicuous, and the orange of the hindwing, both above and below, is pale lemon yellow; as the insect is decidedly larger than average typical "myrtilli", it may prove a distinct species; at present I have seen only one - taken in the spring of 1909 by Mr N. C. Rothschild, and now in the Tring Museum.

Document (Title: Nipponaclerda biwakoensis): This species has become established (as of 2017) in the United States in the state of Louisiana, where it has rapidly become a serious pest of roseau cane, damaging over 80% of the reeds in some areas such as the Pass a Loutre Wildlife Management Area, where it is referred to by the older common name Phragmites scale insect or the more recently-coined name, roseau cane mealybug.

Document (Title: Dioctria atricapilla): The violet black-legged robber fly, Dioctria atricapilla, is a species of robber fly in the subfamily Dasypogoninae. This 9- to 12-millimeter long insect has a wingspan of roughly 7 to 9 mm and short, three-segmented antennae. It's a predatory insect, feeding mainly on smaller flies and predatory hymenopterans. It primarily thrives in grassland, and is seen from May to July.

Document (Title: Fauna of New Guinea): Notable endemic insect species include "Ornithoptera paradisea", "Ornithoptera chimaera", "Papilio weymeri", "Graphium weiskei", "Ideopsis hewitsonii", "Taenaris catops", "Parantica rotundata", "Parantica clinias", "Rosenbergia rufolineata", "Mecopus doryphorus", "Mecopus serrirostris", "Sphingnotus mirabilis", "Sphingnotus insignis", "Belionota aenea", "Poropterus solidus", "Poropterus gemmifer", "Aesernia splendens", "Aporhina bispinosa", "Eupholus petitii", "Eupholus bennetti", "Schizoeupsalis promissa", "Barystethus tropicus", "Eupholus geoffroyi", "Rhinoscapha loriai", "Rhinoscapha funebris", "Rhinoscapha insignis" "Alcides exornatus", "Alcides elegans", "Xenocerus lacrymans", "Arachnobas sector", "Arrhenodes digramma", "Eupholus magnificus", "Mecopus bispinosus", "Callictita" spp.. Also known from New Guinea are "Batocera wallacei", "Ithystenus curvidens", "Meganthribus pupa", "Sipalinus gigas", "Pelargoderus rubropunctatus", "Rhynchophorus bilineatus", "Gasterocercus anatinus", "Acalolepta australis", "Actinus imperialis", "Megacrania batesii".

· · ·

Document (Title: Melanopsis brevicula): Melanopsis brevicula is a small species of gastropod endemic to small streams near Agourai, Morocco. It is distinctive due to its minute size, flattened sculpture, low spire, and small aperture. It is known from a single location 10 km in area(Oued Ain Maarouf) which has been well surveyed, and found to be threatened by increasing human population, droughts of increasing extremity, water diversion, and pastoralization. Shell collecting presents a minor threat to populations. The species has been classified as Critically endangered by the IUCN.

Question: 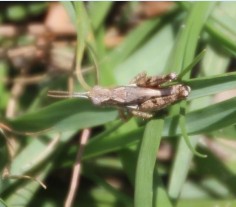 Which place is this insect endemic to?

Figure 28: Example of InfoSeek dataset in the VRAG category.

You are given a set of images. Please answer the question in Yes or No based on the given images. Write your answer in the following format: Answer: [answer]

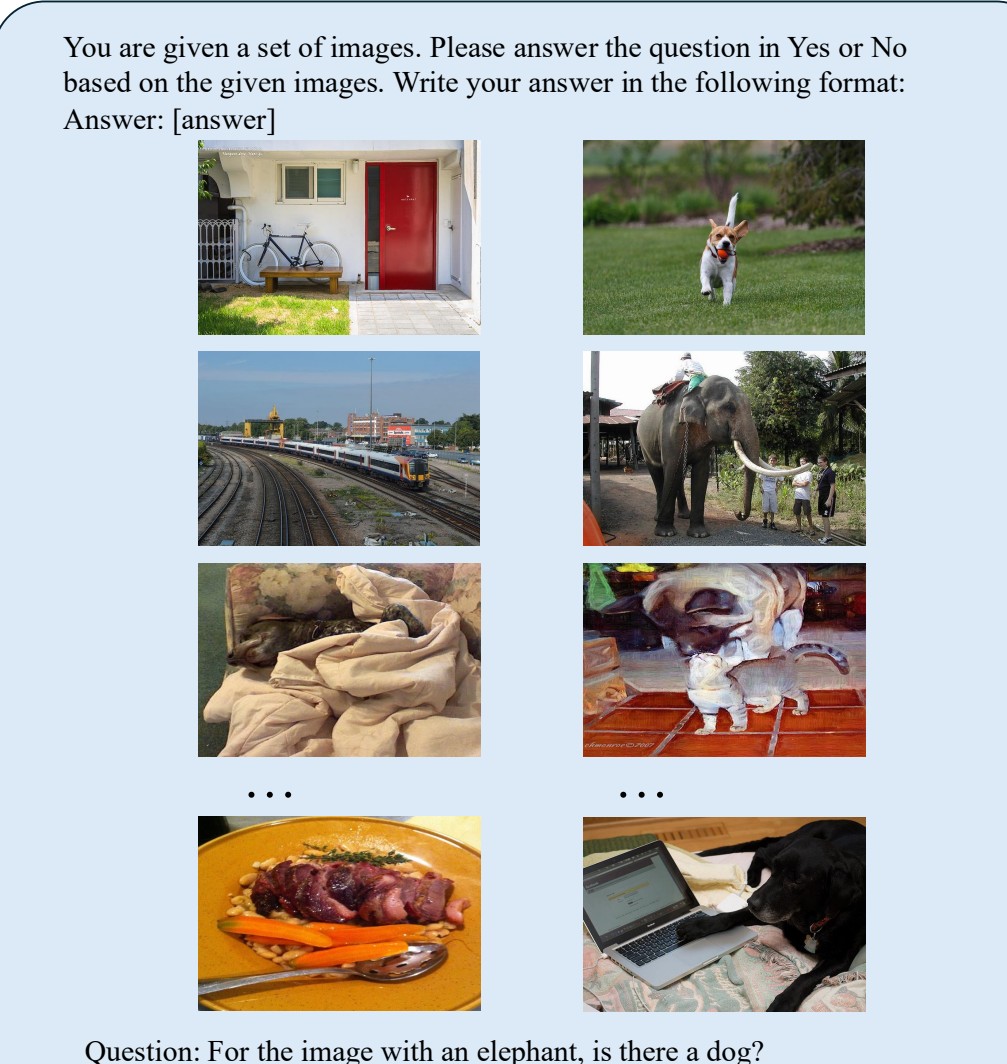

Question: For the image with an elephant, is there a dog?

Figure 29: Example of Visual Haystak-Single dataset in NIAH category. *Note: The input image list is shown in two columns for display clarity; in the actual input, the images are arranged in a single sequence.*

You are given interleaved text and images. Please answer the question with the option's letter (A, B, etc.) based on the given text and images. Write your answer in the following format:

Answer: [answer]

He also featured as Cooper in the worldwide tv show Game of Thrones in the episode "The Watchers on the Wall."He is the first son of the union between Tim Roth and Nikki Butler. He was named Timothy Hunter Roth; Timothy after his father and the Hunter after the popular journalist Hunter S.

• • •

I like my scones seasoned. Whether it's sweet or savoury, always add some salt to it. In terms of cooking, when you make your dough don't play with it. Just fold all the crumbs together, and it doesn't matter if it's bubbly. A lot of people play with the dough because they think it makes it smoother, but when a scone falls to pieces, it's because you've played with the dough too much."

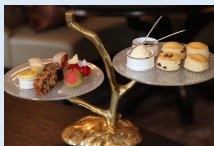

He was a musician with original compositions, skilled at playing the guitar. In November, 2021 Cormac was diagnosed with 3 germ cell cancer. George of the Jungle Star Brendan Fraser and his family story
Pamela Adlon: The Stardom FamilySmart Watches – Honest review based on my experience

• • •

Caretaker Sporting boss Tiago Fernandes said: 'The players did exactly what I asked them to do. In our game plan we know we had to be rigorous and they were almost perfect on that. 'We were aware of the opponent's quality but we, knowing our capacity and being creative and aggressive with and without ball, could try to surprise here.'Bruce Willis has reprised his iconic role as John McClane for a new Die Hard video.

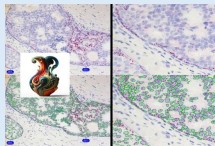

Question: Which of the following images appears in a certain image of the above document?

A. 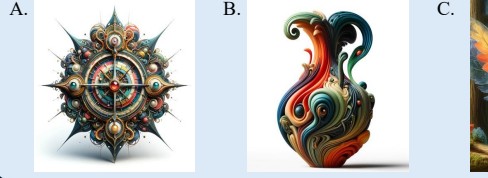 B. C. D. 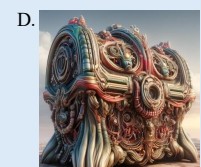

Figure 30: Example of MM-NIAH-Ret dataset in NIAH category.

You need to recognize entities in images. Use the provided mapping from the image to label to assign a label to the test image. Only output "label: {label}" and nothing else.

Training examples:

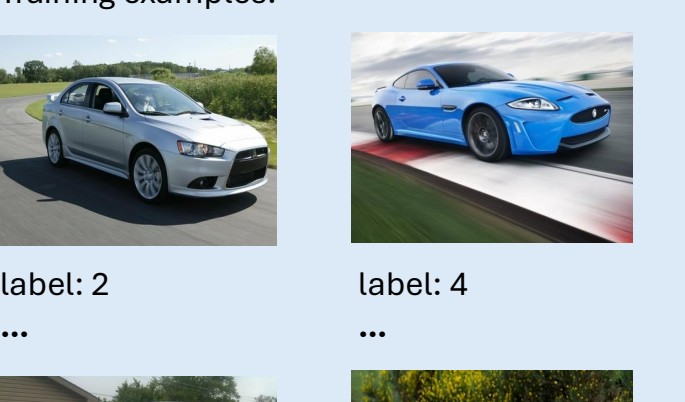

label: 2

...

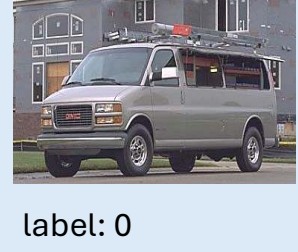

label: 4

...

label: 0

...

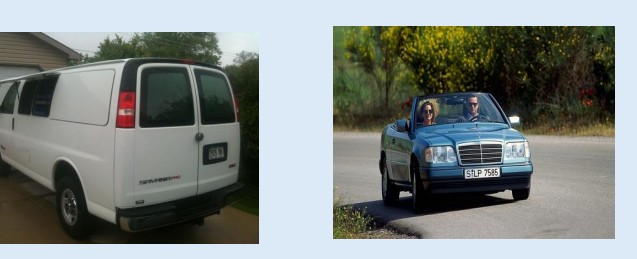

label: 0

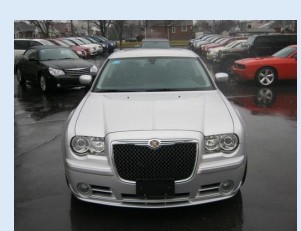

label: 3

label: 1

Now classify this image: 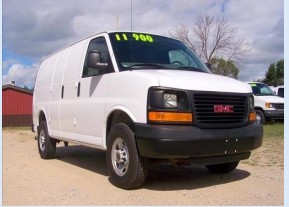

Figure 31: Example of the Stanford Cars dataset in the ICL category. *Note: The input image list is shown in three columns for display clarity; in the actual input, the images are arranged in a single sequence.*

You are given a government report from U.S. Government Accountability Office (GAO), and you are tasked to summarize the report. Write a concise summary (around 550 words) organized in multiple paragraphs. Where applicable, the summary should contain a short description of why GAO did this study, what GAO found, and what GAO recommends.

Government Report:
Document gao-12-156 (page 0):

Document gao-12-156 (page 1):

...
Now please summarize the report.

Figure 32: Example of GovReport in the summarization category. We only show two pages due to limited space.

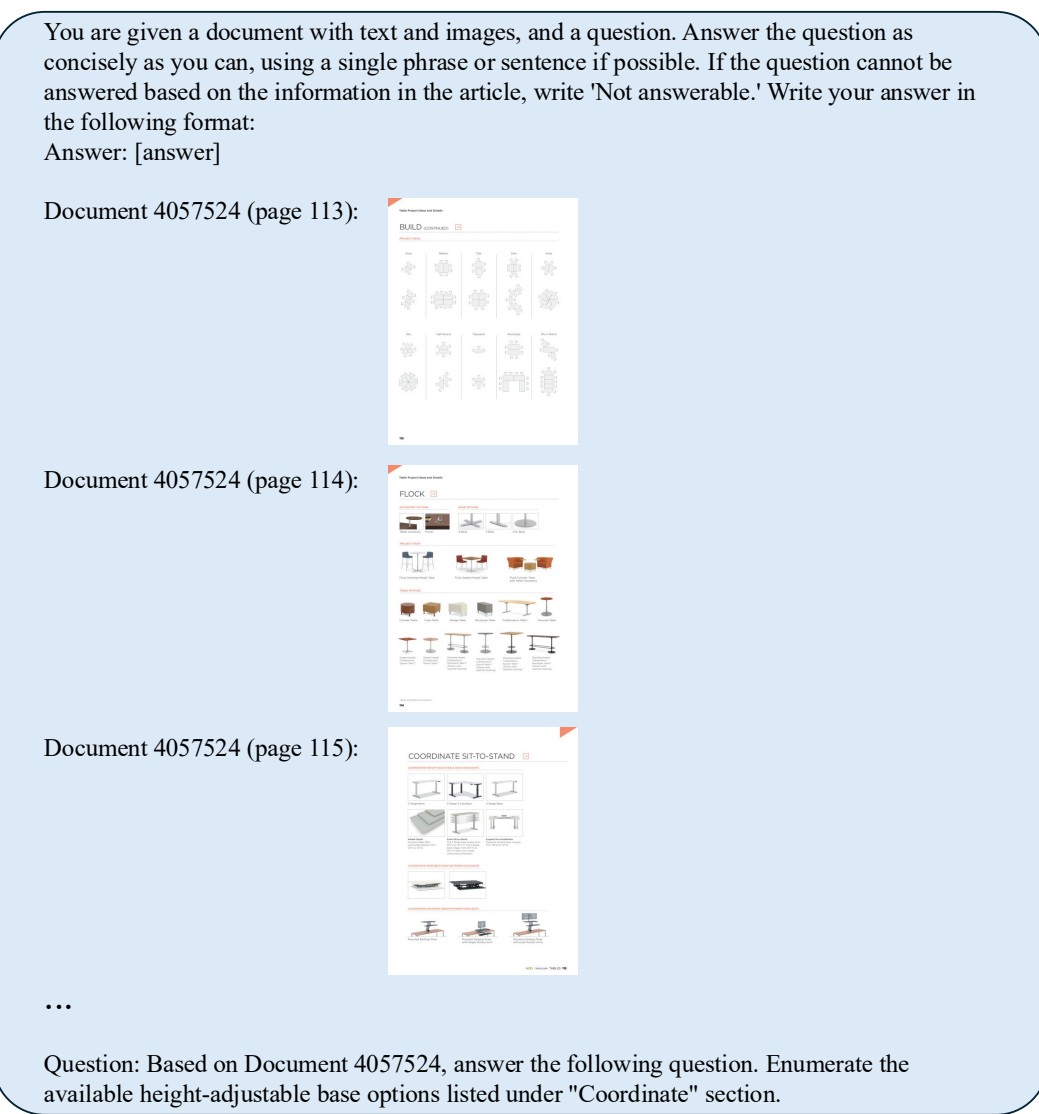

You are given a document with text and images, and a question. Answer the question as concisely as you can, using a single phrase or sentence if possible. If the question cannot be answered based on the information in the article, write 'Not answerable.' Write your answer in the following format:
Answer: [answer]

Document 4057524 (page 113):

Document 4057524 (page 114):

Document 4057524 (page 115):

…

Question: Based on Document 4057524, answer the following question. Enumerate the available height-adjustable base options listed under "Coordinate" section.

Figure 33: Example of LongDocURL dataset in the DocVQA category. We only show three pages due to limited space.

