# OpenReview forum: "MMLongBench: Benchmarking Long-Context Vision-Language Models Effectively and Thoroughly"
_NeurIPS.cc/2025/Datasets_and_Benchmarks_Track — NeurIPS 2025 Datasets and Benchmarks Track spotlight_

### Official Review · Reviewer_by6z · 2025-06-11

**Rating:** 6
**Confidence:** 4

**Summary:**

This paper proposes a benchmark, namely MMLongBench, to evaluate the performance of long-context vision-language models. According to the authors, MMLongBench covers a variety of long-context vision-language tasks, such as VRAG, NIAH, ICL, Summ, and DocVQA. By contrast, most of the existing benchmarks only focus on few tasks. Also, MMLongBench supports five input length, which may bring about positive impacts on model evaluation. Experimental results on selected LLMs show that the proposed benchmark has ability to reflect the performance of different models.

**Dataset Code Accessibility:**

Yes

**Dataset Code Comments:**

The dateset and related codes can be downloaded via https://github.com/EdinburghNLP/MMLongBench, and the installation guidelines are included.

**Ethical Considerations:**

No, there are no or only very minor ethics concerns

**Final Justification:**

I think that the author rebuttal has solved all my concerns about the original paper.

**Limitations Weaknesses:**

1. The experiments part only consider the performance of the proposed MMLongBench. It is better to include the comparsion with the existing benchmarks. For instance, for the NIAH task, the performances of the selected LLMs on MM-NIAH benchmark may also be included to compare with MMLongBench.

2. It is better to move the analyses of limitations from appendix to the main text of the paper.

**Strengths Contributions:**

1. The benchmark proposed in this paper is well designed, which covers several important tasks, thus it has ability to evaluate the performance of LLMs in different aspects.

2. Comparing with the previous counterparts, MMLongBench supports more types of images and 5 different context lengths.

3. Experimental results on many state-of-the-art LLMs show that MMLongBench has ability to evaluate the perfermance of LLMs in  appropriate way.

---

> ### Author Rebuttal · Authors · 2025-07-29
>
> > Weakness #1: The experiments part only consider the performance of the proposed MMLongBench. It is better to include the comparsion with the existing benchmarks. For instance, for the NIAH task, the performances of the selected LLMs on MM-NIAH benchmark may also be included to compare with MMLongBench.
>
> Thanks for your suggestions. Here, we provide two qualitative comparisons between MM-NIAH, Visual Haystack, and our reorganized versions in MMLongBench. We observed that LVLMs show a similar trend on both original datasets and our reorganized versions, showing the correctness of our reorganization.
>
> Meanwhile, it is challenging to conduct quantitative comparisons because of different context lengths and evaluated models: (1) For context lengths, MM-NIAH has random length from 1K to 72K, and Viusal Haystack uses the number of images as context length. In contrast, MMLongBench has five standard context lengths of 8K, 16K, 32K, 64K, and 128K. (1) For evaluated models, most models evaluated on MM-NIAH and Visual Haystack are no longer SoTA models, like InternVL-1.5 [1] and Gemini 1.5 [2]. We used InternVL-3 [5] and Gemini-2.5 [6] in our benchmarking.
>
> Thus, we provide two qualitative comparisons to show some insightful findings:
>
> **(1) Comparison with Visual Haystack.** We find that LVLMs show similar performance on both benchmarks. In Section 4.2 and Appendix D.1, we conducted a very comprehensive analysis of LVLMs' performance on Visual Haystack (binary classification, random guess is 50% accuracy). In this analysis, we identify that Visual Haystack is a very challenging dataset. Even Geimini-2.5-Pro struggles to surpass 80% accuracy, despite the context length being only 8K tokens (approximately 22 images). Even worse, open-source LVLMs typically perform at a level similar to random guess accuracy. This conclusion is consistent with the results of the Visual Haystack paper [3]. From Figure 2 in the original paper [3], we can see that all models perform poorly when there are 20 images in the input. Thus, we find LVLMs consistently show similar trends on both Visual Haystack and our reorganized version, showing the persistent challenge of multi-image retrieval.
>
> **(2) Comparison with MM-NIAH.** We also find that LVLMs show similar trends on the three different tasks (retrieval, counting, and reasoning) on both MM-NIAH and our re-organized version. Specifically, as shown in Section 4.2 (and Figure 2), we can find that while retrieval is a relatively easier task, most LVLMs achieve low performance on the counting and reasoning tasks. This is consistent with the results in the MM-NIAH paper [4]. As shown in Tables 3-5 in the original paper [4], InternVL-1.5 can maintain high performance on the retrieval task even when the context length is 64\~72K. In contrast, its performance falls quickly on counting and reasoning when the context length is just 24\~32K. Thus, we again find that LVLMs show similar trends on MM-NIAH and our reorganized version, demonstrating the difficulty of counting and reasoning in long-context input.
>
> > Weakness #2: It is better to move the analyses of limitations from appendix to the main text of the paper.
>
> Thanks for your suggestions. Currently, we discussed the limitations of our paper in Appendix F, such as evaluating larger models with more than 8xA100 GPUs and finding a cheaper method for assessing summarization. Given one additional page after rebuttal, we will include the limitation part in the main text of our final revision.
>
> #Reference
> 1. Chen, Zhe, et al. "How far are we to gpt-4v? closing the gap to commercial multimodal models with open-source suites." Science China Information Sciences 67.12 (2024): 220101.
> 2. Team, Gemini, et al. "Gemini 1.5: Unlocking multimodal understanding across millions of tokens of context." arXiv preprint arXiv:2403.05530 (2024).
> 3. Wu, Tsung-Han, et al. "Visual haystacks: A vision-centric needle-in-a-haystack benchmark." arXiv preprint arXiv:2407.13766 (2024).
> 4. Wang, Weiyun, et al. "Needle in a multimodal haystack." Advances in Neural Information Processing Systems 37 (2024): 20540-20565.
> 5. Zhu, Jinguo, et al. "Internvl3: Exploring advanced training and test-time recipes for open-source multimodal models." arXiv preprint arXiv:2504.10479 (2025).
> 6. Comanici, Gheorghe, et al. "Gemini 2.5: Pushing the frontier with advanced reasoning, multimodality, long context, and next generation agentic capabilities." arXiv preprint arXiv:2507.06261 (2025).

---

> > ### Comment · Reviewer_by6z · 2025-08-04
> >
> > Thanks for the comment. Based on your rebuttal, I would like to update my score to 6.

---

### Official Review · Reviewer_XLhL · 2025-06-29

**Rating:** 5
**Confidence:** 4

**Summary:**

The paper proposes a new benchmark for evaluating the performance and robustness of long sequence vision language models. A total of 13,331 examples are included in the benchmark, spanning across five standardized input lengths (8K-128K tokens). In total, 46 long sequence models are evaluated. The paper first identified a list of problems with existing benchmarks for long sequence models: limited coverage of downstream tasks, insufficient coverage of image types, lack of context length control, single context length. The results of the benchmark comparison suggested i) performance on a single task poorly reflects overall long-context ability; ii) long-context vision-language tasks present significant challenges for both closed-source and open-source models; and iii) models with stronger reasoning ability tend to exhibit better long-context capabilities.

**Dataset Code Accessibility:**

Yes

**Ethical Considerations:**

No, there are no or only very minor ethics concerns

**Final Justification:**

I will maintain the score at accept.

**Limitations Weaknesses:**

1. LLM as a judge may be susceptible to verbosity bias and self-enhancement bias [1]. It would be more beneficial to have benchmarks that are easily verifiable.
2. ICL testing is limited only to image classification with large label spaces, where in particular the labels are discrete integer labels. I think it would be a lot more interesting if one can build ICL benchmarks that evaluate VLM’s spatial understanding [2,3].
3. No evaluation on videos. Many of the long context vision LMs are built to take videos as input. It would be a really to have multiple videos as input and perform needle in the haystack evaluation / other evaluation metrics to test that capability.

[1] Wang, Peiyi, Lei Li, Liang Chen, Zefan Cai, Dawei Zhu, Binghuai Lin, Yunbo Cao, Qi Liu, Tianyu Liu, and Zhifang Sui. "Large language models are not fair evaluators." arXiv preprint arXiv:2305.17926 (2023).
[2] Bai, Yutong, Xinyang Geng, Karttikeya Mangalam, Amir Bar, Alan L. Yuille, Trevor Darrell, Jitendra Malik, and Alexei A. Efros. "Sequential modeling enables scalable learning for large vision models." In Proceedings of the IEEE/CVF Conference on Computer Vision and Pattern Recognition, pp. 22861-22872. 2024.
[3] Yeh, Chun-Hsiao, Chenyu Wang, Shengbang Tong, Ta-Ying Cheng, Ruoyu Wang, Tianzhe Chu, Yuexiang Zhai, Yubei Chen, Shenghua Gao, and Yi Ma. "Seeing from another perspective: Evaluating multi-view understanding in mllms." arXiv preprint arXiv:2504.15280 (2025).

**Strengths Contributions:**

1. The paper identifies the limitations of the current long context sequence model benchmarks and tasks that existing benchmarks do not sufficiently cover.
2. The benchmark identifies Text-image interleaved Needle In the Haystack as a major bottleneck/challenge for both open and closed source long context VLM.
3. The paper is well written and is able to convey the key messages.

---

> ### Author Rebuttal · Authors · 2025-07-29
>
> > Weakness #1: LLM as a judge may be susceptible to verbosity bias and self-enhancement bias [1]. It would be more beneficial to have benchmarks that are easily verifiable.
>
> Thanks for your suggestion. First, we want to clarify that **LLM-as-a-judge is only used in the summarization task**; other four tasks are all evaluated using easily verifiable metrics. Second, we provide detailed experiments to show that there is no verbosity bias and self-enhancement bias.
>
> We offer a thorough discussion of these two points separately:
>
> **(1) For using the LLM-as-a-judge:**
>
> We would like to emphasize that we only utilized LLM-as-a-judge in the summarization task. For the other four tasks (i.e., Visual RAG, NIAH, Many-Shot ICL, and DocVQA), we only used metrics that can be automatically verified, such as substring exact match (SubEM) and Accuracy. Thus, our benchmark is easily verifiable.
>
> Furthermore, here is an explanation of why we chose to use the GPT-4o score for summarization. N-gram overlap metrics, such as ROUGE-L, have long been condemned for their poor correlation with human judgment for long-form generation [1, 2]. We provide a very detailed comparison of ROUGE-L and GPT-4o score **in Appendix A.4 (also in Figure 8)**, where we find that ROUGE-L exhibits minimal difference across various LVLMs and input lengths. For example, InternVL3-2B achieves ROUGE-L scores comparable to GPT-4o.
>
> Thus, following previous works [3], we choose to use the GPT-4o score for the summarization task, which shows higher correlation with human judges. Meanwhile, more and more works started to use LLM-as-a-judge in their evaluation [3-7].
>
> **(2) For the bias of LLM-as-a-judge:**
>
> We provide various experiments to show that there is no verbosity bias and self-enhancement bias. (1) For the verbosity bias, we strictly set the generation length (the max_new_token argument in Transformers) to be the same for all the LVLMs. For example, it is ​​384 tokens for Summarization. (2) For self-enhancement bias, many existing works have also employed LLM-as-a-judge for summarization [3, 4, 6] and found that self-enhancement bias is not a significant issue given the strong ability of current LLMs. Furthermore, we conducted **"Atomic Claims Verification" and "GPT-4o Judgment Verification" in Appendix A.4**, where we find GPT-4o is virtually always reliable for evaluating summarization.
>
> > Weakness #2: ICL testing is limited only to image classification with large label spaces, where in particular the labels are discrete integer labels. I think it would be a lot more interesting if one can build ICL benchmarks that evaluate VLM’s spatial understanding.
>
> Thanks for your concern. However, it's worth noting that our choice of including image classification is reasonable and not limited. Below, we justify our design from three perspectives: (1) it is hard to include spatial understanding, (2) our classification datasets are very diverse, and (3) including classification is a pretty common practice.
>
> First, it is hard to include spatial understanding in our long-context benchmark because of the pre-trained knowledge of LVLMs. While we can provide models with many in-context exemplars, it is unclear whether models correctly conduct spatial understanding based on the in-context exemplars or their pre-trained knowledge. If the pre-trained knowledge of LVLMs plays the critical role, our benchmark will not test their long-context ability. In contrast to this, as we discussed in Section 3.1 and Appendix A.3, we mapped the original natural language labels (e.g., food names) to class IDs (e.g., 0, 1, ...) for classification tasks. This can ensure that our tasks require models to learn new tasks rather than relying on pre-training knowledge.
>
> Second, our many-shot ICL data are not limited. We have included four datasets from different domains. Specifically, they are Stanford Cars for diverse cars, Food101 for various food, SUN397 for scenes,  and iNat2021 for species. Thus, our design for many-shot ICL provides a comprehensive coverage of classification data.
>
> Third, classification with large label spaces is a common practice of testing many-shot ICL ability. For example, HELMET [3], a text-only long-context benchmark, included five datasets for intent classification and didn't incorporate other datasets. Similarly, many other studies [8-11] have also adopted classification tasks to evaluate the many-shot ICL capabilities of models.
>
> Thanks for your suggestion again. We will include more discussion in our final revision to make this part more straightforward.
>
> > Weakness #3: No evaluation on videos. Many of the long context vision LMs are built to take videos as input. It would be a really to have multiple videos as input and perform needle in the haystack evaluation / other evaluation metrics to test that capability.
>
> Thanks for your suggestion! Nonetheless, we found that there are already many established long-context benchmarks for video and audio [12-17]. Thus, we didn't consider video and audio in our benchmark repeatedly, and including those datasets will be very trivial and redundant. Meanwhile, our benchmark targets at filling the gap of evaluating LVLMs on various vision-language scenarios (i.e., text-image interleaved ones), which is still missing in current literature. Furthermore, we provide a thorough explanation with the following points:
>
> **(1) Long-context benchmarks for video and audio are well-established [12-17]:** For example, LVLMs developers can use MovieChat-1K [14] to evaluate LVLMs' ability to understand long videos (10K to 12K frames per video). Also, LongVideoBench [13] evaluates the ability to understand long interleaved video and text. Also, for audio, both AudioBench [16] and BLAB [17] have long audio data (in some cases, more than 100 hours). Thus, current long-context benchmarks for video and audio are well-established, and including that data will be very redundant and trivial.
>
> We mentioned this in the "Long-context benchmarks" part in Section 2 Related Work: "... only report long-context performance on other modalities, such as video [9, 65, 66] or audio [10, 33]..." We will make this discussion more detailed in our final revision.
>
> **(2) Unlike video and audio, there is currently a lack of benchmarks on the vision-language (interleaved text and images) long-context data:** Existing vision-language long-context datasets (e.g., MM-NIAH [18], Visual Haystack [19], etc) have several design issues, like limited coverage of downstream tasks, lack of context length control, and so on.
>
> Therefore, the motivation behind our work is to propose the first benchmark that can effectively and thoroughly evaluate LVLMs on long-context vision-language tasks. In our benchmarks, we include 13,331 examples from 16 datasets across five categories of different vision-language tasks, spanning Visual RAG, Many-Shot ICL, Long DocVQA, etc. We designed various ways for different datasets to build long contexts. Also, we introduced a cross-modal tokenization scheme that combines vision patches and text tokens to achieve precise control over context length.
>
> **(3) For LVLM developers, they can use multiple long-context benchmarks to have an evaluation of all modalities:** In practice, LVLM developers can use all the long-context benchmarks of text-only, vision-language, video, and audio data to have a holistic understanding. This is standard practice in current LVLMs development. For example, Phi-4-mini [20] utilizes all of ChartQA, DocVQA, InfoVQA, and OCRBench to gain a holistic understanding of its document understanding performance.
>
> # Reference
> 1. Goyal, Tanya, Junyi Jessy Li, and Greg Durrett. "News summarization and evaluation in the era of gpt-3." arXiv 2209.12356.
> 2. Deutsch, Daniel, Rotem Dror, and Dan Roth. "Re-Examining System-Level Correlations of Automatic Summarization Evaluation Metrics." NAACL. 2022.
> 3. Yen, Howard, et al. "HELMET: How to evaluate long-context models effectively and thoroughly." ICLR. 2025.
> 4. Crupi, Giuseppe, et al. "On the Effectiveness of LLM-as-a-judge for Code Generation and Summarization." IEEE TSE (2025).
> 5. Li, Haitao, et al. "Llms-as-judges: a comprehensive survey on llm-based evaluation methods." arXiv 2412.05579.
> 6. An, Chenxin, et al. "L-Eval: Instituting Standardized Evaluation for Long Context Language Models." ACL. 2024.
> 7. Li, Xuechen, et al. "Alpacaeval: An automatic evaluator of instruction-following models." May 2023.
> 8. Li, Tianle, et al. "Long-context LLMs Struggle with Long In-context Learning." TMLR.
> 9. Bertsch, Amanda, et al. "In-Context Learning with Long-Context Models: An In-Depth Exploration." First LCFM Workshop@ ICML 2024.
> 10. Ratner, Nir, et al. "Parallel Context Windows for Large Language Models." ACL. 2023.
> 11. Xu, Xiaoyue, Qinyuan Ye, and Xiang Ren. "Stress-testing long-context language models with lifelong icl and task haystack." NIPS (2024): 15801-15840.
> 12. Chen, Yukang, et al. "LongVILA: Scaling Long-Context Visual Language Models for Long Videos." ICLR (2025).
> 13. Wu, Haoning, et al. "Longvideobench: A benchmark for long-context interleaved video-language understanding." NIPS (2024): 28828-28857.
> 14. Song, Enxin, et al. "Moviechat: From dense token to sparse memory for long video understanding." CVPR. 2024.
> 15. Team, Gemini, et al. "Gemini 1.5: Unlocking multimodal understanding across millions of tokens of context." arXiv 2403.05530.
> 16. Wang, Bin, et al. "Audiobench: A universal benchmark for audio large language models." arXiv 2406.16020.
> 17. Ahia, et al. "BLAB: Brutally Long Audio Bench." arXiv 2505.03054.
> 18. Wang, et al. "Needle in a multimodal haystack." NIPS (2024): 20540-20565.
> 19. Wu, et al. "Visual haystacks: A vision-centric needle-in-a-haystack benchmark." arXiv 2407.13766.
> 20. Abouelenin, et al. "Phi-4-mini technical report: Compact yet powerful multimodal language models via mixture-of-loras." arXiv2503.01743.

---

### Official Review · Reviewer_Uv7t · 2025-07-02

**Rating:** 6
**Confidence:** 4

**Summary:**

This paper addresses the gap that current Multimodal Large Language Models (MLLMs) possess long context windows but lack suitable benchmarks for evaluating this capability. The authors propose MMLongBench, a large-scale multimodal benchmark encompassing five representative long-context tasks. It is composed of 13,331 examples derived from open-source benchmarks and integrating both natural and synthetic images. The authors systematically controls context length to evaluate a wide range of proprietary and open-source MLLMs (total 46 models) on the proposed benchmark. The study reveals three key findings, providing a clear roadmap for future research on long-text MLLMs.

**Additional Feedback:**

1. It is better to provide the `average context length` or `average image number` for the five categories in Table 2.

**Dataset Code Accessibility:**

Yes

**Dataset Code Comments:**

N/A

**Ethical Comments:**

The datasets are developed based on the open-source, academic datasets. No notable ethical concerns have been identified in this context.

**Ethical Considerations:**

No, there are no or only very minor ethics concerns

**Final Justification:**

The rebuttal has addressed all my concerns and I believe this is a high-quality paper.

**Limitations Weaknesses:**

1. While the ablation study on closed-sourced Gemini 2.5 vs. Gemini 2.5-T suggests reasoning models perform better, this single-model validation lacks generalizability. How do other open-source models like Kimi-VL-Instruct/Thinking perform under similar conditions? Additionally, were different prompting strategies applied to these model types?
2. For the summarization category, the authors employ GPT-4o as the evaluation model, which incurs significant costs. How would the results differ when using locally-deployed LLM models (e.g. Qwen, DeepSeek) instead?

Regarding the contribution of introducing a large-scale, comprehensive benchmark for evaluating the long-context capabilities of MLLMs, as well as the workload and presentation demonstrated in the paper, the current version is indeed deserving of acceptance. My questions are raised with the aim of encouraging the authors to provide additional evidence and further strengthen the paper.

**Strengths Contributions:**

1. The benchmark effectively addresses the research gap in evaluating long-context capabilities of current MLLMs under a diverse range of downstream tasks, filling a critical void in the field.
2. The authors rigorously control context length in their evaluations, enabling developers to clearly understand performance trends as context length varies.
3. The evaluation framework is comprehensive and insightful, providing valuable insights into model behaviors under long-context scenarios.
4. The paper's presentation is commendable—well-organized, structurally sound, and highly transparent, enhancing reader comprehension.

---

> ### Author Rebuttal · Authors · 2025-07-29
>
> > Weakness #1: While the ablation study on closed-sourced Gemini 2.5 vs. Gemini 2.5-T suggests reasoning models perform better, this single-model validation lacks generalizability. How do other open-source models like Kimi-VL-Instruct/Thinking perform under similar conditions? Additionally, were different prompting strategies applied to these model types?
>
> We are grateful for your suggestions. First, we want to clarify that we actually included **three reasoning models in our current paper, not just a single model**. Meanwhile, we included two more open-source reasoning LVLMs in our rebuttal to support our claim. Here is the in-depth discussion:
>
> First, there are actually three reasoning LVLMs in our current paper. **Other than Gemini-2.0-Flash-T (Thinking) [2], there are also Gemini-2.5-Flash [3] and Gemini-2.5-Pro [3].** According to the Geimin API [1], Gemini-2.5 models are reasoning models by default. Thus, we have actually included three reasoning LVLMs in our current paper, not just a single model.
>
> Second, we also tested two new open-source reasoning models, namely MiMo-VL-7B-SFT and Kimi-VL-A3B-Thinking (2506 version). For Kimi-VL-A3B-Thinking, we compare it with its corresponding non-reasoning model, Kimi-VL-A3B-Instruct. For MiMo-VL-7B-SFT, there is no corresponding non-reasoning model. Thus, we compare MiMo-VL-7B-SFT with Qwen2.5-VL-7B, one of the SoTA models on MMLongBench. We list the performance of these two models below. (Due to time constraints of the rebuttal and the need for long reasoning generation, we only evaluated up to 32K. 64K and 128K will be included in the final version.)
>
> **(1) MiMo-VL-7B-SFT (Reasoning) v.s. Qwen2.5-VL-7B (Non-Reasoning):** As shown in the tables below, we observe that the average performance of MiMo-VL-7B-SFT is much higher than Qwen2.5-VL-7B across all context lengths. For instance, at 8K context length, MiMo-VL-7B-SFT achieves 66.0 on average, substantially surpassing Qwen2.5-VL-7B’s 57.4. Meanwhile, MiMo-VL-7B-SFT consistently obtains higher performance on each task, except Summarization (This is quite common; see Section 4.1: Different models exhibit different strengths). This advantage continues as the context length increases.
>
> | MiMo-VL-7B-SFT | VRAG | NIAH | ICL | Summ | DocVQA | Avg. |
> |:--------------|:----:|:----:|:----:|:----:|:----:|----:|
> | 8K | 67.4 | 75.5 | 98.0 | 20.0 | 69.1 | 66.0 |
> | 16K | 63.6 | 71.5 | 93.2 | 21.2 | 68.1 | 63.5 |
> | 32K | 63.0 | 65.6 | 90.5 |  21.2 | 67.2 | 61.5 |
>
> | Qwen2.5-VL-7B | VRAG | NIAH | ICL | Summ | DocVQA | Avg. |
> |:--------------|:----:|:----:|:----:|:----:|:----:|----:|
> | 8K | 50.1 | 57.3 | 95.6 | 23.5 | 60.7 | 57.4 |
> | 16K | 48.7 | 53.0 | 91.5 | 29.1 | 57.1 | 55.9 |
> | 32K | 43.2 | 47.7 | 78.5 | 30.8 | 57.2 | 51.5 |
>
> **(2) Kimi-VL-A3B-Thinking (Reasoning) v.s. Kimi-VL-A3B-Instruct (Non-Reasoning):** Again, we can observe that reasoning models can outperform the non-reasoning version. Specifically, at 8K context length, Kimi-VL-A3B-Thinking achieves 61.1 on average, while Kimi-VL-A3B-Instruct only reaches 55.2. This observation further supports our finding that reasoning models can outperform other models.
>
> | Kimi-VL-A3B-Thinking | VRAG | NIAH | ICL | Summ | DocVQA | Avg. |
> |:--------------|:----:|:----:|:----:|:----:|:----:|----:|
> | 8K | 52.2 | 63.3 | 99.0 | 27.1 | 63.7 | 61.1 |
> | 16K | 51.7 | 60.3 | 96.2 | 29.0 | 59.6 | 59.4 |
> | 32K | 50.1 | 54.7 | 93.0 | 29.5 | 61.6 | 57.8 |
>
> | Kimi-VL-A3B-Instruct | VRAG | NIAH | ICL | Summ | DocVQA | Avg. |
> |:--------------|:----:|:----:|:----:|:----:|:----:|----:|
> | 8K | 50.3 |  57.9 | 97.0 | 22.3 | 48.6 | 55.2 |
> | 16K | 47.8 | 55.0 | 95.5 | 23.6 | 45.8 | 53.5 |
> | 32K | 44.8 | 48.7 | 86.5 | 26.4 | 51.5 | 51.56 |
>
> > Weakness #2: For the summarization category, the authors employ GPT-4o as the evaluation model, which incurs significant costs. How would the results differ when using locally-deployed LLM models (e.g. Qwen, DeepSeek) instead?
>
> Thanks for your advice. Here, we introduce two new models as judges for the Summarization task: Qwen2.5-7B-Instruct and Qwen2.5-72B-Instruct. Then, we evaluated the summaries from 8 different LVLMs as shown in the tables below. We find that Qwen2.5-7B-Instruct cannot be used as the judge since it cannot distinguish different input lengths and models of different sizes. Better than the 7B model, Qwen2.5-72B-Instruct could be a low-cost replacement for GPT-4o; however, it still makes occasional mistakes. Thus, given the stronger ability of GPT-4o, we choose to use GPT-4o for the evaluation of summarization in our paper.
>
> Here is a detailed discussion of Qwen2.5-7B-Instruct and Qwen2.5-72B-Instruct as the judge:
>
> **(1) Using Qwen2.5-7B-Instruct:** First, we observe that Qwen2.5-7B-Instruct cannot distinguish different input lengths. For example, on Multi-LexSum, Gemma3-12B achieves scores that are consistently around 40, regardless of whether the input length is 8K or 128K tokens. Other models, like Qwen2.5-VL-7B-Instuct and InternVL3-14B, also demonstrate the same problem on Multi-LexSum.
>
> Second, another issue is that Qwen2.5-7B-Instruct cannot distinguish models of different sizes. For example, on the GovReport, InternVL3-14B obtains scores similar to GPT-4o when the input lengths are 32K, 64K, and 128K. Furthermore, on Multi-LexSum, Qwen2.5-VL-7B-Instruct achieves scores similar to Qwen2.5-VL-72B-Instruct when the input lengths are 8K, 16K, and 32K.
>
> | GovReport (Qwen2.5-7B-Inst) |   8K |   16K |   32K |   64K | 128K |
> |:--------------------|-------:|--------:|--------:|--------:|---------:|
> | GPT-4o              |   33.2 |    39.5 |    42.7 |    52.2 |     57.0 |
> | InternVL3-2B        |   24.9 |    40.1 |    32.9 |    44.1 |     50.2 |
> | InternVL3-14B       |   34.4 |    35.2 |    42.6 |    50.0 |     57.8 |
> | Qwen2.5-VL-7B-Inst  |   39.2 |    45.6 |    45.0 |    47.6 |     50.8 |
> | Qwen2.5-VL-72B-Inst |   32.6 |    38.5 |    46.5 |    52.4 |     43.3 |
> | Gemma3-4B           |   27.8 |    36.4 |    44.5 |    52.9 |     52.5 |
> | Gemma3-12B          |   31.1 |    37.0 |    44.1 |    47.5 |     53.4 |
> | Gemma3-27B          |   34.5 |    42.5 |    47.5 |    57.4 |     55.5 |
>
>
> | Multi-LexSum (Qwen2.5-7B-Inst) |   8K |   16K |   32K |   64K | 128K |
> |:--------------------|-------:|--------:|--------:|--------:|---------:|
> | GPT-4o              |   39.3 |    42.9 |    45.7 |    44.3 |     49.3 |
> | InternVL3-2B        |   27.6 |    29.4 |    31.0 |    29.7 |     29.6 |
> | InternVL3-14B       |   37.9 |    38.8 |    40.8 |    37.9 |     42.0 |
> | Qwen2.5-VL-7B-Inst  |   39.5 |    39.1 |    43.4 |    37.9 |     42.2 |
> | Qwen2.5-VL-72B-Inst |   40.0 |    39.5 |    42.2 |    43.0 |     27.6 |
> | Gemma3-4B           |   30.8 |    34.4 |    32.1 |    34.6 |     34.2 |
> | Gemma3-12B          |   40.5 |    40.5 |    44.2 |    41.6 |     40.7 |
> | Gemma3-27B          |   37.5 |    37.8 |    42.8 |    45.0 |     43.2 |
>
> **(2) Using Qwen2.5-72B-Instruct:** We find that Qwen2.5-72B-Instruct can distinguish different input lengths and models of different sizes. However, there are still some occasional mistakes. For example, on GovReport, Gemma3-4B and Gemma3-12B obtain quite similar scores when the input length is 128K. Also, InternVL3-14B obtains a score higher than GPT-4o when the input length is 128K on the GovReport.
>
> Thus, we conclude that Qwen2.5-72B-Instruct can be a low-cost replacement for GPT-4o; however, it still makes occasional errors. For a more reliable evaluation, we choose to use GPT-4o in our paper.
>
> | GovReport (Qwen2.5-72B-Inst) |   8K |   16K |   32K |   64K | 128K |
> |:--------------------|-------:|--------:|--------:|--------:|---------:|
> | GPT-4o              |   29.7 |    38.9 |    45.5 |    54.5 |     59.4 |
> | InternVL3-2B        |   23.5 |    29.7 |    33.4 |    46.5 |     46.7 |
> | InternVL3-14B       |   30.3 |    35.2 |    39.7 |    53.0 |     61.1 |
> | Qwen2.5-VL-7B-Inst  |   28.4 |    34.7 |    37.7 |    46.0 |     51.7 |
> | Qwen2.5-VL-72B-Inst |   27.6 |    32.9 |    38.8 |    50.1 |     45.1 |
> | Gemma3-4B           |   27.2 |    37.1 |    43.9 |    49.8 |     51.2 |
> | Gemma3-12B          |   31.6 |    38.5 |    42.3 |    48.7 |     52.0 |
> | Gemma3-27B          |   32.2 |    40.1 |    46.5 |    59.1 |     63.5 |
>
> | Multi-LexSum (Qwen2.5-72B-Inst) |   8K |   16K |   32K |   64K | 128K |
> |:--------------------|-------:|--------:|--------:|--------:|---------:|
> | GPT-4o              |   58.1 |    64.7 |    69.0 |    69.1 |     73.5 |
> | InternVL3-2B        |   40.7 |    45.7 |    48.9 |    44.1 |     49.6 |
> | InternVL3-14B       |   51.5 |    59.4 |    62.5 |    66.8 |     67.4 |
> | Qwen2.5-VL-7B-Inst  |   49.3 |    54.8 |    57.9 |    61.6 |     58.8 |
> | Qwen2.5-VL-72B-Inst |   53.2 |    60.4 |    61.8 |    62.6 |     39.2 |
> | Gemma3-4B           |   43.5 |    51.8 |    54.5 |    56.6 |     58.1 |
> | Gemma3-12B          |   54.5 |    61.0 |    64.4 |    67.4 |     68.8 |
> | Gemma3-27B          |   54.1 |    62.8 |    64.1 |    68.0 |     70.3 |
>
>
> #Reference
> 1. Gemini API Documentation, 2025
> 2. Introducing Gemini 2.0: our new AI model for the agentic era, 2025
> 3. Comanici, Gheorghe, et al. "Gemini 2.5: Pushing the frontier with advanced reasoning, multimodality, long context, and next generation agentic capabilities." arXiv preprint arXiv:2507.06261 (2025).

---

> > ### Comment · Reviewer_Uv7t · 2025-08-04
> >
> > Thanks for the authors' response. Regarding the new experiments on the open-source instruction/reasoning models and utilizing qwen2.5-vl-instruct as the judge, the results further strengthen the findings in the paper. Therefore, I would like to increase my score to 6.

---

### Official Review · Reviewer_aW4B · 2025-07-07

**Rating:** 5
**Confidence:** 4

**Summary:**

This paper provides a benchmark for Long-Context Vision Language Models, testing 46 models frontier Vision Language Models including open-source as well as closed source models. This paper assesses five VLM tasks including (1) Visual Retrieval-Augmented Generation (VRAG), (2) Needle-in-a-Haystack (NIAH), (3) Many-Shot In-Context Learning (ICL), (4) Summarization, and (5) Long-Document Visual Question Answering (DocVQA). Across these tasks, the benchmark covers 13,331 examples.

Authors also address longer context lengths for text prompts, which is relevant consideration for long-context vision language models.

**Dataset Code Accessibility:**

Yes

**Ethical Considerations:**

No, there are no or only very minor ethics concerns

**Final Justification:**

Authors have addressed my concerns regarding video and audio modeling, and I maintain the current scoring since it is adequate.

**Limitations Weaknesses:**

only considers static image and text, however long-context VLM evaluations should also consider video and audio modalities

**Strengths Contributions:**

1) authors evaluate both open source and closed source VLMs, including Gemini 2.5 pro which is fairly recent and are comprehensive in their evaluation of state-of-art models
2) the metrics designed cover diverse long-context relevant tasks
3) key observations include:
i) models that are trained on shorter context lengths of 32k tokens can still generalize well on 128k input length as well, since they are able to leverage the strength of the underlying LLM.
ii) reasoning improves long-context ability
iii) diverse range of metrics are required to qualify long-context abilities of LVLMs (different models are better at different metrics).

---

> ### Author Rebuttal · Authors · 2025-07-29
>
> > Weakness #1: only considers static image and text, however long-context VLM evaluations should also consider video and audio modalities
>
> Thanks for your suggestion! Nonetheless, we found that there are already many established long-context benchmarks for video and audio [1-6]. Thus, we didn't consider video and audio in our benchmark repeatedly, and including those datasets will be very trivial and redundant. Meanwhile, our benchmark targets at filling the gap of evaluating LVLMs on various vision-language scenarios (i.e., text-image interleaved ones), which is still missing in current literature. Furthermore, we provide a thorough explanation with the following points:
>
> **(1) Long-context benchmarks for video and audio are well-established [1-6]:** For example, LVLMs developers can use MovieChat-1K [3] to evaluate LVLMs' ability to understand long videos (10K to 12K frames per video). Also, LongVideoBench [2] evaluates the ability to understand long interleaved video and text. Also, for audio, both AudioBench [5] and BLAB [6] have long audio data (in some cases, more than 100 hours). Thus, current long-context benchmarks for video and audio are well-established, and including that data will be very redundant and trivial.
>
> We mentioned this in the "Long-context benchmarks" part in Section 2 Related Work: "... only report long-context performance on other modalities, such as video [9, 65, 66] or audio [10, 33]..." We will make this discussion more detailed in our final revision.
>
> **(2) Unlike video and audio, there is currently a lack of benchmarks on the vision-language (interleaved text and images) long-context data:** Existing vision-language long-context datasets (e.g., MM-NIAH [7], Visual Haystack [8], etc) have several design issues, like limited coverage of downstream tasks, lack of context length control, and so on.
>
> Therefore, the motivation behind our work is to propose the first benchmark that can effectively and thoroughly evaluate LVLMs on long-context vision-language tasks. In our benchmarks, we include 13,331 examples from 16 datasets across five categories of different vision-language tasks, spanning Visual RAG, Many-Shot ICL, Long DocVQA, etc. We designed various ways for different datasets to build long contexts. Also, we introduced a cross-modal tokenization scheme that combines vision patches and text tokens to achieve precise control over context length.
>
> **(3) For LVLM developers, they can use multiple long-context benchmarks to have an evaluation of all modalities:** In practice, LVLM developers can use all the long-context benchmarks of text-only, vision-language, video, and audio data to have a holistic understanding. This is standard practice in current LVLMs development. For example, Phi-4-mini [9] utilizes all of ChartQA, DocVQA, InfoVQA, and OCRBench to gain a holistic understanding of its document understanding performance.
>
> # Reference
> 1. Chen, Yukang, et al. "LongVILA: Scaling Long-Context Visual Language Models for Long Videos." ICLR (2025).
> 2. Wu, Haoning, et al. "Longvideobench: A benchmark for long-context interleaved video-language understanding." NIPS (2024): 28828-28857.
> 3. Song, Enxin, et al. "Moviechat: From dense token to sparse memory for long video understanding." CVPR. 2024.
> 4. Team, Gemini, et al. "Gemini 1.5: Unlocking multimodal understanding across millions of tokens of context." arXiv preprint arXiv:2403.05530 (2024).
> 5. Wang, Bin, et al. "Audiobench: A universal benchmark for audio large language models." arXiv preprint arXiv:2406.16020 (2024).
> 6. Ahia, Orevaoghene, et al. "BLAB: Brutally Long Audio Bench." arXiv preprint arXiv:2505.03054 (2025).
> 7. Wang, Weiyun, et al. "Needle in a multimodal haystack." NIPS (2024): 20540-20565.
> 8. Wu, Tsung-Han, et al. "Visual haystacks: A vision-centric needle-in-a-haystack benchmark." arXiv preprint arXiv:2407.13766 (2024).
> 9. Abouelenin, Abdelrahman, et al. "Phi-4-mini technical report: Compact yet powerful multimodal language models via mixture-of-loras." arXiv preprint arXiv:2503.01743 (2025).

---

> > ### Comment · Reviewer_aW4B · 2025-08-02
> >
> > Thanks for the comment, please incorporate the justifications (1-3) above in the paper. I have no further comments.

---

> > > ### Author Response · Authors · 2025-08-03
> > > **Response to Reviewer aW4B**
> > >
> > > Thank you for your response and for acknowledging our detailed justifications. We will incorporate these three points in our final revision.
> > >
> > > **We have carefully addressed all concerns and believe that our paper offers a solid contribution to the field:** including examples from 16 datasets across five categories, evaluating 46 different LVLMs, and providing comprehensive insights. If you feel that the revised version meets the criteria for a stronger endorsement, **we would be truly grateful if you could consider increasing your recommendation to a higher level.**
> > >
> > > Thank you for your consideration.

---

### Note · Authors · 2025-08-13

We sincerely thank all reviewers for their detailed and constructive feedback on our manuscript. Before the rebuttal, all four reviewers had already rated our work with scores of 5 (accept). After the rebuttal, we are glad to see that **we have resolved all the reviewers' concerns.** Meanwhile, Reviewer Uv7t and by6z increased their scores to 6 (strong accept). Furthermore, all reviewers have acknowledged our pioneering contributions and recognized the efforts and strengths of our work:

1. Comprehensive evaluation of SoTA models (Reviewer aW4B, Uv7t, and by6z)
2. Diverse and well-designed long-context tasks (Reviewer aW4B and by6z)
3. Extensive insightful findings (Reviewer aW4B, Uv7t, and XLhL)
4. Filling a critical research gap of multimodal long context (Reviewer Uv7t)
5. Rigorous context length control for clear benchmarking (Reviewer Uv7t)
6. Commendable and well-organized presentation (Reviewer Uv7t and XLhL)
7. Clear motivation regarding limitations of existing works (Reviewer XLhL)
8. Including new tasks for multimodal long context (Reviewer XLhL)

Specifically, we have clearly explained the following concerns:

1. Add more models to confirm that reasoning models perform better (Reviewer Uv7t): Besides the three reasoning LVLMs in our current manuscript, we also added two new reasoning models to further support this finding (MiMo-VL-7B-SFT and Kimi-VL-A3B-Thinking).
2. More discussion about summarization evaluation with GPT-4o (Reviewer Uv7t and XLhL): In the rebuttal, we provide extensive evidence to show that the GPT-4o is not biased. Also, we tested two more models (Qwen2.5-7B-Instruct and Qwen2.5-72B-Instruct) as the judge, providing a more comprehensive discussion.
3. Add comparison with the existing benchmarks (Reviewer by6z): We provide a comprehensive comparison between our benchmark and existing ones, including MM-NIAH and Visual Haystack. We observed consistent findings across benchmarks, showing the correctness of our data.
4. In our rebuttal, we also provided detailed explanations for the long video and audio data (Reviewer aW4B and XLhL) and the many-shot ICL task design (Reviewer XLhL).

Once again, we sincerely thank all the reviewers for their thorough evaluation of our manuscript. Their valuable insights have significantly contributed to improving our work. **Our comprehensive responses addressed all the concerns** and further validated the significance of our research contribution.

Best Regards,

Authors from Submission 330

---

### Decision · Program_Chairs · 2025-09-18

**Decision:**

Accept (spotlight)

**Comment:**

This paper aims to address a gap in the current literature, i.e. benchmarks are lacking to assess if current MLLMs possess  long context windows.  The proposed MMLongBench is large scale, covering several tasks. The evaluation is rich, comprising both proprietary and open-source models. All the reviewers were very positive with the paper since the beginning. After the rebuttal, some reviewers also decided to raise their score. The AC agrees that this is a solid submission and recommends acceptance.